# Twenty-four-hour rhythmicities in disorders of consciousness are associated with a favourable outcome

Florent Gobert [1,2,3 ✉], Alexandra Corneyllie[3], Hélène Bastuji[4,5], Christian Berthomier[6], Marc Thevenet[3], Jonas Abernot[3], Véronique Raverot [7], Frédéric Dailler[1], Claude Guérin [8,9], Claude Gronfier [10], Jacques Luauté[2,11,12] & Fabien Perrin[3,12]

Fluctuations of consciousness and their rhythmicities have been rarely studied in patients with a disorder of consciousness after acute brain injuries. 24-h assessment of brain (EEG), behaviour (eye-opening), and circadian (clock-controlled hormones secretion from urine) functions was performed in acute brain-injured patients. The distribution, long-term predictability, and rhythmicity (circadian/ultradian) of various EEG features were compared with the initial clinical status, the functional outcome, and the circadian rhythmicities of behaviour and clock-controlled hormones. Here we show that more physiological and favourable patterns of fluctuations are associated with a higher 24 h predictability and sharp up-and-down shape of EEG switches, reminiscent of the Flip-Flop model of sleep. Multimodal rhythmic analysis shows that patients with simultaneous circadian rhythmicity for brain, behaviour, and hormones had a favourable outcome. Finally, both re-emerging EEG fluctuations and homogeneous 24-h cycles for EEG, eye-opening, and hormones appeared as surrogates for preserved functionality in brainstem and basal forebrain, which are key prognostic factors for later improvement. While the recovery of consciousness has previously been related to a high short-term complexity, we suggest in this exploratory study the importance of the high predictability of the 24 h long-term generation of brain rhythms and highlight the importance of circadian body-brain rhythms in awakening.

[1] Neuro-Intensive care unit, Hospices Civils de Lyon, Neurological hospital Pierre-Wertheimer, 59 Boulevard Pinel, Bron, France. [2] Trajectoires Team, Lyon Neuroscience Research Centre (Université Claude Bernard Lyon 1, INSERM U1028, CNRS UMR5292), Bâtiment Inserm 16 avenue Doyen Lépine, Bron, France. [3] CAP Team (Cognition Auditive et Psychoacoustique), Lyon Neuroscience Research Centre (Université Claude Bernard Lyon 1, INSERM U1028, CNRS UMR5292), 95 boulevard Pinel, Bron, France. [4] Sleep medicine centre, Hospices Civils de Lyon, Bron F-69677, France. [5] Neuropain Team, Lyon Neuroscience Research Centre (Université Claude Bernard Lyon 1, INSERM U1028, CNRS UMR5292), 59 Boulevard Pinel, Bron, France. [6] PHYSIP SA, Paris, France. [7] Hormone Laboratory, Hospices Civils de Lyon, Neurological hospital Pierre-Wertheimer, 59 Boulevard Pinel, Bron, France. [8] Intensive care unit, Hospices Civils de Lyon, Croix-Rousse hospital, 103 Grande-Rue de la Croix-Rousse, Lyon, France. [9] Intensive care unit, Hospices Civils de Lyon, Édouard Herriot hospital, 5 Place d'Arsonval, 69003 Lyon, France. [10] Waking team (Integrative Physiology of the Brain Arousal Systems), Lyon Neuroscience Research Centre, INSERM UMRS 1028, CNRS UMR 5292, Université Claude Bernard Lyon 1, Université de Lyon, Lyon, France. [11] Neuro-rehabilitation unit, Hospices Civils de Lyon, Neurological hospital Pierre-Wertheimer, 59 Boulevard Pinel, Bron, France. [12] These authors contributed equally: Jacques Luauté, Fabien Perrin. ✉email: florent.gobert01@chu-lyon.fr

How would you guess in the blink of an eye that someone is asleep rather than in a coma? Although this usually becomes self-evident after a few stimulations, such an assessment remains trickier in comatose individuals suffering from severe brain damages that alter the spontaneous behavioural expressions of internal states, the responsiveness to environment including the communication of self-report, and wakefulness.

For healthy individuals, normal consciousness is usually described as a two-dimensional process[1], associating the level of consciousness (i.e. wakefulness or arousal) and the content of consciousness of both the environment and the self, (i.e. awareness[2]).

This static description of consciousness may seem operational for healthy participants, but research in comatose and post-comatose patients shows that the assessment of consciousness modulations is dependent on the preservation of three complementary dynamic features: i) the access to complex cognitive functions[3], ii) the quality of arousal reactivity (defined as "an upward change in the level of alertness[4]"), and iii) the physiological rhythmicity of wakefulness during the sleep-wake cycle as our group has recently shown[5]. In physiology, the transient disappearance of consciousness during sleep[6] is in phase with the environment and is confirmed by the eye-opening behaviour, assuming that both dimensions vary linearly (for non-Rapid Eye Movement [NREM] sleep to aware wakefulness, but not REM sleep during which, by definition, modulations differ paradoxically).

Normal variations within levels of wakefulness are physiologically determined by several constraints. First, they are strongly influenced by the genetically-based circadian rhythmicity, which is generated in the suprachiasmatic nucleus [SCN] of the hypothalamus[7] and is conveyed to virtually all organs and cells throughout the organism (including the sleep-wake regulation structures[8]) to time precisely their activity over 24 h. Second, the homoeostatic regulation of the sleep process[6] is related to the intrinsic history of brain metabolism during wake[9] and allows a functional and structural maintenance. Third, wakefulness allows to react quickly to environmental constraints/stimuli, this ability being usually associated with the functionality of the ascending arousal network[10]. Complementary hypothalamic functions however, might be involved in wakefulness modulations: the SCN, which is sensitive to day/night-light/dark long-range fluctuations[11] can also present rapid modulations after short light exposure during sensitive phases in the night[12–15] but also over 24 h[16]. Overall, wakefulness and sleep result from the interplay between internal processes and external influences that lead to EEG fluctuations at different scales.

Brain damages may alter both the genesis of endogenous rhythms and their ability to be synchronised with environmental cues. The comatose state is the starting phenotype of the Disorders Of Consciousness [DOC] group and is defined by a transient loss of both wakefulness and awareness at the acute stage of brain injury (no reactivity to stimuli and no clue of wakefulness defined by the absence of any eye-opening moments[2]). Classically, the occurrence of eye-opening moments has been used as a sign for the re-appearance of sleep-wake cycles[17], which was used to define the Vegetative State [VS], now referred to as the Unresponsive Wakefulness Syndrome [UWS][18]. Nevertheless, it remains debated whether the behavioural assessment of eyelid movements is a reliable marker of wakefulness fluctuations because the heterogeneity of lesions could imply dissociated patterns (e.g. default of the output to oculomotor nuclei commanding eye-opening after strategic mesencephalic lesions)[19]. Indeed, several questions remain regarding the relationship between cerebral fluctuations, behavioural changes in wakefulness clues, clinical status, and outcome. Moreover, the chronology of changes in each consciousness dimension, whose classical ordering was recently debated by the attribution of circadian rhythms as a prerequisite for awareness rather than wakefulness[5], also remains to be studied. Thus, can we affirm that brain rhythms are not present before the appearance of the eye-opening moments? Would the main pattern of fluctuations be rather non-rhythmic, circadian (defined herein by a 24-h rhythmicity without the presumption of a purely endogenous SCN drive), or ultradian (defined as alternative cyclic patterns whose periods are shorter than 24 h)?

The present study aimed to evaluate, in acute DOC patients, the potential relationship between the recovery of consciousness in its different dimensions and the reappearance of rhythmicity among cerebral, biological, and behavioural fluctuations (analysed in their continuity during a complete nychthemeral period). First, we hypothesised that the characteristics of DOC patients could be related to long-term fluctuations (defined in this setting as being measurable at the scale of a 24-h EEG). Notably, we anticipated that the return of a conscious state would not only be associated with short-term EEG features assessing the local cerebral complexity (as it was suggested in the literature[20]) but also with *24 h long-term variables* assessing an increased predictability of brain activity (measured by low values of entropy variables which allows to evaluate fluctuation reproducibility over time) as a proof of a recovered periodic alternation of functional states (Analysis N°1). Second, we hypothesised that the recovery of brain circadian rhythms could be associated with the recovery of behaviour and/or clock-controlled hormone circadian rhythms in acute DOC patients (Analysis N°2).

We eventually demonstrate that multimodal 24-h long assessment is a complementary tool for late outcome prediction. Thus, it may catch a peculiar dimension of a basic brain functionality that could be seen as a foundation for more advanced cognitive recovery. At the group level, the presence of highly predictable *24 h long-term EEG patterns* was associated with a favourable outcome in terms of consciousness emergence. Remarkably, categorising patients depending on the presence and quality of circadian rhythms using an integrated multimodal recording (i.e. for EEG, behaviour, and hormones, coined as HOMOGENEOUS PRESENCE OF ALL CIRCADIAN RHYTHMS) enabled to identify a small group of patients who systematically regained consciousness (with a low sensitivity but a high positive predictive value).

## Results

In brief (see Fig. 1 and the Methods section for details), synchronised assessments of behavioural (continuous video of eye-opening/closing using a dual-camera recording), environmental (nursing periods assessed by continuous video, sound and light levels assessed by continuous sonometer and luxmeter recordings respectively), hormonal (urinary melatonin/cortisol every 2 h, to assess the endogenous drive by clock-controlled hormones), and neurophysiological markers of brain function (13-channel EEG) were performed in 18 acute DOC patients.

We evaluated *24 h long-term variables* (*24-h distribution*, *24 h long-term predictability*, and circadian/ultradian rhythmicity) of various short-term EEG features, the combination of which led to the analysis of 168 EEG parameters. The EEG features were selected through a hypothesis-free approach on the basis of their complementarity to describe EEG in its various dimensions and on their previous use in DOC patients. The short-term EEG features retained were then grouped as power[21], spectrum[20,22–25], complexity[20,24–31], and spatial variability[32] (see Fig. 1 for description). Rhythmicity was also calculated for eye-opening periods (circadian and ultradian) and for hormonal results

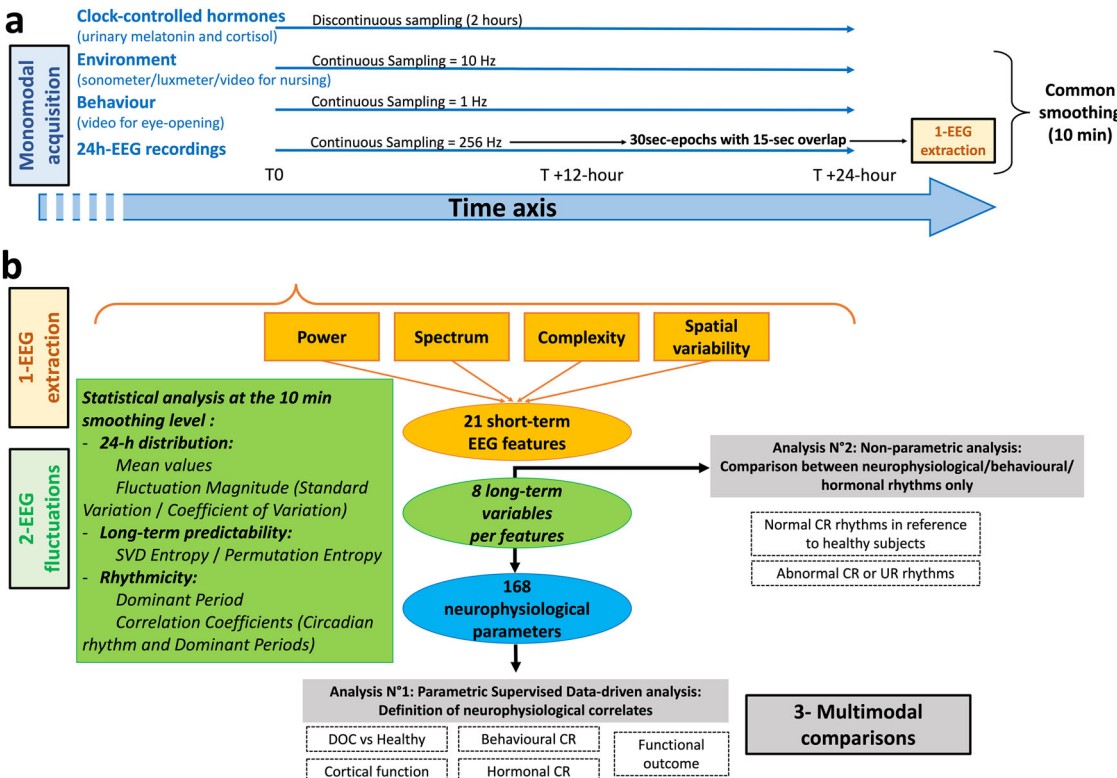

**Fig. 1 Methodological overview.** Pipeline describing the method of acquisition (**a**) for synchronised assessments (blue) of behavioural, environmental, hormonal, and neurophysiological markers of wakefulness. The EEG metrics were then pre-processed according to a step-by-step protocol (**b**) to be compared with other modalities. Concerning EEG, 21 features (orange) were calculated on a 30-s scale to describe the EEG signal to look for an ultradian or circadian rhythmicity (see Step 3 -Multimodal comparisons) and regrouped as: - Power (with Total Power and Variance). - Spectrum (with the values of Absolute and Relative Power for Delta/Theta/Alpha/Beta bands, the Alpha Dominant Frequency, and an autoregressive modelling of low (4th) order [AR4 model] summarising the spectral fluctuation in magnitude and frequency). - Complexity (with Determinism as the probability for the occurrence of recurrent patterns, Singular Value Decomposition [SVD]-entropy as a temporal entropy metric and Detrended Fluctuation Analysis [DFA] as the geometrical precision required to describe EEG). - Spatial variability (assessed as the standard deviation across the full bipolar montage considering the Relative Power in each band). After a 10-min smoothing, EEG fluctuations were described by *24 h long-term variables* (green) regrouped as: - *24 h distribution variables*: *Mean* assessing the relative values of each feature; *Standard Deviation* assessing the absolute magnitude of the feature fluctuation; *Coefficient of Variation* assessing the relative magnitude of the feature fluctuation compared to the mean. - *24 h long-term predictability variables*: based on Information theory, *SVD-entropy* and *Permutation Entropy* assessed the *long-term complexity of the fluctuations* (following the general rule "the lower the complexity, the higher the predictability"). - *Rhythmicity variables*: the *Correlation Coefficient of the Circadian Period* indicated the fit with a sinusoid waveform presenting a period between 23.5 and 24.5 h; to catch ultradian fluctuations as well, the *Correlation Coefficient of the Dominant Period* was calculated for the *Dominant Period* (± 5%) extracted from a spectral analysis of each smoothed curve. In the Step 3 - Multimodal comparisons, the 168 EEG parameters (blue) combining the 21 features * 8 variables were used in a supervised data-driven analysis to find out the EEG correlates for clinical and circadian rhythmicity factors (Analysis N°1) and each modality were used for the multimodal rhythmic analysis (Analysis N°2) focused on the homogeneity of circadian rhythms.

(circadian only due to a difference in the sampling rate with that of EEG and behaviour). Rhythmicity was then dichotomised as present or absent for each of these modalities according to circadian and ultradian fits.

First, a supervised data-driven analysis was used to evaluate the EEG fluctuations (Analysis N°1) that were indicative of: i) clinical factors, namely the initial behavioural cortical function status (based on the CRS-R) and the final functional outcome (based on the Glasgow Outcome Scale [GOS]); ii) circadian rhythmicity factors, namely for eye-opening behaviour and clock-controlled hormones[5]. Of note, the combination of various parameters was specific for each factor. Second, circadian and ultradian rhythmicities regarding brain (from EEG), eye-opening moments (from video), and hormones (from urinary clock-controlled excretion) were also evaluated with the aim to classify patients according to their proximity with normal rhythmicity patterns (Analysis N°2). Third, the prognostic value of the simultaneous presence of these 3 circadian rhythms was also compared in Analysis N°2 to usual markers to predict a favourable outcome,

defined as a full recovery of consciousness (i.e. patients reaching the Exit-Minimally Conscious State [MCS] level, including some cases with severe disability).

Overall, 3 healthy participants and 19 patients were included; 18 acute patients in ICU with a clinical outcome assessed 2 years after the brain injury by the GOS, 2 of whom underwent 2 recordings, and 1 chronic patient. A favourable outcome was observed for 9 acute patients.

**Description of EEG fluctuation levels.** Using an exploratory visual approach, the qualitative observations of EEG patterns enabled to evidence a dichotomic separation among patients (fluctuating versus non-fluctuating patterns; Fig. 2a). A skilled sleep scorer (H.B.) failed to build complete and homogeneous hypnograms by classifying each 30-sec EEG pages as sleep stages and wake moments using classical rules[33,34]. The exploratory analysis however, showed that fluctuations – unlike mean values – appeared to be related to the outcome (Fig. 2b, c). As the amplitude of the fluctuations alone was not able to predict the

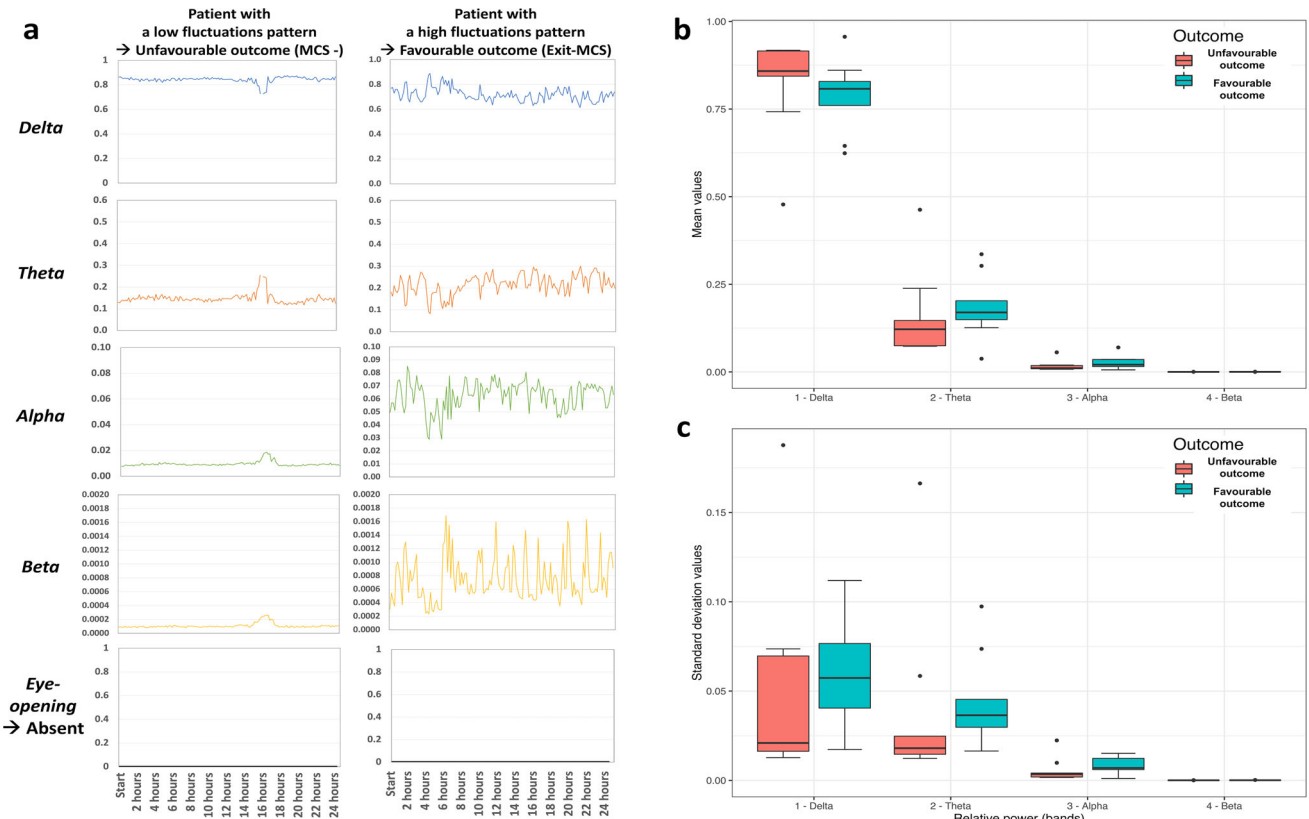

**Fig. 2 Illustration of exploratory analysis.** The patterns of *24 h long-term time series* were qualitatively described as "complex, chaotic, unpredictable", "cyclic", or "steady". **a** Examples of opposite dissociations between EEG and behavioural patterns with a common X axis = 24 h for two patients. Identification from Fig. 6: Patient N°12 (first recording) on the left, Patient N°10 on the right. Both patients had their eyes constantly closed defining a confirmed comatose state (the steady pattern of eye-opening moments in black is common to both patients). In each case, a different cyclic pattern was observed with: i) on the left, a unique and common fluctuation; ii) on the right, pseudo-ultradian fluctuations for the Beta Relative Power and a complex, chaotic, unpredictable pattern for Delta and Theta Relative Power. Of note a period is missing for Patient 12 (first recording) on the left in Delta and Theta curves due to an artefactual period. Spectral analysis using Relative Power for the 4 canonical bands illustrated by mean values (**b**) and standard deviations (**c**) and presented according to the outcome ($n = 18$, $n = 9$ in each group). The fluctuating patterns qualitatively observed in A could be more accurately objectified by a metric of fluctuation amplitude (Theta and Alpha standard deviation for example) rather than the differences between mean values. However, the Delta mean value was associated with outcome while the Delta standard deviation was not.

outcome accurately enough (Fig. 2c), complementary metrics assessing this visual aspect were used. This discrimination was further explored by using original features of the EEG signal fluctuations.

**Clinical status and outcome were related to the *24 h long-term predictability* of broad fluctuations (Analysis N°1).** The presence of the "Behavioural Cortical Function" factor (Figs. 3a and 4a, with 4 occurrences vs 17 without) was associated with 3 different EEG parameters (*Akaike Information Criterion [AIC]* = 17.23; $p = 0.01028$; *False Positive Risk [FPR]* = 0.0925; *Likelihood ratio [LR]* = 9.81). These 3 parameters (detailed in Fig. 4) indicated that, in case of "Behavioural Cortical Function" in patients at inclusion, the fluctuations of EEG signal were consistently described by *a higher long-term predictability* (for power, spectrum, and short-term EEG complexity; Fig. 4c).

The presence of the final "Functional Outcome" prognostic factor (Figs. 3b and 4a, with 9 occurrences of favourable outcome vs 9 unfavourable outcome) was associated with 3 EEG parameters. These 3 parameters (detailed in Fig. 4) were related to the dynamic in high frequency (*AIC* = 8; $p = 0.00422$; *FPR* = 0.0451; *LR* = 21.18) as the favourable "Functional Outcome" was associated with *a higher magnitude of fluctuation within the 24-h distribution and a higher long-term predictability*

(for spectrum and spatial variability, in particular in the high frequency bands; Fig. 4c).

**The circadian-related factors were inconstantly related to the rhythmic fluctuations of EEG features (Analysis N°1).** The presence of the "Behavioural Circadian Rhythmicity" factor (i.e., the circadian rhythmicity for eye-opening/closing; Figs. 5a and 4b, with 8 occurrences of "Behavioural Circadian Rhythmicity" vs 13 without) was associated with 3 EEG parameters (*AIC* = 8; $p = 0.00432$; *FPR* = 0.0411; *LR* = 23.33). These 3 parameters (detailed in Fig. 4) indicated *a lower long-term predictability* (for spatial variability), *a higher magnitude of fluctuation within the 24-h distribution and a shorter period of rhythmicity* (for spectrum; Fig. 4c). Notably, the group of patients without eye-opening circadian rhythmicity seemed heterogeneous as their 3-dimensional cluster was wide because of a higher dispersion regarding the *Dominant Period* of Delta Relative Power.

The presence of the "Hormonal Circadian Rhythmicity" factor (i.e., the circadian rhythmicity for clock-controlled hormonal secretion; Figs. 5b and 4b, with 9 occurrences of "Hormonal Circadian Rhythmicity" vs 10 without) was associated with 3 EEG parameters (*AIC* = 8; $p = 0.00045$; *FPR* = 0.0149; *LR* = 65.93). These 3 parameters (detailed in Fig. 4) were all related to the *rhythmicity variables* (for spatial variability and spectrum).

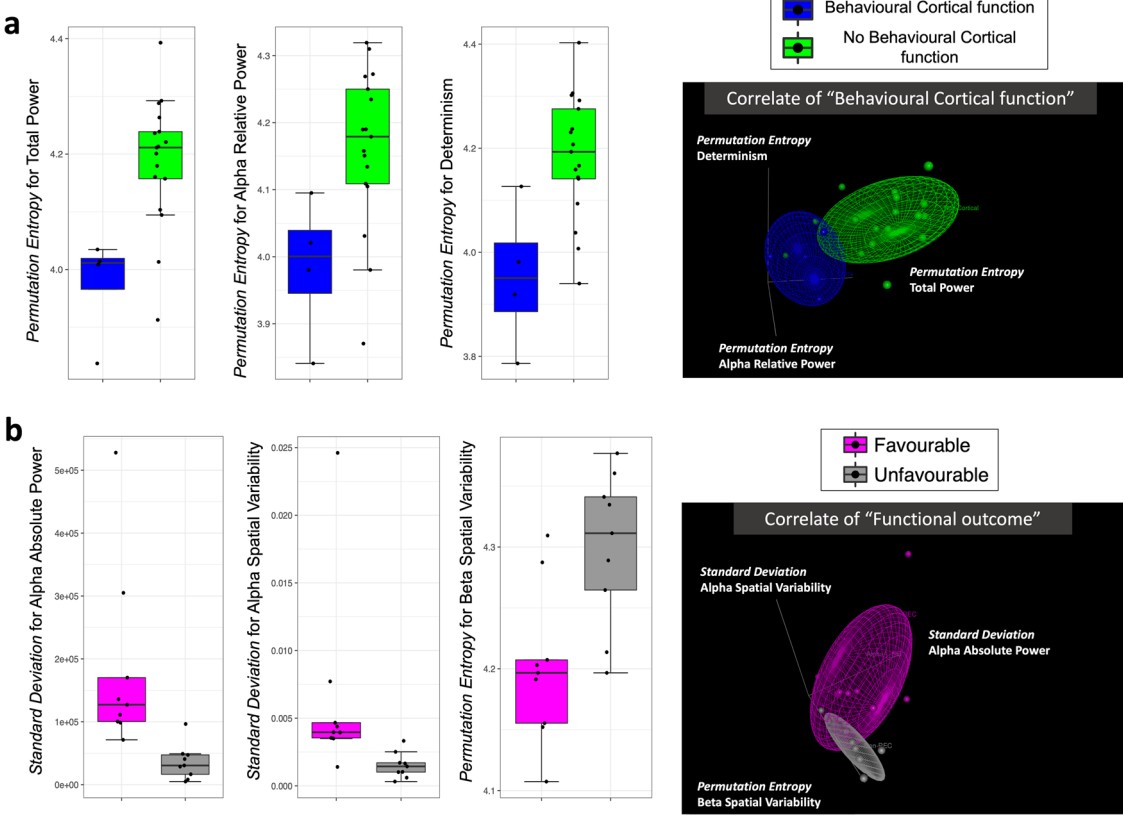

**Fig. 3 Supervised data-driven analysis concerning the parameters related to clinical factors (Analysis N°1).** Multimodal classification of patients based on EEG parameters to perform a dichotomic segregation for 2 clinical factors. **a** "Behavioural Cortical Function" at inclusion ($n = 21$ biologically independent samples, presence in blue, $n = 4$, vs absence in green, $n = 17$). Patients with behavioural cortical function, compared to those without, had significantly lower *Permutation Entropy* values for Determinism, Total Power, and Alpha Relative Power. **b** Final "Functional Outcome" ($n = 18$ biologically independent samples, Exit-MCS in magenta, $n = 9$, vs grey for unfavourable outcome, $n = 9$). The patients with a favourable outcome had a higher *Standard Deviation* for Alpha Absolute Power and for Alpha Spatial Variability compared to the patients with an unfavourable outcome. They also had a lower *Permutation Entropy* for Beta Spatial Variability. Among the parameters univariately associated with the factor using a binomial logistic regression, the models with the 3 most accurate parameters were selected by a backward method minimising the AIC parameter to allow the comparison between each factor. The relative distance between clusters is illustrated on the right (groups illustrated by coloured ellipsoids; patients illustrated by coloured dots of different sizes as a 3D-effect) and details for each parameter are illustrated by boxplots on the left. The selected parameters were specific to each factor.

Notably, *a short period of rhythmicity with a tendency toward an ultradian fit* was specifically observed in high-frequency bands when the presence of an endogenous circadian rhythm was demonstrated (Fig. 4c).

**DOC classification can be refined using their multimodal rhythmic pattern (Analysis N°2).** An overview of these results is presented in Fig. 6 while more details regarding the categorisation are presented in Supplementary Fig. 1. Representative patients are illustrated in Fig. 7 and in Supplementary Figs. 2–4. Supplementary Figure 5 illustrates the coexistence between normal and abnormal circadian rhythms among patients.

Some insights could be outlined from the pattern of HOMOGENEOUS PRESENCE OF ALL CIRCADIAN RHYTHMS. For each brain (i.e. EEG), behavioural (i.e. eye-opening), and hormonal assessment, the individual description (columns in Fig. 6) enabled to categorise patients according to the homogeneity of the circadian rhythm pattern, by defining a HOMOGENEOUS PRESENCE OF ALL CIRCADIAN RHYTHMS (orange-frame), a homogeneous absence of circadian rhythms (green-frame), and a "heterogeneity in the presence of circadian rhythms" (blue-frame). The analysis of the phase angle between each circadian rhythm was consistent with a close phase relationship between EEG/behavioural rhythms and

at least one clock-controlled hormone in 6 out of 7 patients (see Supplementary Data 5).

Compared to the initial clinical assessment by CRS-R (Fig. 6), the HOMOGENEOUS PRESENCE OF ALL CIRCADIAN RHYTHMS appeared as a powerful marker of favourable outcome for acute patients. It predicted a complete consciousness recovery (i.e. Exit-MCS with communication ability) with a 100% positive predictive value (PPV). This information was not redundant with the classifications based on behaviour as 2 acute patients were UWS/VS and 1 was MCS/CMS. One chronic UWS/VS patient presented this pattern as well. None of the patients categorised as Coma (3 patients without eye-opening during the CRS-R including 2 patients with eyes constantly closed during the 24-h recording) presented this favourable profile.

The predictive value of HOMOGENEOUS PRESENCE OF ALL CIRCADIAN RHYTHMS can be compared to classic prognostic markers: the predictive value of this qualitative pattern HOMOGENEOUS PRESENCE OF ALL CIRCADIAN RHYTHMS was assessed in association with classic clinical and neurophysiological dichotomic prognosis markers such as EEG reactivity, bilateral abolition of cortical components, morphological alteration of Evoked Potentials [EPs], presence of N100 (Negative response at 100 ms for oddball auditory stimulus)/Mismatch Negativity[MMN]/P300 (Positive response at 300 ms for subject own name stimulus) to test the hypothesis

**a**

| Factor | "Behavioural cortical function" | | "Functional outcome" | |
|---|---|---|---|---|
| Group | Present | Absent | Favourable | Unfavourable |
| Parameter 1 | Permutation Entropy for Total Power (p = 0.0418) | | Standard Deviation for Alpha Absolute Power (p = 0.0380) | |
| Values | 3.97 (±0.09) | 4.19 (±0.11) | 183098 µV² (±146175) | 35905 µV² (±27793) |
| Parameter 2 | Permutation Entropy for Alpha Relative Power (p = 0.0444) | | Standard Deviation for Alpha Spatial Variability (p = 0.0184) | |
| Values | 3.98 (±0.11) | 4.16 (±0.12) | 0.006 (±0.0070) | 0.002 (±0.0009) |
| Parameter 3 | Permutation Entropy values for Determinism (p = 0.0382) | | Permutation Entropy for Beta Spatial Variability (p = 0.0222) | |
| Values | 3.95 (±0.14) | 4.18 (±0.12) | 4.20 (±0.06) | 4.30 (±0.06) |

**b**

| Factor | "Behavioural circadian rhythmicity" | | "Hormonal circadian rhythmicity" | |
|---|---|---|---|---|
| Group | Presence of CR | Absence of CR | Presence of CR | Absence of CR |
| Parameter 1 | Standard Deviation for AR4 Total (p = 0.0442) | | Dominant Period for Beta Relative Power (p = 0.0480) | |
| Values | 2.22 (±0.41) | 1.67 (±0.54) | 5.18h (±4.57) | 12.44h (±7.44) |
| Parameter 2 | Permutation Entropy for Theta Spatial Variability (p = 0.0449) | | Correlation Coefficient of the Dominant period for Beta Spatial Variability (p = 0.0247) | |
| Values | 4.28 (±0.06) | 4.15 (±0.13) | 0.299 (±0.1441) | 0.063 (±0.1130) |
| Parameter 3 | Dominant Period for Delta Relative Power (p = 0.0447) | | Correlation Coefficient of the Dominant period for Delta Spatial Variability (p = 0.0419) | |
| Values | 5.85h (±4.81) | 15.48h (±9.84) | 0.091 (±0.1397) | 0.269 (±0.1660) |

**c**

| Long-term EEG analysis : VARIABLES | | Short-term EEG analysis : FEATURES | | | |
|---|---|---|---|---|---|
| | | Power | Spectral analysis | Signal complexity | Spatial variability |
| | Mean values | | | | |
| | Fluctuation Magnitude | | . Behavioural circadian rhythmicity . Functional Outcome | | .. DOC status . Functional Outcome |
| | Fluctuation Complexity | . Behavioural cortical function | . Behavioural cortical function | . Behavioural cortical function | . Behavioural circadian rhythmicity . Functional Outcome |
| | Fluctuation Rhythms | | . Behavioural circadian rhythmicity . Hormonal circadian rhythmicity | . DOC status | . Hormonal circadian rhythmicity |

**Fig. 4 Numerical and synthetic results of Analysis N°1.** Details for the parameters related to the clinical factors in **a** (from Fig. 3) and to the circadian rhythmicity factors in **b** (from Fig. 4). No parameter was independently associated with these factors in multivariate logistic regression. Values: mean (±SD) compared between groups with their respective p-values (univariate binomial logistic regression). A green shade indicates the type of variable (light green for magnitude, standard green for complexity, and dark green for rhythmicity). The units are given in the table (hours, µV2) when available (except for correlations, relative power ratio and derivates, complexity measures). **c:** Synthesis of the most pertinent short-term features (orange) and *24 h long-term variables (green)*. A green shade indicates the type of variable (light green for magnitude, standard green for complexity, and dark green for rhythmicity). A blue shade indicates the number of occurrences for all factors (light blue = 1, azure = 2, dark blue = 3). CR Circadian Rhythms, DOC Disorders of consciousness.

that an original (i.e. non-redundant) predictive value was associated with this qualitative pattern (Fig. 6b). No marker reached the statistical threshold to predict, on its own, a favourable outcome (Supplementary Table 3, left-part). The HOMOGENEOUS PRESENCE OF ALL CIRCADIAN RHYTHMS pattern did not reach the significance threshold (Se = 37.5%; Fisher's Exact test, $p = 0.2$). The presence of a behavioural cortical function (according to the CRS at the recording) was a predictor of a favourable outcome with a 100% PPV but a 33% sensitivity (Fisher's Exact test, $p = 0.206$). The absence of EEG reactivity was the best predictor of an unfavourable outcome (NPV = 100%) but had a 44% specificity (Fisher's Exact test, $p = 0.082$). The combined marker (presence of a behavioural cortical function or HOMOGENEOUS PRESENCE OF ALL CIRCADIAN RHYTHMS) had a sufficient sensitivity (Se = 55.6%) to reach significance (Fisher's Exact test, $p = 0.029$; FPR = 0.1424; LR = 6.02).

A classification-tree based on a combination of dichotomic factors (initial status, EEG reactivity, mechanism of lesions, and the criterion HOMOGENEOUS PRESENCE OF ALL CIRCADIAN RHYTHMS) may delineate original boundaries for the prediction of a favourable or unfavourable outcome by minimising the grey-zone of uncertainty (Fig. 8).

Further results regarding the radiological features of patients' lesions, (Supplementary Table 2 and Supplementary Data 1), the correlations between features (Supplementary Table 3 and Supplementary Data 2), the EEG correlate of DOC (Supplementary Fig. 6 and Supplementary Data 3), the environmental recordings (Supplementary Table 4 and Supplementary Data 4), the phase angle between circadian rhythms (Supplementary Data 5), the existence of circadian rhythm among patients that were absent from healthy subjects (Supplementary Data 6), the qualitative dichotomic parameters (Supplementary Table 5 and Supplementary Data 7), the quantitative continuous parameters related to the most powerful prognostic parameters observed in the data-driven analysis (Supplementary Table 6 and Supplementary Data 8) and are available in the Supplementary Information.

## Discussion

To understand the way out of coma, we proposed to explore the *24-h long-term fluctuations (variables)* of the cerebral activity (EEG features) in link with those of the body (clinical scales, eye-

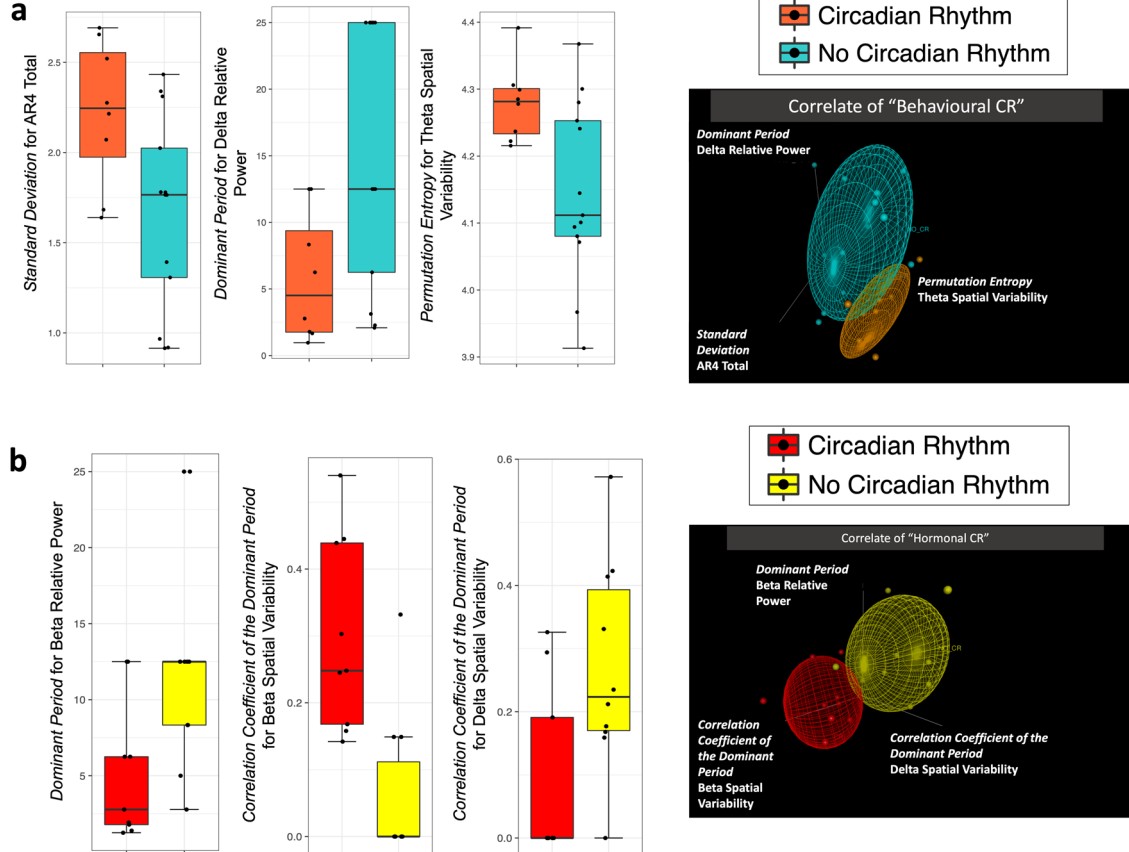

**Fig. 5 Supervised data-driven analysis concerning the circadian rhythmicity factors (Analysis N°1).** Multimodal classification of patients based on EEG parameters to perform a dichotomic segregation for 2 circadian rhythmicity factors. **a** "Behavioural Circadian Rhythmicity" ($n = 21$ biologically independent samples, presence of eye-opening circadian rhythms in orange, $n = 8$, vs cyan for absence, $n = 13$). The patients with a 24-h pattern of behavioural wakefulness presented a higher *Permutation Entropy* for Theta Spatial Variability compared to the patients without circadian rhythmicity and a higher *Standard Deviation* for AR4 Total. They also had a lower *Dominant Period* for Delta Relative Power. **b** "Hormonal Circadian Rhythmicity" ($n = 19$ biologically independent samples, presence of a circadian rhythm for at least one clock-controlled hormone in red, $n = 9$, vs yellow for absence, $n = 10$). For patients presenting circadian rhythms of clock-controlled hormones, the *Correlation Coefficient of the Dominant period* was higher for Beta Spatial Variability compared to other patients and lower for Delta Spatial Variability. The *Dominant Period* for Beta Relative Power was lower compared to patients without hormonal circadian rhythms. Among the parameters univariately associated with the factor using a binomial logistic regression, the models with the 3 most accurate parameters were selected by a backward method minimising the AIC parameter to allow the comparison between each factor. The relative distance between clusters is illustrated on the right (groups illustrated by coloured ellipsoids; patients illustrated by coloured dots of different sizes as a 3D-effect) and details for each parameter are illustrated by boxplots on the left. The selected parameters were specific to each factor.

opening/closing moments, and urinary clock-controlled hormones).

Contrary to previous studies that have focused on sleep-wake classification to delineate the wakefulness changes during the nychthemeral period in chronic DOC[35], we were unable to build such a classical and synthetic view in a way that remained consistent at the population level. In contrast, choosing a mathematical fluctuation description allowed us to explicit a general rule. In addition, using linear descriptions of EEG features in Analysis N°1 enabled to avoid defining sharp thresholds of classification based on common[36] or specific[35] minimal requirements on 30-sec EEG epochs. In other words, we assumed a fine-grained mathematisation of brain function fluctuations – revealing its real shape and complexity – instead of conforming it to previous arbitrary ordinal categorisation without nuance.

Assessing these fluctuations from "brain and body" is technically challenging (because it implies prolonged assessments and large amounts of data that may be difficult to interpret because of confounding factors, e.g. external movements induced by nursing, noisy environment, medication, and intercurrent diseases) but also theoretically challenging (because the relationship with

outcome is not straightforward as wakefulness modulation is not cortically mediated). However, it appeared that a favourable outcome was predicted at the individual level and that it was clinically relevant because it was limited to the Exit-MCS group, contrary to some previous studies[37,38] which considered reaching the MCS outcome as favourable.

Considering cerebral complexity against predictability, as a matter of temporal scale (Analysis N°1), we observed that previous studies have proposed that, among a large set of features, a high level of short-term EEG complexity was a marker of awareness[20,31,39]. In DOC patients, short-term complexity decreased during the changes of wakefulness leading to a loss of awareness[31]. In a 30-min protocol developed by Sitt et al., the mean values of complexity (standing for a global metric) and the standard deviation values (standing for a *long-term metric*, at the minutes-of-EEG scale) appeared complementary[20]. Indeed, the MCS/CMS patients presented the same pattern as those herein with *low long-term fluctuations* (i.e. standard deviation) of high short-term complexity (i.e. mean values of the permutation entropy in the Theta and Alpha bands). The *long-term fluctuations of complexity* have been previously analysed using spectral

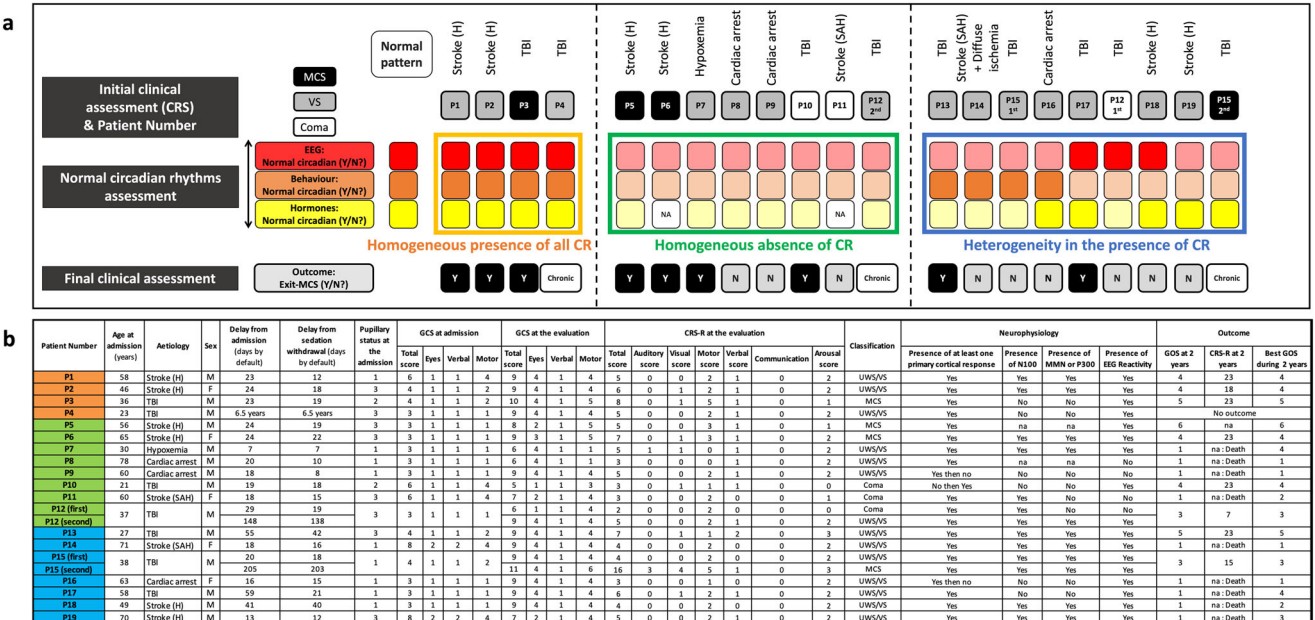

**Fig. 6 Synthetic overview of homogeneity and heterogeneity between all circadian rhythms (Analysis N°2) with details of the clinical population description. a** Rhythm-based categorisation of patients (one patient per column, with the cause of coma on the top), using a combination of dichotomic assessments of circadian rhythms for hormones (yellow box), behaviour (orange box), and EEG (red box). The colour code defining the group of homogeneous presence/absence of CR is the same for the examples given in Supplementary Fig. 1. The complementary analysis of abnormal ultradian/circadian rhythms is presented in Supplementary Fig. 5. Patients' numbers are defined by their group of classification rather by the recording order, whose clinical description is indicated in **b** with the same order and following the same colour-code of classification. NA Non-Available, CR Circadian Rhythms, UR Ultradian Rhythms, CRS-R Coma Recovery Scale – Revised, GCS Glasgow Coma score, GOS Glasgow Outcome Scale (1: Death; 2: UWS/VS; 3: MCS; 4: Severe disability; 5: Moderate disability; 6: Good recovery), UWS Unresponsive Wakefulness Syndrome, VS Vegetative State, MCS Minimally Conscious State, TBI Traumatic Brain Injury, Stroke (H) Intra-parenchymal haematoma, Stroke (SAH) Sub Arachnoid haemorrhage, EEG Electroencephalogramm, N100 Negative response at 100 ms for oddball auditory stimulus, MMN Mismatch Negativity, P300 Positive response at 300 ms for subject own name stimulus. Pupils code: 1 for no mydriasis, 2 for unilateral mydriasis, 3 for bilateral mydriasis.

entropy as a marker of cognitive fluctuations in 4-h EEG recordings[39]. The standard deviation and the coefficients of variation segregated UWS/VS from MCS/CMS better than the mean value on its own. The spectral entropy (estimating the power spectral distribution, which is supposed to be high when consciousness is present) appeared to be steady in UWS/VS but fluctuated with a 70-min period for MCS minus patients (but without analysing sleep-wake cycles at the 24-h scale).

In the present study, while the mean values of EEG features were not discriminant at the group level, *24 h long-term fluctuations* (in particular predictability and rhythm-related variables) were more frequently favourable when they were less complex and therefore more predictable. For example, the *24 h long-term predictability* of Determinism fluctuations was higher when a behavioural cortical function assessed by the CRS-R was present.

Based on Analysis N°1, the most discriminant EEG features describing fluctuations at a short-term scale (accounting for the thalamo-cortical wakefulness effector[40]) were related to spectral analysis, spatial variability, and EEG complexity. In particular, Determinism is a feature assessing the predictability of EEG times series. It is, by definition, inversely related to complexity[41]. The mean value of *short-term* complexity (with every feature) was not discriminant. Conversely, the circadian cyclicity of two features assessing *short-term* complexity (Determinism and Detrended Fluctuation Analysis) was pertinent for Analysis N°2: the existence of circadian cyclicity for both was considered as a reference to define brain function circadian rhythmicity (as they were included in the 4 EEG parameters selected from healthy subjects).

Beyond fluctuation description, we moved forward to analyse ultradian and circadian rhythmicities (Analysis N°1 and N°2). Hormonal and non-hormonal fluctuations have been evaluated as circadian rhythmicities during post-coma using temperature[42,43], hormonal markers[5,44–46], actimetry[47,48], and EEG[49,50] to predict mortality[43], awakening[5,44], and cognitive[48] outcomes. As previously reported[5], some studies have evidenced an association between circadian rhythmicity and behavioural assessment despite inconsistent findings for melatonin/orexin A variations to predict the one-month outcome[46] (UWS/VS vs MCS and Exit-MCS). However, the analysis was only performed at the group level and the sampling rate was low (one per 6 h).

In a previous work based on the longitudinal assessment of two patients from the same cohort[5], it appeared that recovering a circadian rhythmicity for clock-controlled hormones was associated with a cognitive evolution (in the awareness dimension of consciousness) and not with the fluctuation evolution (in the wakefulness dimension of consciousness). The present study further suggests that some EEG parameters are related to the existence of a hormonal circadian rhythmicity (Analysis N°1). It concerned fluctuations in the Beta band (Relative Power and Spatial Variability), classically related to cognitive contents[40]. Moreover, recovering towards an Exit-MCS state required a hormonal rhythmicity as well as an EEG rhythmicity. Although this is surprising as the EEG rather than the hormonal modality was expected to be closely related to cognitive contents, it is consistent with findings previously reported by our group[5].

By considering that the metric of short-term complexity (Determinism) fluctuated with a shorter period, EEG fluctuations in the population of DOC patients appeared as ultradian (i.e. with

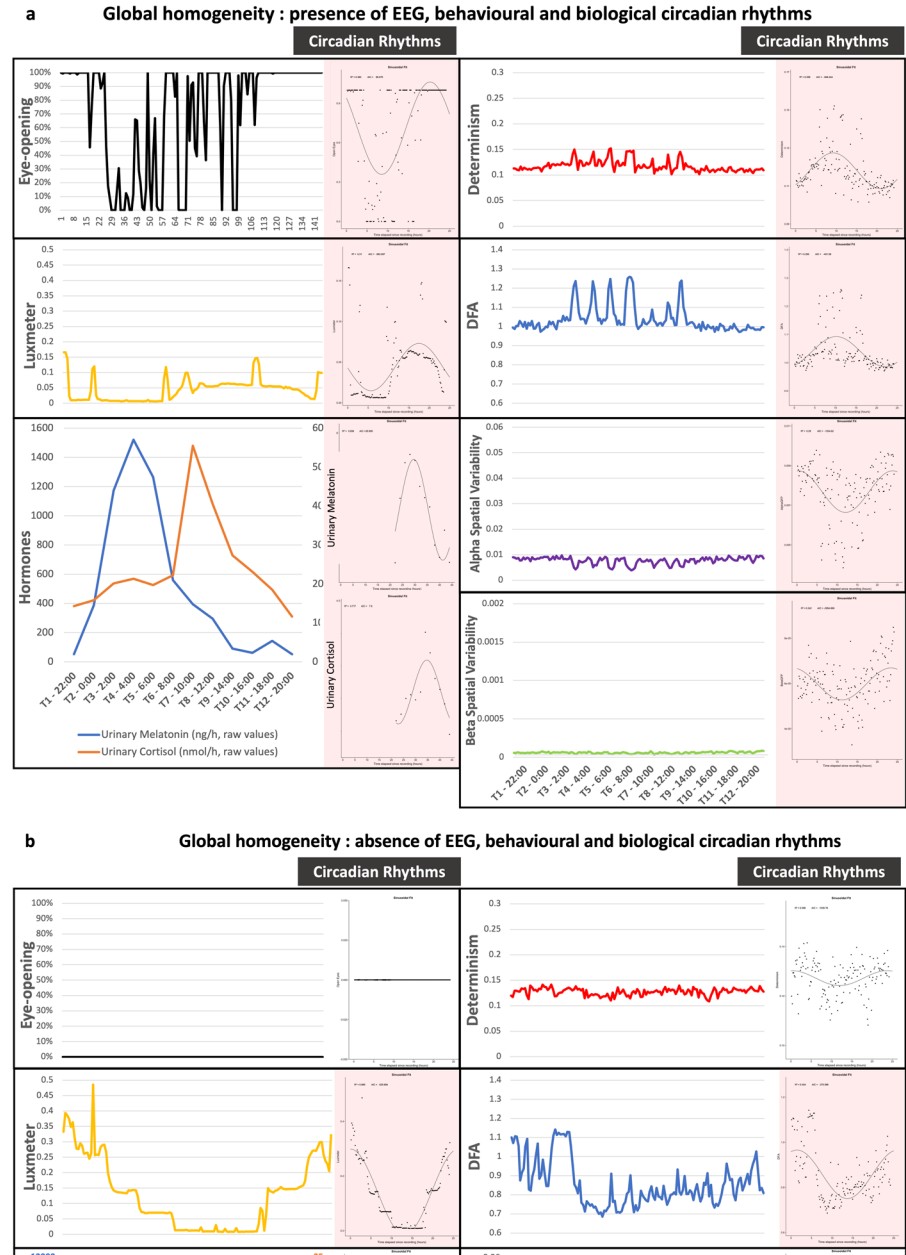

**a** **Global homogeneity : presence of EEG, behavioural and biological circadian rhythms**

**b** **Global homogeneity : absence of EEG, behavioural and biological circadian rhythms**

a period lower than 24 h) rather than circadian, in line with previous studies using spectral entropy oscillations in chronic DOC patients[39,50]. It is tempting to propose that this observation could result from a weakening of the circadian signal from the SCN in DOC patients, which would in turn reveal the ultradian

activities. However, although thought to be absent in healthy human participants due to masking effects[51], ultradian rhythmicities and their reciprocal interactions have been described in a wide range of sleep and endocrine activities, and found to be strong throughout 24 h[52–54]. In DOC patients, the respective

**Fig. 7 Detailed description for opposite patterns of circadian rhythmycities among acute DOC.** Illustration N°1 is provided by Patient N°1 (**a**) and N°10 (**b**) from Fig. 5 (more illustrations are available in the SI). These patients presented a complete opposition of their rhythmic patterns with HOMOGENEOUS PRESENCE OF ALL CIRCADIAN RHYTHMS (**a**) and a homogeneous absence of all circadian rhythms" (**b**) Patient 1 was a 58 y-o male presenting a haemorrhagic stroke. GCS at admission was 6 without pupilar abnormalities. At the date of evaluation, the patient remained UWS/VS (with a CRS-R = 5 and GCS = 9). The neurophysiological battery was favourable, and the patient evolved towards a favourable awakening outcome despite a persisting severe disability. Patient 10 was a 21 y-o male presenting a severe traumatic brain injury. GCS at admission was 6 with a unilateral mydriasis. At the date of evaluation, the patient remained comatose (with a CRS-R = 3 and GCS = 5). The neurophysiological battery was not favourable excepted EEG reactivity, but the patient evolved towards a favourable awakening outcome despite a persisting severe disability. Raw data (continuous curves with uniformed scales across each illustration on the left) and circadian fits (plots with ad hoc scales on the right) are presented for behavioural data (eye-opening moments in black), environmental data (luxmeter in yellow), clock-controlled hormones (blue for melatonin and orange for cortisol with raw values for the continuous curves and log transformed values for the plots), and EEG with a common timescale in abscise (24 h). The 4 EEG features (red: Determinism, blue: Detrended Fluctuation Analysis, violet: Alpha Spatial Variability, green: Beta Spatial Variability) are those presenting a significant circadian rhythm among healthy participants. EEG is defined as circadian when more than one feature presented a circadian fit. Plots with significant circadian fits are indicated by red frames.

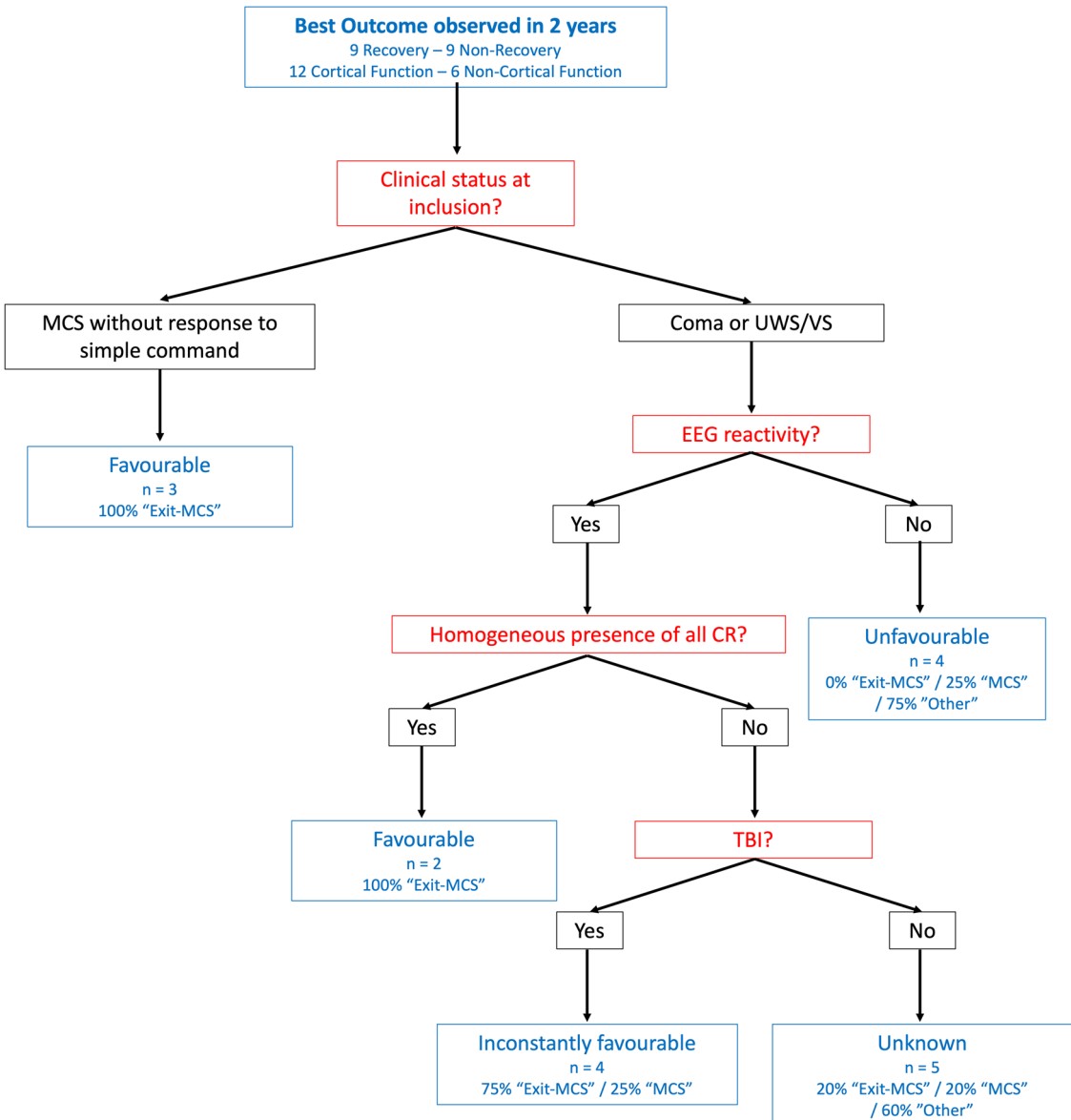

**Fig. 8 Classification tree for outcome prediction in acute patients.** The classification was based on the most discriminant markers (in red) defined in Supplementary Table 3. The order of markers was defined according to the clinical management proceedings (clinical markers then simple EEG then complex multimodal analysis) as long as the statistics allowed it. The outcome observed in each case is in blue. The initial clinical status and the HOMOGENEOUS PRESENCE OF ALL CIRCADIAN RHYTHMS had a 100% PPV for functional recovery (but only their association had a significant Fisher's Exact test, *p*-value = 0.029). EEG reactivity had a 100% NPV (but a non-significant Fisher's Exact test, *p*-value = 0.082). MCS Minimally Conscious State, UWS/VS Unresponsive Wakefulness Syndrome/Vegetative State, CR Circadian Rhythms, TBI Traumatic Brain Injury.

roles of ultradian and circadian rhythms remain unclear because, beyond the particular circumstance of a HOMOGENEOUS PRESENCE OF ALL CIRCADIAN RHYTHMS (considering Analysis N°2) among few acute patients, the unpredictable changes of wakefulness were not modelled by circadian rhythmicities. This observation could explain why wakefulness is usually described as instable and fluctuates clinically within short periods in the DOC population[55].

Analysis N°1 provided some insights into the mechanism of wakefulness instability. Indeed, assessing *24 h long-term neuro-physiological fluctuations* is not solely focused on a direct analysis of the cortical function (as determined by the choice of an EEG technique) but allows an indirect registration of deep structures such as mesencephalon and diencephalon, which modulate the thalamo-cortical circuit[6] (e.g. fast activated EEG associated with wakefulness depending on the ascending reticular activating system[56]).

Favourable outcome was more frequently associated with *24 h long-term fluctuations with a high magnitude and a reduced complexity* (Fig. 4c). This result delineates a sharp up-and-down shape for the patients presenting the more physiological profile. This graphical description of EEG fluctuation shape should not be considered as up and down states since the states of wakefulness herein could not be labelled as an optimum regarded as wake and a nadir regarded as sleep. The shape of the EEG pattern expected from the integrated results (synthesised in Figs. 3 and 4 and illustrated in Fig. 7 and Supplementary Figs 1–4) might be reminiscent of the Flip-Flop balance model regulating sleep-wake cycles, as previously described[6].

Assessing the stability and cyclicity of EEG activity after severe brain injury appears to be an innovative and non-invasive way for investigating the function of strategic brain structures that regulate the balance between different states of wakefulness by the effect of brainstem structures – including the ascending reticular activating system – on thalamo-cortical loops. In a translational perspective, these complementary metrics should be regarded as an additional prognostic biomarker to be validated in independent cohorts and compared to other neurophysiological and neuroimaging techniques to estimate the cost-effectiveness of a 24-h long procedure.

Few theoretical perspectives can be outlined based on our results. First, the complementarity of short and long-term EEG metrics (as defined in Analysis N°1) may illustrate the difference between local and global states of consciousness[57] (see Supplementary Note 1). Second, the implications of Analysis N°2 for the consciousness embodiment hypothesis should be considered (see Supplementary Note 2). Indeed, interrogating further the harmony between EEG cyclicity, clock-controlled hormonal rhythms (directly driven by the hypothalamic circadian timing system), behaviour (such as the mesencephalic output to the oculomotor nucleus commanding eye-opening), and environment (to assess the reactivity to sensory inputs) may provide a comprehensive understanding of consciousness loss and reappearance, in accordance with a "world-body-brain" holistic perspective[58] (see Fig. 6). Third, the theoretical relationship between predictability and rhythmicity were addressed in both Analysis N°1 and Analysis N°2[59] (see Supplementary Note 3).

Several methodological limitations should be discussed.

First, from a statistical perspective, even if the relatively low number of patients included in the present study ($n = 19$ patients in total) could appear as a limitation, it is noteworthy that the repartition of acute patients according to clinical outcome was well-balanced (9 patients in each group). In addition, in accordance with the recent recommendations from the American Statistical Association[60] based on the limitations of p-values and "statistically significant" thresholds, we used statistical approaches based on False Positive Risk and Likelihood Ratios. We reached the same conclusions as with the classic technique of p-value correction for multiple comparisons, which underlines the robustness of the present findings (in particular for the outcome prediction) despite this limited sample size. Importantly, the group-level observation was built in a data-driven analysis including rhythmic and entropic metrics without any a priori selection. Thus, the rhythm-based interpretation of this dataset has emerged without selection by a hypothesis-driven approach. Then, the individual-based prediction was made according to a selection of features and thresholds based on an independent cohort of healthy participants and not on the cohort of DOC patients to limit the over-fitting bias. However, this overfitting bias remained in absence of replication by a validation cohort. Therefore, the interpretation of these results in the clinical context should be taken with caution, with a risk of overinterpreting non-replicated EEG data from Analysis N°1 as biomarkers with potentially harmful consequences for future patients. Therefore, instead, we used the results from the Analysis N°2 in the comparison to current neuro-prognostication tools. Nonetheless, despite the statistical weakness related to the limited sample size and the exploratory nature of the analysis, an indirect argument in favour of the consistency of our results consists in their redundancy through different approaches. Indeed, we emphasise the place of fluctuations: i) for EEG only in a hypothesis-free paradigm (Analysis N°1); ii) for multimodal metrics in a hypothesis-driven approach focused on the circadian system (Analysis N°2).

Second, from a nosological perspective, there was a wide range of brain injuries in the study population (e.g. stroke, traumatic brain injury, diffuse ischaemia). However, the common patterns observed across patients despite this large heterogeneity was suggestive of a common hub of pathogenesis. Of course, the universality of the present findings should be confirmed in further studies including a larger number of patients and presenting successively more homogeneous patterns of injury.

Third, from a "circadian system" perspective, using the term "circadian" for each occurrence of a 24-h rhythmicity should be further discussed as only the analysis of clock-controlled hormones could reliably tell whether the circadian clock was functional. As previously stated, the so-called "circadian" rhythmicities for behavioural and neurophysiological 24-h rhythms were descriptive in the present study. Due to the difficulties to implement a complete constant routine protocol in the clinical setting, using "circadian" for "24 h" did not assume an unequivocal implication of the circadian pacemaker (SCN) in the genesis of rhythms. However, DOC patients presented brain lesions in the mesencephalo-diencephalic structures that are at the same time responsible for wakefulness, reactivity, and motor expression in the one hand, and are the seat of the endogenous circadian pacemaker and its hormonal effectors in the other hand[10]. Consequently, the cause of dissociation between the circadian system and each possible effector are multiple. Assuming that the existence of a hormonal circadian rhythmicity suffices to prove the functionality of the SCN, it does not imply the existence of non-hormonal circadian rhythmicity. Such dissociations (2 occurrences, see the ## symbols in Supplementary Fig. 5) could be due to strategic lesions on key relays of wakefulness expression. Conversely, the absence of a hormonal circadian rhythmicity does not prove that the clock is non-functional because it could be related to a disruption of the pathway throughout the glands secreting melatonin and cortisol[5]. In such instances, direct synaptic inputs from the clock through the brain via the hypothalamic circadian integrator could be better illustrated by other effectors than by a failed hormonal pathway (4 occurrences of non-hormonal 24 h rhythms in

absence of hormonal circadian rhythms within the group "heterogeneity in the presence of circadian rhythms", in Supplementary Fig. 5). However, only a putative (and ethically concerning) protocol avoiding the confounding effect of exogeneous inputs (nursing, sound, and light) would be legitimate to confirm this hypothesis. Altogether, this limitation, which concerns mostly the dissociations between 24 h rhythmicities, does not change our main conclusion consisting in the favourable prognosis associated with the concomitant existence of each 24 h rhythmicity. In particular, the final awakening in this group could be a late argument for the initially preserved function of the circadian clock and for the respect of circadian effectors.

Fourth, from a neurophysiological perspective, our definition of "normal" EEG features could be debated. The inclusion of healthy participants for a direct comparison to patients in a logistic regression analysis could be criticised because both populations were not matched or balanced for EEG recordings (21 recordings for the 19 patients in total versus 3 recordings for healthy participants). However, the distributions in age and sex represent a sufficient variety of normal sleep-wake patterns to define normative boundaries. As the difference between EEG patterns was broad for the three selected parameters (see Supplementary Fig. 6) and delineated a homogeneous cluster of healthy participants, it is unlikely that including more healthy participants would have changed our results. At last, even though our classification was based on our definition of normal patterns, it was not associated with a random population. The final clinical outcome confirmed this non-supervised selection because "healthy-like" patterns predicted the progression toward a "normalised" brain function, defined by the late clinical recovery.

Fifth, the lack of direct comparison with previous similar studies to confirm the extrinsic validity of our results – except for few quantitative fluctuation measurements during short (30 min[20]) and middle (4 h[39]) time windows – is a paradoxical limitation that confirms the originality of our approach and emphasises the need for further studies. However, it is worth noticing that we performed continuous behavioural assessment only for the wakefulness dimension (eye-opening) of consciousness. On the contrary, the definition of the behavioural cortical function was based on a single-point evaluation by a CRS-R at the end of each recording rather than a 5-point assessment over a 10-day moment, as proposed by Wannez et al. for chronic DOC patients (after the beginning of the present study[55]).

Sixth, the present results are based on a protocol conducted under highly-controlled and technically-constrained conditions, which narrows its application in the clinical setting because: i) the space-dimension cannot be easily summarised (all EEG channels must be used to get the relevant Spatial Variability metrics); ii) the time-dimension implies a minimal recording window of 24-h (a 20-min EEG recording can catch neither a sufficient amount of fluctuations nor a rhythmicity with a 24-h period); iii) the prognostic value based on the HOMOGENEOUS PRESENCE OF ALL CIRCADIAN RHYTHMS implies to respect the multimodality of the recording; iv) the failure of defining concomitant hypnograms prevents from an easy extrapolation based on this universal ordinal classification of wakefulness.

Concerning the clinical interpretation, it is of note that the results from analysis N°1 were not proposed as a biomarker for prognosis in absence of a sufficient sample to allow a validation cohort. Taking the prognostic results from Analysis N°2 altogether, it appeared that clinical data were not major contributors to the outcome prediction. These findings however, reinforced the theoretical framework highlighting the dependence between the neural basis of consciousness and the internal-external world relationship[58,61]. The HOMOGENEOUS PRESENCE OF ALL CIRCADIAN RHYTHMS was a marker of favourable outcome that outperformed

the classic EPs and ERPs markers. Despite a low sensitivity, it improved the overall prediction in a classification-tree based on dichotomic factors. Moreover, associating quantitative EEG markers from the 24-h recordings outperformed the associations with clinical markers (at the date of admission or at the date of recording) but could be synergic with the initial Glasgow Coma Scale [GCS] or with the cortisol fit. Notably, the outcome prediction herein was more clinically relevant than that of previous studies in the field as we decided to include Exit-MCS but not MCS/CMS as a favourable outcome. In other words, the threshold was the "communicative ability" or "functional object use"[62] rather than "any clue of cortical function"[3], which could be of ethical concern when considering the MCS minus category as favourable. Although early assessment of MCS/CMS criteria may have a predictive value for a late favourable outcome[63], the probability to reach a good functional status for later MCS/CMS observation (at a 2-year follow-up) appeared as low in a previous report[64].

Classic prognostic markers have usually not been able to predict outcome beyond MCS/CMS. Cognitive ERPs after musical stimulations predicted any improvement on the CRS-R[37] while the existence of the new CMS sign "habituation of the Auditory Startle Reflex" was able to predict improvement within and beyond the group of MCS patients defined by the standard CRS-R[65]. The use of MMN in anoxic coma was able to anticipate with a 100% PPV the evolution to the MCS/CMS or higher for Coma or UWS/VS patients[38]. The same performance has been observed for the local-global MMN paradigm at the population level[66] despite the first report of this technique used to predict a good recovery beyond a "minimal awareness" but only for patients already in a MCS/CMS at the date of recording[67].

As for previous biomarkers used for neuroprognostication purposes (e.g. novelty P3: Se = 0.71, PPV = 0.81 ; P3b: Se = 0.46, PPV = 0.92 for Fischer et al.[68]), the existence of such a predictive pattern (see Supplementary Table 3) with low sensitivity/NPV but high specificity/VPP for a favourable outcome implies a high risk of false negative results that should be interpreted carefully. The clinical management should not be modified if one fails to demonstrate a HOMOGENEOUS PRESENCE OF ALL CIRCADIAN RHYTHMS, as is it currently unclear whether some patients are not on the verge of further recovery. In the same vein, the historical interpretation of MMN absence was described as a possibly transient phenomenon. Therefore, only positive results should be interpreted as it they have been shown to precede the return of behavioural signs of consciousness within 48 h[69].

At last, we proposed another narrative for the natural history of coma recovery: while consciousness has previously been related to a high complexity of the cortically mediated brain activity (for short-term information integration), this study proposes an original association with a high predictability of brain fluctuations (for 24 h-long-term generation of rhythms). Although they were not cortically mediated, these biomarkers were related to the later recovery of awareness in this exploratory study in which predictions were made on a descriptive level and should be confirmed in additional validation cohorts.

The present study outlines a theoretical chronology of the favourable recovery from coma based on a 24 h long-term rhythm analysis. Despite the absence of an individual description of the natural history of each coma recovery, it could be used to build an innovative methodology to assess the interactions between the content of consciousness and wakefulness assessments. At first, the rhythmic pattern would consist in an overall chaos for most patients. Then, if the brainstem metronome of frail wakefulness modulations rises again, it could be assessed by the metrics of fluctuation amplitude and predictability or, in rare instances, by a transient ultradian rhythm (in the absence of circadian

repression). Later, the reappearance of a circadian rhythm for one or some modalities, then for all of them, and at last, its harmonisation with the environment would be the final step of recovery.

In the initial steps of this process, some specific fluctuating profiles could predict the later steps of recovery. Such profiles suggest that a resilient sleep-wake promoting system remains able to change the "global states of consciousness" (assessed by wakefulness) which, in turn, could modulate the "local states of consciousness" (expressed by consciousness contents or awareness).

## Methods

This observational study was managed in a tertiary neurological intensive care unit (Lyon teaching hospital). Inclusion and exclusion criteria (see below) allowed the diagnosis of Coma, Unresponsive Wakefulness Syndrome[UWS][18] (previously named Vegetative State[VS][17]), and Minimally Conscious State [MCS][70] (or Cortically Mediated State [CMS][3]) without response to simple command, corresponding to the sub-group MCS minus[71], according to the French version of the Coma Recovery Scale-Revised [CRS-R][62]. In the present study, these clinical states were coined as Coma, UWS/VS, and MCS/CMS, respectively.

The primary endpoint was the best functional outcome that could be observed during a 2-year follow-up (recruitment period: January 2014 – October 2017, last follow-up in October 2019). This outcome was described using the GOS assessed at 3, 6, 12, and 24 months and, if unconscious, by a clinical assessment of the CRS-R.

**Experimental design: inclusion and exclusion criteria**. The *inclusion criteria* were: being aged between 18 and 80 years old; suffering from a disorder of consciousness [DOC] after traumatic brain injury, infarct, intracerebral haemorrhage, sub-arachnoid haemorrhage, or post-anoxic encephalopathy; not responding to simple command more than 2 days after sedation withdrawal; requiring a persistent intubation for mechanical ventilation or a tracheotomy due to a wakefulness level incompatible with physiological airway management; being in a stable clinical state; having relatives signing an informed consent.

The *exclusion criteria* were: displaying neurophysiological signs of bilateral deafness on the Peak 1 and 2 of the Early Auditory Evoked Potentials (EAEPs) as patients were concomitantly included in an oddball paradigm associated with a musical stimulation[37]; suffering from severe non-controlled neurovegetative crisis; suffering from non-controlled epilepsy; being pregnant; being in a coma of unknown aetiology or without defined evolution (such as auto-immune encephalitis).

Clinical evaluations of the outcome were performed by FG. For 4 patients inaccessible to a clinical examination after a long-range transfer, a systematic interview of the physician in charge (for institutionalised unconscious patients) or relatives (for conscious patients at home) was performed.

The experiment was conducted in agreement with the guidelines of the Declaration of Helsinki and approved by the local ethics committee (CPP Sud-Est II, NCT02742506), and required a written informed consent from the patients' relatives.

**Experimental design: population description at inclusion**. 22 participants were analysed in total: 3 healthy participants and 19 patients, of whom 18 acute patients (5 women) were included in the main prognostic study.

The final diagnosis retained was Coma for 3 patients (absence of spontaneous eye-opening during the CRS-R assessment but possible eye-opening after stimulation), UWS/VS for 12 patients,

and MCS/CMS for 3 patients (without response to simple command). The CRS-R was performed only once, at the end of the recording, to avoid modifying the spontaneous behaviour related to naturalistic ICU stimulation (and its neurophysiological correlate).

The aetiology was traumatic brain injuries for 7 patients, strokes for 8 patients, severe hypoxia for 1 patient, and anoxia after cardiac arrest for 3 patients.

Three recordings were added for patients at the chronic stage. It included: i) 2 longitudinal cases of acute patients previously recorded who were both alive and not recovered at 6 months: ii) one original patient classified as UWS/VS who was recorded 6.5 years after the injury. For comparison, 3 healthy participants (2 women) were also included with a mean (±SD) age of 46 years (±19.3).

The individual clinical features are detailed in Fig. 6b. Descriptive statistics are described for relevant features in Supplementary Table 1 using the mean (±SD), median 25th and 75th percentiles along with the range (minimum and maximum values).

The imaging features are detailed in Supplementary Table 2. MRI was available for all patients excepted one post-anoxic patient with a contra-indication to MRI. If more than one MRI was provided, the one with the shortest delay with the multimodal rhythmic assessment was used. They illustrate the respective prevalence of cortical and non-cortical lesions including hypothalamic lesions assessed by MRI (T2SE or FLAIR sequences to assess primary lesions and secondary oedema; T2* and Susceptibility-Weighted-Imaging to assess more precisely diffuse axonal lesions). The delay between the MRI and the multimodal rhythmic assessments ranged between 0 and 42 days (mean = 9.4 days). The lesions were described qualitatively (presence or absence) in selected cortical and sub-cortical areas (see Supplementary Table 2 A, B) then regrouped at the system-level according to the following considerations: diffuse telencephalic lesions by combining cortical and hemispheric white matter areas; diffuse lesions in the mesocircuit by combining basal ganglia and thalamus; diffuse lesions involving the ascending reticular activating system in the brainstem excluding the cerebellum; lesions in the corpus callosum alone as it may imply a diffuse impairment of anatomical connectivity. The hypothalamus was scrutinised per se due to its implication in the circadian disruption patterns described herein. For the semi-qualitative description of lesions provided in the Supplementary Data 1: i) the ventricular system items were excluded because of their questionable causality in consciousness disruption; ii) lesions were considered for each system as predominant on the right/left hemispheres or as bilateral if the sum score difference was ≤2 for cortical/white matters areas, ≤1 for mesocircuit and brainstem, <1 for hypothalamus and cerebellum; iii) the percentage of right/left/bilateral/median (for corpus callosum) occurrences of lesions were described at the population-level and at the group-level.

**Experimental design: population outcome**. The condition of 9 patients evolved favourably (i.e. at least Exit-MCS): 6 patients recovered with a severe disability, 2 patients with a moderate disability, and 1 patient had a good recovery. The other 9 patients had an unfavourable outcome, among them 4/9 patients died before the first follow-up at 3 months, and the best assessment was MCS/CMS for 3/9 or UWS/VS for 2/9. A longitudinal recording was also performed for 2 of these patients more than 6 months after a traumatic brain injury[5] (outcome UWS/VS for 1 patient after 6 months and MCS/CMS for the other patient after 7 months). Their late outcome was MCS/CMS for both, but they

did not present any reliable sign of communication during the 2-year follow-up.

**Acquisition: neurophysiological recordings**. The following multimodal system was recorded continuously for 24 h. The recording was shorter for one patient because of a medical indication for a thoracic CT-scan implying the disconnection of the patient from the EEG after a 23-h long recording. Other recordings were 24-h to 25-h long. In healthy participants, only the polysomnography was recorded for 24 h (no in-place bladder probe for urine, behaviour from a wake-sleep diary).

The polysomnographic recording was performed using a SystemPlus amplifier (Micromed®, Italy). Thirteen EEG electrodes were placed on the scalp according to the 10–20 system (F3, Fz, F4, C3, Cz, C4, P3, Pz, P4, T3, T4, O1, O2), the sampling rate was 256 Hz (16 bits), the analogic filtering was 0.16 to 60 Hz. There was also a common reference on the nose and a ground electrode on the forehead. The polygraph montage consisted in 5 bipolar electrodes: on the submental area for EMG, on the precordial area for ECG, around the orbits for vertical and horizontal EOG, on the epigastric area for the respiratory piezoelectric device.

**Acquisition: behavioural recordings**. A dual video acquisition (sampling rate = 1 Hz) was provided using a standard camera (Hercules®) and an infra-red camera (Carl Zeiss Tessar, HD 1080p, Logitech®), combined with an infra-red light (850 nm, EcoLine®) placed on the wall and oriented towards the patient's face. Both cameras were placed on a metallic arm whose position and distance (between 30 and 50 cm) in front of the patient's face was continuously adapted by the caregivers after the nursing moments or the spontaneous movements. The complete device was regularly checked by FG at least once every 2 h.

**Acquisition: environmental recordings**. Environmental light and sound intensities were recorded using a dedicated luxmeter (CHY630, IDDM + ®) and sonometer (SoundTest-Master, configured in dBA, FAST: 125 ms, setup level between 30 and 130 dB, Laserliner®). Both devices were placed near the head to be as close as possible to the patients' eyes (pointing to the same direction) and ears.

The position of each patient in the unit, and therefore their orientation relative to the sun, was not specifically recorded. The orientations of the rooms were as follows: 2 to the east, 5 to the west, 7 to the south, and 6 to the north. Every room received sunlight as their windows are open to the outdoors or to a large courtyard.

All the previous recordings were synchronised using a homemade system running under Labview (National Instrument) and using a common clock both for video and signal acquisitions. A trigger was sent both on the acquisition board (NI USB-6008 card, used for sound and light acquisition) and the external EEG system. The trigger was generated every 2 min to correct a potential temporal drift at the end of the 24-h period. The homemade software Volcan was used for the synchronised acquisition and for the analysis of behavioural (video) data and the extraction of environmental data. As the faces of the nursing staff were not filmed (the cameras focused on the patients' faces and, incidentally, on the staff's hands), informed consent from the staff was waived.

**Acquisition: hormonal recordings**. The levels of Urinary Free Cortisol [UFC], CMIA technology, Abbott Diagnostics, Illinois, USA) and urinary 6 sulfatoxymelatonin (aMT6S, the main hepatic melatonin metabolite, assessed by Radio-immuno

Assay[72]) were also measured on urinary samples taken every 2 h during the 24-h recording (sampling rate = 5.55 $10^{-4}$ Hz). To that end, the urine bag was emptied before each 2-h sampling, the total urinary output for this 2-h period was recorded, and a fraction of this miction was collected, numbered, and kept at 4 °C until the 12th sampling. Then, the 12 samples were sent at once to the hospital hormone laboratory.

**Analysis: EEG preprocessing**. The recordings were first synchronised on the common triggers sent by the Labview system using a python script. A pre-processing step of artefact rejection was then conducted by FG using an ad hoc python software with a direct visualisation of the raw EEG signal on 30-s pages with low- and high-pass filters (data viewer). First, the optimal high- and low-pass filters were selected and uniformly applied for every patient, to avoid excluding prolonged EEG low rhythm artefacts (due to slow electric wire movement and sweating) and rapid artefacts (increased muscular tonus and electric noise at 50 Hz). The analysis was finally restricted to the 1–30 Hz band.

Second, an automatic exclusion of extreme values (assessed as the 5 first/last percentiles in the distribution of each 30-s moments) was performed to obtain a complementary correction of artefacts limited in time (<1 s) and space (one derivation).

Then, a manual exclusion was performed. Artefacts were uniformly defined after the inclusion of all patients and during a unique moment of analysis as: i) artefact duration >1 s, ii) visible only using the bipolar montage, iii) concerning more than one bipolar derivation (F3-C3, C3-P3, P3-O1, Fz-Cz, Cz-Pz, F4-C4, C4-P4, P4-O2, T3-C3, C3-Cz, Cz-C4, C4-T4), iv) the interpretability of the median line derivation (Fz-Cz) was checked to be used as a reliable and continuous marker during the complete 24-h recording.

**Analysis: short-term EEG description**. Four types of continuous features were then extracted from the raw EEG signal to describe it at a short-term temporal scale. In total, they constituted twenty-one short-term linear or non-linear EEG features which were calculated on 30-s windows with 15-s overlaps (on Fz-Cz for the 3 first categories as mid-line derivations seemed usually less impaired by muscle artefacts[49] and on 13-channel EEG for the last category):

- Power was described using the Total Power (1–30 Hz) and the Variance of the EEG signal.
- Spectrum was expressed by the Absolute Power in each band (Delta 1-4-Hz, Theta 4–8 Hz, Alpha 8–12 Hz, Beta 12–30 Hz), the Relative Power (Absolute Power in each band divided by the Total Power, normalised between 0 and 1), the Alpha Dominant Frequency (dominant frequency in the Alpha-band), and an AutoRegressive model[73] of order 4 [AR4]. The latter integrated spectral analysis and was used as a frequential smoothing (less sensitive to small spectral changes as the order is low) based on a temporal prediction (assuming a short-term stationarity hypothesis); its magnitude and frequency peak were expressed in dB and Hz, respectively, and were summarised by the AR4 Total features (=magnitude + frequency peak).
- Complexity was described using python scripts (from the PyEEG library): the Determinism[28,41] (assessing a predictability value as the probability for the occurrence of recurrent patterns), the Singular Value Decomposition [SVD]-entropy (a temporal entropy metric usually related to cortical information processing[29] and calculated in this case after a singular value decomposition[74]), and the Detrended Fluctuation Analysis [DFA], defined as the geometrical precision required to summarise the raw EEG[75]

which was previously validated to catch wakefulness changes from EEG markers but was also recently proposed to catch long-term temporal correlation during DOC[30].

- Spatial Variability was evaluated using the standard deviation of the Relative Power ratios in the Delta/Theta/Alpha/Beta-bands throughout all the bipolar montage by analogy to the Global Field Power metric (computed for EEG time series instead of evoked potentials[76]).

**Analysis: 24 h long-term EEG description**. In a second step, the time dimension was summarised by calculating 8 different *variables of 24 h long-term fluctuations* on the 21 features of short-term signal description, following a 10-min smoothing (to limit missing data because of artefact EEG moments). As a convention, every reference to *24 h long-term variables* describing fluctuations at the 24-h scale is written in italic in figures and texts and are placed before the EEG features as follows: *long-term variables* for EEG features (e.g. *Permutation Entropy* for Delta Absolute Power). These *variables* were defined as follows:

- *24-h distribution* was explored by the mean values and fluctuation magnitude assessments, namely the *Standard Deviation* (representing the overall level and the fluctuation magnitude of each feature, respectively) and the *Coefficients of Variation* (Std/Mean*100, in %, representing a relative fluctuation magnitude).
- *24 h long-term predictability* was assessed by fluctuation complexity expressed by information theory metrics and estimated by *SVD-entropy* and *Permutation Entropy* (PyEEG library).
- *Rhythmicity* was explored by calculating the *Dominant Period*, the *Correlation Coefficient of the Circadian Period* (without assuming the existence of an unequivocal SCN drive of this rhythmicity) and the *Correlation Coefficient of the Dominant Period* (which might be circadian if the 24-h period was at the highest or ultradian if the period was <24 h).

The *Dominant Period* of ultradian fluctuations was calculated using R (RCore Team, 2015, R: A language and environment for statistical computing, R Foundation for Statistical Computing, Vienna, Austria, *spectrum function*) and was comprised between 20 min and 24 h (according to the Nyquist-Shannon theorem as the sampling was 10 min after smoothing, which is compatible with tracking classic ultradian rhythms in DOC[39]). The 10-min smoothing allowed enough values to manage statistical analysis of distribution and correlations between pairs of features but missed the fluctuations below 10 min. However, this threshold remained compatible with the analysis of usual ultradian oscillations (according to the theoretical framework of the Basic Rest-Activity Cycle or BRAC[77] initially described around 40 min) in particular around 70 min as recently demonstrated for DOC[39]. In case of missing values despite smoothing, a linear interpolation was performed before running the *spectrum function*.

Ultradian and circadian sinusoidal fits were assessed for every time series using a least-square single-harmonic regression model (Rcore Team, 2015, "ggplot2, gridExtra, cowplot, mosaic, readxl, nlstools, matrixStats, questionr" packages). The so-called circadian period was a 24-h rhythmicity constrained between 23h30 and 24h30[78]. Rhythmicity variables were calculated for EEG and eye-opening periods (circadian and ultradian) and for hormonal results (circadian only because of the 2-h sampling rate preventing the assessment of ultradian rhythms for periods below 4 h) and dichotomised as presence or absence of these rhythms using ad hoc threshold definitions for the coefficient of correlation of fits which were both sufficiently high and validated a visual fit. Notably,

the existence of this rhythmicity was descriptive and did not assume an unequivocal implication of the circadian pacemaker (SCN) in its genesis. The best ultradian fit was constrained around the *Dominant Period* given by the *spectrum function* (+/−5%), provided that the *Dominant Period* was not 24 h (theoretical maximum = 12 h; theoretical minimum = 20 min). If the *Dominant Period* was 24 h, it was reported as such for this variable but without ultradian rhythmicity. Regression coefficients ($R2$) were reported as continuous parameters if they reached a minimal threshold ($R2 > 0.1$).

At the end of the complete process, the combination "features*variables" allowed the exploration of 168 EEG parameters (Fig. 1) used to build a data-driven analysis (see below).

**Analysis: EEG rhythm description**. From the previous analyses, we built another categorisation of patients according to the presence/absence of circadian (24-h) rhythms and ultradian rhythms in EEG. In healthy participants, circadian rhythms were found for short-term complexity (Determinism, DFA) and the Alpha and Beta Spatial Variability (mean $R2 > 0.2$, minimum $R2 > 0.1$). In contrast, no normal ultradian-like EEG pattern was observed. The patients were categorised as presenting a normal circadian EEG pattern if more than one of these 4 EEG features had a high circadian fit ($R2 > 0.2$). An abnormal circadian EEG pattern was defined if higher circadian fits were observed for any other EEG feature not found in healthy participants ($R2 > 0.3$). An abnormal ultradian EEG pattern was retained if higher ultradian fits were observed for any EEG feature ($R2 > 0.3$).

**Analysis: behavioural analyses using dual video recordings**. Standard and infra-red video recordings were visualised twice to manually score the general motricity (rest and agitation for endogenous movements and nursing for exogeneous movements) and the eye-opening moments (full eye-opening, slight eye-opening, then regrouped as eye-opening in opposition to eye-closing moments) using the ad hoc scoring system built-in from the Volcan homemade software. Certain time windows were scored as non-available when the patient was out of the field and/or when the patient was completely turned over for nursing.

**Analysis: comparison between EEG and behavioural metrics**. To limit missing data (because of non-available moments) and to allow comparison with EEG data, a 10-min smoothing was also applied to behavioural data.

The *Dominant Period*, the *Correlation Coefficient of the Circadian Period* (24 h), and the *Correlation Coefficient of the Dominant Period* were calculated for behavioural assessments, using the same procedure as for EEG features. Patients were categorised as having recovered a behavioural circadian (24 h) rhythm when the $R2$ for the fit was >0.1 (R script) and if it was confirmed by a significant *p*-value (<0.05) using a complementary assessment of the model fitting (SigmaPlot 12, Systat Software Inc., San Jose, CA, USA) as reported previously[5]. In healthy participants, a strong circadian (24 h) fit was identified ($R2 > 0.6$). The process allowed to create another classification: with or without behavioural circadian rhythm. An abnormal ultradian behavioural pattern was retained if high ultradian fits were observed for eye-opening moments ($R2 > 0.2$).

**Analysis: environmental assessments**. Despite the circadian implication of this study, it was not possible for ethical reasons to hold patients in a dim-light protocol. Comatose patients presented several features of a constant routine protocol (the gold standard to experimentally unmask circadian rhythmicity in healthy participants[79]) with enteral continuous feeding and continuous position lying in a bed. However, the absence of

constant light levels in our study would not qualify our protocol as a gold standard constant routine but rather as a near-constant routine protocol[5] that could be regarded as close to free running activity. Healthy participants were not maintained in a constant routine either.

Raw sound and light values were extracted using a Python script without artefact rejection and resampled at 10 Hz. Results were provided in Volt and then converted in Lux (1 V = 1000 Lux) and dB (1 V = 100 dB).

**Analysis: urinary measures**. A bladder probe and a per-hour urine quantification were used. Results were given in ng/h for aMT6S and nmol/h for UFC. These measurements were available for most acute and post-acute DOC patients except for 2 due to logistical issues at the beginning of the study. aMT6S assays were run in duplicate samples by batch. Samples from the same assessment (one cycle per patient) were run all together in a batch, but samples from different assessments (for each patient) were run in different batches. Assays of UFC were run for single samples on a i2000 instrument (Abbott diagnostics). Urinary instead of plasmatic markers have been previously used as appropriate markers of circadian rhythms to diagnose Circadian Rhythm Sleep Wake Disorders[80], to study the circadian rhythm of blind subjects[81], and to analyse UFC and aMT6S oscillations in elderly subjects with a 90-min sampling[82], which was close to the present 2-h sampling.

The 2-h sampling rate prevented us from assessing ultradian rhythms for periods below 4 h (according to the Nyquist-Shannon theorem and more conservatively <8 h) which were of the highest importance for comparison with continuous (behavioural and neurophysiological) recordings. Therefore, only the *Correlation Coefficient of the Circadian Period* (24 h) was provided by using the same procedure as for EEG features and behavioural data. Patients were categorised as having recovered a hormonal circadian rhythm when the R2 for at least one hormonal fit (melatonin or cortisol) was >0.1 (R script) and if it was confirmed by a significant *p*-value (<0.05) using a complementary assessment of the model fitting (SigmaPlot 12, Systat Software Inc., San Jose, CA, USA) as reported previously[5]. The process allowed to create another patient categorisation: with or without hormonal circadian rhythm.

**Statistics and reproducibility: analysis N°1 and analysis N°2**. Concerning the sample size, as the study was exploratory and descriptive, no power analysis could be performed a priori as the amplitude of the prognostic effect could not be calculated due to the absence of equivalent study in the literature. The potential outliers were not handled in a specific way and were described as the other cases. Unless specified otherwise, variables are expressed as mean (±SD, standard deviation).

Concerning reproducibility, due to the exploratory nature of the study and the difficulties to tackle synchronously each recording in an ICU environment for rare patients, no replicates were performed.

Concerning the supervised data-driven analysis for comparing EEG data with behavioural and hormonal assessments (Analysis N°1), as the number of EEG parameters was high (168), a supervised data-driven analysis was conducted to find the most relevant *24 h long-term variables* descriptive of EEG features by using clinical, behavioural, and biological factors.

The choice of metrics was based on their mathematical complementary to describe several EEG dimensions of fluctuations in the short and long-term perspectives without a priori. It should be noted that this strategy was appropriate to the exploratory nature of this work as the aim was not to confirm the superiority of a candidate biomarker. It was rather to define

which group of parameters – assessing either mean values of EEG feature or the description of these features' fluctuations over 24 h – contained more information associated with a close-to-physiology brain functioning after injury.

As a convention, every reference to factors is written as follows: "Factors" (e.g. "Behavioural Cortical Function"). Two dichotomic factors were selected to summarise the clinical data. The 168 EEG parameters were compared to the presence of clinical "Behavioural Cortical Function" at the assessment (as defined by cortically mediated behaviour CMS 3b[3] without presuming an indirect assessment of electrical cortical function as in CMS 3a). Based on the CRS-R evaluation, it led to compare MCS/CMS[3] patients with the composite population including Coma and UWS/VS. The EEG data were also compared to the final "Functional Outcome" among acute DOC patients, defined as Favourable in case of a full recovery of consciousness (i.e. Exit-MCS patients, including some cases with severe disability defined by the GOS) vs Unfavourable (i.e. non-recovery of consciousness including Coma, UWS/VS, and MCS/CMS).

In addition, two factors were selected to summarise behavioural and hormonal data. The 168 EEG parameters were compared to the existence of a "Behavioural Circadian Rhythmicity" in DOC patients using the eye-opening moments as a clue of wakefulness modulations (i.e. Presence vs Absence of eye-opening/closing moments with a 24-h period). These parameters were also compared to the existence of a "Hormonal Circadian Rhythmicity" in DOC patients using the urinary assessments of clock-controlled hormones, melatonin and cortisol (i.e. Presence vs Absence of a 24-h rhythmicity for at least one clock-controlled hormone).

The categorisation of patients according to these factors was performed by including the 168 EEG parameters using logistic regression models (Fig. 1). The models were performed using binomial logistic regression for univariate analysis as this classical linear analysis has been proposed to be non-inferior to more complex machine learning algorithms for outcome prediction in dichotomic classifications[83].

A multivariate binomial logistic regression was used to define the parameters independently related to each factor (Rcore Team, 2015, package Rcmdr, *glm function*) by building the best model using parameters with a *p*-value < 0.05 in univariate analysis (after backward selection method based on the minimisation of the Akaike Information Criterion [AIC] metric, which estimated the validity of the model). The parameters finally selected were presented in 2-dimensional boxplots (centre line: median; box limits: upper (75th) and lower (25th) quartiles; whiskers: 1.5x interquartile range above 75th quartile and below 25th quartile; points: outliers) and 3-dimentional graphs in R (Rcore Team, 2015, *ggplot2* and *scatter3D packages*, respectively).

The statistical significance of between-group differences was tested by a Multivariate Analysis Of VAriance [MANOVA] (Rcore Team, 2015, Rcmdr package, *manova function*) with a corrected threshold for 5 factors using the Bonferroni method (corrected *p*-values < 0.01). In addition, the False Positive Risk [FPR] and Likelihood ratio [LR][60] were assessed *post hoc* to evaluate the statistical results at the model level using the False Positive Risk Web Calculator (version 1.7 Longstaff,C. and Colquhoun D, http://fpr-calc.ucl.ac.uk/ Last accessed 2021-01-07, with a conservative setting: *p*-equals case method, prior probability = 0.5, effect size = 1)[60].

Concerning the multimodal rhythmic analysis (Analysis N°2), as we assumed that the description of rhythmicity had a major relevance, DOC patients were categorised according to the presence of circadian and/or ultradian rhythms in several modalities, in particular for some parameters found to be rhythmic in healthy participants (see Pre-processing, EEG analyses). The EEG features presenting a circadian rhythmicity among healthy participants were

Determinism, Detrended Fluctuation Analysis, Alpha Spatial Variability, and Beta Spatial Variability.

According to the combination of three 24-h rhythms for EEG, behaviour, and hormones, 3 groups of patients were defined. In case of homogeneity among circadian rhythms (for EEG, behaviour, and hormones), the HOMOGENEOUS PRESENCE OF ALL CIRCADIAN RHYTHMS PATTERN was defined in opposition to the patterns of homogeneous absence of circadian rhythms and heterogeneity in the presence of circadian rhythms. As a convention, every reference to this combination of rhythms is written as follows: HOMOGENEOUS PRESENCE OF ALL CIRCADIAN RHYTHMS. Then, the abnormal circadian and ultradian dynamics (for EEG features and behaviour, if R2 > 0.3 and higher than the maximal value of healthy participants) were described qualitatively as possible adaptive patterns related to the absence of circadian rhythms or as additional rhythms added on a normal circadian rhythm.

Among the patients presenting a hormonal circadian rhythmicity (standing for the closest indicator of the circadian system function), the phase relationship with behavioural and cerebral circadian (24 h) rhythmicity was described qualitatively as the difference between the acrophases of each modality (i.e. the peak of the circadian fluctuation, according to the sinusoidal estimation). The EEG and behavioural cycles were considered "in phase" with the melatonin/cortisol peak (depending on which fit was significant) if at least one EEG acrophase (among the 4 features considered as normally circadian) or the eye-closing acrophase had a phase angle of less than 3 h with the hormonal peak.

**Statistics: comparison with the clinical prognostic markers**. Considering qualitative dichotomic parameters (Analysis N°2), the specific prognostic value provided by the HOMOGENEOUS PRESENCE OF ALL CIRCADIAN RHYTHMS pattern as the most integrated result from the 24-h multimodal recordings was tested. This result was compared to the previously described dichotomic neurophysiological markers, such as *EEG reactivity*[84,85], *bilateral abolition of cortical components* (for Somatosensory Evoked Potentials[84] or Middle Latency Auditory Evoked Potentials[86]), *morphological alteration* (Somatosensory Evoked Potentials N20 amplitude or Middle Latency Auditory Evoked Potentials Na-Pa amplitude or Brainstem Auditory Evoked Potentials Peak V amplitude[87]), *presence of N100*[87,88], *presence of MMN*[87], *presence of P300*[68]. The significance of the association with the outcome was tested using the Fisher's Exact test (as the gold standard for small samples, after a selection with Chi2 values). Sensitivity, specificity, positive and negative predictive values were calculated using contingency tables. The FPR and LR were provided for the markers with a *p*-value < 0.05.

Considering quantitative parameters (Analysis N°1), the variables selected from the 24-h recordings (EEG features correlated to outcome, behavioural, and hormonal circadian fit values) were compared to quantitative clinical markers (initial GCS and pupilar abnormalities, GCS and CRS-R at the date of recording) by MANOVA (after Bonferroni correction for each class of comparison, i.e. within clinical or non-clinical markers and clinical vs non-clinical markers) and illustrated using 2D scatter plots in R (*ggplot2 package*).

**Reporting summary**. Further information on research design is available in the Nature Portfolio Reporting Summary linked to this article.

## Data availability

The source (EEG, polygraphy, video, biology, clinical data) and intermediate data (processed EEG, synthetic table) that support the findings of this study are stored in the Lyon Neuroscience Research Centre server. They are available from the corresponding author upon reasonable request (mail: florent.gobert01@chu-lyon.fr), excepted the source data of video for behavioural analysis to protect study participant privacy. The source data for all the figures and plots in the manuscript are available in the Supplementary Information (Supplementary File 1 and 2). The original codes that support these findings are available from the corresponding author upon reasonable request and at https://github.com/Lx37/data_viewver for the ad hoc data viewer allowing manual artifact rejection.

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

## Acknowledgement

The authors are indebted to Maude Beaudoin-Gobert for revising thoroughly the manuscript and for her continuous perspicacity to design appropriately each illustration, to Hélène Boyer and Véréna Landel for their help in manuscript preparation, to Lionel Naccache for his advice in manuscript redaction, and to the complete medical and nursing team from the Lyon neurological hospital ICU for their time and support at every step of data acquisition and manuscript preparation. The study was partially funded by the "Gueules Cassées" foundation (20 rue d'Aguesseau, 75008 Paris).

## Author contributions

F.G.: Conceptualisation, Methodology, Formal analysis, Investigation, Writing - Original Draft, Visualisation. A.C.: Methodology, Software, Resources, Data Curation, Writing -Review & Editing. H.B.: Methodology, Writing - Review & Editing C.B.: Methodology, Software, Writing - Review & Editing. M.T.: Software, Resources J.A.: Software V.R.: Methodology, Investigation, Resources. F.D.: Investigation C.G.: Investigation C.G.: Methodology, Formal analysis, Resources, Writing - Review & Editing J.L.: Conceptualisation, Writing - Review & Editing, Supervision. F.P.: Conceptualisation, Writing - Review & Editing, Supervision.

## Competing interests

The authors report no competing interests.
