## [Peer Review File · Communications Biology]

Reviewers' comments:

Reviewer #1 (Remarks to the Author):

The authors provide the analysis of a thoroughly acquired dataset characterizing circadian and ultradian characteristics of a modest sample of (mostly acute) patients with a disorder of consciousness. They focus specifically on long-duration fluctuations as this is poorly addressed in the literature so far. They find that a recovery of circadian rhythms (measured with EEG behavior and hormonal biomarkers) is associated with favorable outcome. Although the investigation seems comprehensive, the paper is really not straightforward to read and I suggest the authors to seriously rewrite the paper. Also, there are some methodological points that need to be addressed.

Major points:

The introduction provides interesting background information but is very long. I encourage the authors to reduce the review-like style and only highlight literature directly relevant to the empirical results presented in the paper. E.g., The paragraph starting at line 165 poses relevant questions in the field, but these are not the main questions under investigation in the paper. There are more sections like that. Also the discussion is very lengthy, which makes it hard for the reader to grasp the main take home messages. Editing the discussion would really help to convey the main messages of the results. E.g., the last paragraph of the discussion come a little out of the blue and either needs clear explanation or should be removed).

As mentioned by the authors, the suprachiasmatic nucleus [SCN] of the hypothalamus is important for the circadian rhythms. Likewise, DoC patients usually present with severe cortical lesions that might be related to circadian alterations (see <https://pubmed.ncbi.nlm.nih.gov/35793707/>) but independent from consciousness levels per se. Is any (quantitative or qualitative) neuroimaging available for these patients to assess lesions (in the area of) the SCN and cortical areas?

After reading "Nevertheless, it remains debated whether the behavioural assessment of eyelid movements is a reliable marker of wakefulness fluctuations because the heterogeneity of lesions could imply dissociated patterns" in the introduction, I am surprised to read that eye opening and closure has been considered as the gold standard for awake and sleep in these patients. What does that mean for the interpretation of the results? Do they really reflect how well the circadian rhythm is maintained, as measures by various techniques, or rather how well these techniques follow eye opening and closure patterns? One way the authors might attempt to deal with this is by considering 10-minute epochs only, however, the choice of the 10-minute epoch should be justified. Do the results change with shorter or longer epochs? As the hallmark of non-favourable outcome is the lack of rhythmicity, I would like to understand better how much fractured the rhythms really are.

The results are started with a 2-page long succinct description of methods. This part is hard to follow as not enough detail is provided (e.g., line 212 mentions 4 groups of measures, 3 only are mentioned, in the figure only the 4th is mentioned). Furthermore, I wonder whether this is needed at all. Perhaps a VERY brief description of the data used (including important details as how favourable outcome is defined, and other information needed to place the results in the context) in the different analysis right before presenting the results is enough to convey the main findings convincingly.

The sample size is limited, which I can understand given the large number of different datasets used for every subject. That is a nice feat, yet I would therefor keep the dataset homogenous and not include chronic DoC patients, and neither the 2nd assessment of the two patients. Furthermore, I have serious concerns regarding the healthy volunteer "group" consisting of 3 subjects. I doubt that

the conditions in which the HC's were acquired was the same as the patients (this is not explained clearly), meaning in a hospital room without getting out of bed. This is an important source of bias in the acquisition of the data. At the level of analysis, statistics comparing 3 subjects against 19 should be done very carefully and the interpretation even more so. I suggest to remove the HC group from the paper.

Was there any correction for multiple comparisons used? E.g., the EEG correlates were used to identify 3 different outcome measures (HC/DOC, that I suggest to remove; cortical function; functional outcome). Although the statistics used in either of these investigations is non-parametric, the tests are repeated 3x.

The CRS-R scores are used as proxy for cortical function. I find that a very unusual choice (that is not described very clearly from the beginning of the results) and suggest the authors to refer to that as behavioral function instead. As we know, many DoC patients are unable to show behavioral signs of consciousness despite being covertly conscious (e.g., see the work of Jan Claassen and Aurore Thibaut on CMD and MCS*). If the authors would like to assess cortical function, they need to correlate the EEG findings to a direct neuroimaging measure of cortical function instead. Furthermore, if the CRS-R total scores are used, this is not informative for diagnosis/ behavioral "cortical" function, as total scores of reflexes only can overlap with "cortically mediated behavior". A better choice would be to use the CRS-R index, a linear score considering both reflexes and non-reflexes with a cut-off (<https://pubmed.ncbi.nlm.nih.gov/31319707/>).

The discussion regarding global and local states is interesting but needs refinement. In its current form it seems that the authors suggest that global state fluctuations could only occur at larger timescales. I am not sure this is supported by the literature (or cited work).

Table 1: the SD for the absolute alpha power are enormous. Is there something wrong with the data?

Minor points:

Could the authors detail what "convenient thresholds" are in: Finally, the results were dichotomised as "presence" or "absence" of circadian/ultradian rhythms according to convenient thresholds for the correlation fits.

The results sections should each mention the number of subjects in either group, as well as the mean and SD for the values of the metrics.

The authors describe in the discussion the importance of "up and down" states of wakefulness. I suspect they mean wake and sleep? I would consider rephrasing as up and down states usually refer to states modulated by slow waves/ low delta.

How often was the CRS-R performed, in order to attenuate the effect of arousal fluctuations on clinical diagnosis.

How were discontinuities in the EEG signal treated?

A description of measures specifically used in circadian rhythm research (e.g., acrophase) is needed.

Figures 3/4: The images in 3D space very strange. What do the small/big bright/dull spots represent. This should be explained in the caption.

Figure 7: not all percentages add up

Reviewer #2 (Remarks to the Author):

Gobert et al.: Recovered predictable fluctuations and 24-hour rhythmicities in coma are associated with a favourable outcome

In this manuscript, Gobert et al. investigated markers of circadian/ultradian rhythms at several levels in a sample of patients suffering from disorders of consciousness following severe brain injury and coma. Importantly, on a descriptive level, they relate these markers to the patients' recovery and, on this level, establish markers of circadian rhythmicity as predictors for recovery.

Generally, the manuscript is nicely in line and expands previous research and adopts an interesting perspective of rhythms across the body including circadian rhythms in EEG signals. For now, the predictions and use of the additional measures are only at a descriptive level. Unfortunately, the manuscript in its current state was not easy to read and follow for me. This culminates in a very long discussion, where the main aspects unfortunately become lost. Likewise, the figures are extremely hard to understand, also because they are partly not readable/ visible. In addition to my detailed comments below, I think the manuscript will strongly benefit from a reduction to an essential storyline and a concise discussion. Generally, I recommend the paper to be proof-read by a native speaker, who should also support the authors to adopt an easier writing style.

Abstract

- I am unsure whether "long-term modulations" in consciousness is really a good description of alterations between sleep and wakefulness, especially as it is mentioned in one breath with the contents of consciousness, i.e., awareness. I am thinking, perhaps "regular modulations in arousal" (a prerequisite for consciousness) might be a better alternative?
- I would suggest to use "secretion" instead of "excretion". A hormone is secreted from a gland rather than excreted.
- Please include a minimal degree of details about the patients (sample size, age range, time since injury, how many were in UWS/MCS).
- Language: "...has previously been related..." (please correct word order)
- Late or later "recovery of consciousness"?
- Please mention in the abstract that this was an exploratory study and that the predictions were only on a descriptive level.

Introduction

- I like the idea of starting the text with a question. However, to me, the first paragraph seems to take too many turns and it is not fully clear to me. Even stimulation is not "within the blink of an eye" and a sleeping person will eventually respond to stimulation, whereas a person in coma won't. Indeed, using the Coma Recovery Scale-Revised (CRS-R) a patient will be diagnosed with "coma" at a given time if s/he remains completely unresponsive.
- Line 113: doesn't wakefulness and (waking) awareness also disappear during REM?
- A bit of a provocative thought: isn't what the authors call "static" only really static if you only assess consciousness once? Why, from the authors' perspective isn't it enough to just assess consciousness repeatedly, say, every hour?
- Lines 133 et seqq.: according to the CRS-R minimal consciousness is also that someone follows an object with his/her eyes. What is the reason for introducing "situations of danger" in this context? Furthermore, I would appreciate a better introduction of the "embodiment of mind" hypothesis if it is really relevant. I do not think most people involved in disorders of

consciousness research let alone circadian research are familiar with it. I think these are just two examples where the authors veer off the storyline.

- Line 141: note that this is only true during sensitive phases, i.e. during the night (as the cited literature suggests)

- Line 163 et seq.: I have difficulties understanding what the authors wish to say. Please consider revising.

- Line 633: did any of the patients have legal representatives? If yes, please add this in addition to "relatives".

Methods

- Patient characteristics: as they were probably not normally distributed, please also indicate median and 25%/ 75% quantiles. Average and SD is only informative in case of normally distributed values.

- Line 644: in addition to the CRS-R sum score at study inclusion, please also give the diagnosis. The sum score is not (necessarily) related to the diagnosis.

- Line 650: This refers to three additional patients if I am not mistaken? Please do not call it "three assessments" as one can easily think that these are assessments in the sense of a part of a study protocol.

- Line 664: after or during the 2-year follow-up? If after, how was this assessed systematically and when?

- Line 705: how did you perform the sampling? Did you empty the catheter bag every 2 hours and sample from the urine collected during the past two hours? Please include additional details.

- Line 789: why was the "circadian" fit limited to one hour? Previous work has suggested that for DOC patients, the dominant rhythmicity can lie outside this range that has been found for healthy individuals. Also, free-running rhythms can sometimes be as long as 25.2 hours. Admittedly, DOC patients may rather undergo a "free running" protocol than a "constant routine"?

- How did you select the EEG features you have computed for your analysis? Have they previously been implicated in the detection of consciousness? It sometimes sounds a bit as if they had been computed because the python package allowed to do so.

- Did the nursing staff also provide informed consent for being filmed? Please mention.

- How was an ultradian rhythm defined?

- Please give credit to the authors of the R packages used by citing the packages in addition to R.

- I would suggest the authors to visualise their statistical approach. It is currently very hard to understand. I am not sure, but if it makes sense, one could perhaps split Fig. 1 up into two figures.

- If I am not mistaken, information about the luxmeter measurements are missing completely (i.e., device, sampling rate, direction of measurements, etc.)

- Do the authors know anything about the orientation of patient rooms and whether they had sunlight?

Results

- Line 218: why did the data have to be scored manually when there was a software?

- Why does 2-hourly sampling prevent the analysis of ultradian rhythms?

- Figure 1: Is there an x-axis as suggested by the arrows? If yes, what is the axis exactly? I find it hard to understand when the multiple assessments took place, i.e. the study protocol. I would also suggest to distinguish between raw data acquired and the variables computed.

- Figure 2: Axis labels are so small they can only hardly be read or not at all. Even though it might be an issue with the resolution here, it is too small. Referring to the title, please note

that the plural of "analysis" is "analyses", which you may want to use here. Further, please avoid unnecessary abbreviations in the figure legend. Last, I would like to suggest using a colourblind-friendly palette such as the "viridis" palette in R.

–Figure 3: is there an alternative to the black background? Unfortunately, the axes are barely visible and thus the figures are hard to read.

–Figure 5 is again a bit overwhelming I have to admit. Please try to visualise in an intuitive way. Figures should support the readers' understanding and not a lot of extra effort to understand. What does "chronic" mean when it is about E-MCS yes/no?

–Figure 6: circadian/ultradian figures are impossible to read. Please adopt appropriate font sizes for axis labels. Also, what does the "bis" in the title mean?

–Table 1: please harmonise font sizes. Also, please provide a table caption.

–For all figures and tables, I encourage the authors to include them and their captions in the text no matter what the formatting guidelines say (personally, I always state in the cover letter that I will leave it like this for the reviewing process). You can still reformat later if at all requested (I have informed the editorial office that it is just super annoying like this). I admit that this may partly also have added to the difficulties I had to understand the figures – sorry!

Discussion

–10.5 pages of discussion (excluding the supplemental discussion) are too long. Importantly, this is not because I do not like to read so much, but because the main story line gets lost. Eventually, it will lead to the paper not being read/cited as much as it could, because readers do not understand its importance. I would therefore suggest to focus and address the 3-4 most important findings and discuss them in a concise manner.

–Line 636: if the authors wish to announce with this paragraph what they will do next, I recommend omitting such "meta"-phrases and simply continue.

–Line 410: what is the group level? To which groups do you refer?

–Line 529: I am sorry, I am a bit lost by now, but what EEG pattern do you refer to? Perhaps you can direct the reader to the relevant figure?

If I can be of any help in further clarifying my comments, the authors are welcome to contact me

under christine.blume@upk.ch.

To the Reviewers,

Please find herein the “responses to referees” letter that addresses the referees’ comments in a point-by-point manner concerning our manuscript entitled “Twenty-four-hour rhythmicities in disorders of consciousness are associated with a favourable outcome.”.

We would like to thank the reviewers for the thorough evaluation of our work. As expected for this major revision, we have taken all the comments into account and made profound changes in the manuscript to tackle the issues raised by these justified remarks.

Thank you for your consideration,

Yours sincerely,

Dr Florent Gobert, M.D., Ph.D., on behalf of the co-authors

Reviewer #1 (Remarks to the Author):

The authors provide the analysis of a thoroughly acquired dataset characterizing circadian and ultradian characteristics of a modest sample of (mostly acute) patients with a disorder of consciousness. They focus specifically on long-duration fluctuations as this is poorly addressed in the literature so far. They find that a recovery of circadian rhythms (measured with EEG behavior and hormonal biomarkers) is associated with favorable outcome. Although the investigation seems comprehensive, the paper is really not straightforward to read and I suggest the authors to seriously rewrite the paper. Also, there are some methodological points that need to be addressed.

Major points:

The introduction provides interesting background information but is very long. I encourage the authors to reduce the review-like style and only highlight literature directly relevant to the empirical results presented in the paper. E.g., The paragraph starting at line 165 poses relevant questions in the field, but these are not the main questions under investigation in the paper.

The reviewer is right, the introduction has been simplified to focus on the hypothesis and methodology of the study.

There are more sections like that. Also the discussion is very lengthy, which makes it hard for the reader to grasp the main take home messages. Editing the discussion would really help to convey the main messages of the results. E.g., the last paragraph of the discussion come a little out of the blue and either needs clear explanation or should be removed).

We have performed an important shortening of the discussion section, some minor points having been transferred to the Supplementary Information section for the most interested readers.

Concerning the last paragraph, we agree that this synthetic message can be difficult to tackle but, as we believe that this part constitutes the take-home message of our work and the basis for further explorations, we propose to keep it in place with more accurate explanations.

“Altogether, the present study outlines a theoretical chronology of the favourable recovery from coma based on *long-term rhythm analysis*. Despite the absence of an individual description of the natural history of each coma recovery, it could be used to build a novel methodology to assess the interactions between the content of consciousness and wakefulness assessments. At first, the rhythmic pattern would consist in an overall chaos for most patients. Then, if the brainstem metronome of frail wakefulness modulations rises again, it could be assessed by the metrics of fluctuation amplitude and predictability or, in rare instances, by a transient ultradian rhythm (in absence of circadian repression). Later, the reappearance of a circadian rhythm for one or some modalities, then for all of them, and at last, its harmonisation with the environment would be the final step of recovery.”

As mentioned by the authors, the suprachiasmatic nucleus [SCN] of the hypothalamus is important for the circadian rhythms. Likewise, DoC patients usually present with severe cortical lesions that might be related to circadian alterations (see <https://pubmed.ncbi.nlm.nih.gov/35793707/>) but independent from consciousness levels per se. Is any (quantitative or qualitative) neuroimaging available for these patients to assess lesions (in the area of) the SCN and cortical areas?

A dedicated analysis has been carried out to answer this legitimate question. It appears that hypothalamic lesions were rare overall (4 occurrences on the left side, as described in Supplementary Table 2-C) but that they were absent in the group of patients with a homogeneous presence of circadian rhythms.

Further anatomo-clinical considerations are now provided in the Results section of the Supplementary Information (“Radiological features of patients’ lesions”) as well as in 3 Supplementary Tables (2 A-B-C). Moreover, introducing the MRI analysis allowed to illustrate a differential mechanism of Alpha rhythm involvement (Supplementary Results: “Insights from the circadian rhythms absent in healthy participants” and Supplementary Figure 6) that was previously removed from the manuscript to avoid lengthening it.

If the increase in length of the paper is refused by the editor and if this particular point is not regarded as mandatory by the reviewers, we propose that the changes made in the revised version of the manuscript are removed to be published elsewhere as an independent paper.

After reading “Nevertheless, it remains debated whether the behavioural assessment of eyelid movements is a reliable marker of wakefulness fluctuations because the heterogeneity of lesions could imply dissociated patterns” in the introduction, I am surprised to read that eye opening and closure has been considered as the gold standard for awake and sleep in these patients. What does that mean for the interpretation of the results? Do they really reflect how well the circadian rhythm is maintained, as measures by various techniques, or rather how well these techniques follow eye opening and closure patterns?

Although we believe that this behavioural marker is in fact a poor marker of wakefulness, which is regarded herein as the “global state” variation entraining a modulation of every “local state” at once, we decided to keep eyelid movements for further analysis but by delineating its function for an objective assessment of the nyctemeral changes of eye-opening/closure periods. This choice was made in the context where the concept of arousal/wakefulness is still discussed¹ or used² by default. To our knowledge, the more subtle modulations of both dimensions of consciousness (arousal/wakefulness and awareness/content of consciousness) with a finer granularity of biomarkers to define alertness, vigilance, attention^{3,4,5} cannot yet be assessed easily in this setting.

Of note, in this study, we chose not to use the notion of “sleep and wake cycles”, described as necessary for the definition of PVS⁶, due to the impossibility to build hypnograms.

“What happens when eyes open and close” is the subject of a future study analysing the correlation between eye-opening moments (not as gold standard of wakefulness as defined previously, but as its challenged behavioural surrogate) and neurophysiology (same EEG parameters used herein). Based on these still unpublished results, we are now more firmly convinced that the link between behavioural and neurophysiological changes are not straightforward. In particular, it appears that when eyes are closed in phase with a peculiar

EEG pattern compatible with sleep, this constitutes an endophenotype in this acute DOC population which could have a prognostic value *per se*.

The issues raised in this comment are discussed in the paragraph “Cerebral and eye-opening/closing fluctuations during coma” of the Discussion section, which has been moved to the Supplementary Information section following the comments from Reviewer 2.

One way the authors might attempt to deal with this is by considering 10-minute epochs only, however, the choice of the 10-minute epoch should be justified. Do the results change with shorter or longer epochs? As the hallmark of non-favourable outcome is the lack of rhythmicity, I would like to understand better how much fractured the rhythms really are.

We agree with the reviewer that the choice of 10-min epochs was somehow arbitrary and that, in the absence of comparable studies, it cannot be justified by the previous literature as a smoothing period. However, 5 continuous minutes of eye-closing were also arbitrarily considered as relevant in previous work⁷ and EEG data were also averaged over 5 minutes without further justification (“The retained 4 s epochs were then grouped into 5 min consecutive intervals”⁸).

The main interest of using such a smoothing consisted in both reducing the dimensionality of the data and avoiding missing data (e.g. nursing periods, which are usually shorter than 10 minutes but often longer than 5 minutes).

The risk of missing some rhythms below a 10-min period cannot be ruled out. However, previous steps of analysis (not shown in the manuscript but used as an illustration in this response) indicated that non smoothed data appeared as noisy, and that the main relevant fluctuations were kept with the 10-min smoothing choice. Analysing the fluctuations below 10 minutes would request a careful selection of the periods without any artefact but in such an ecological context, it would miss the point of a continuous analysis of the data.

The choice of this period is justified in the Methods/Preprocessing/EEG analyses section:

“The *Dominant Period* of ultradian fluctuations was calculated using R (RCore Team, 2015, R: A language and environment for statistical computing, R Foundation for Statistical Computing, Vienna, Austria, *spectrum function*) and was comprised between 20 minutes and 24 hours (according to the Nyquist-Shannon theorem as the sampling was 10 minutes after smoothing, which is compatible with tracking classic ultradian rhythms in DOC⁸). The 10-minute smoothing allowed enough values to manage statistical analysis of distribution and correlations between pairs of features but missed the fluctuations below 10 minutes. However, this threshold remained compatible with the analysis of usual ultradian oscillations (according to the theoretical framework of the Basic Rest-Activity Cycle or BRAC⁹ initially described around 40 minutes) in particular around 70 minutes as recently demonstrated for DOC⁸. In case of missing values despite smoothing, a linear interpolation was performed before running the *spectrum function*.”

Altogether, this 10-min compromise allowed to demonstrate the presence of 20-min fluctuations (according to the Nyquist – Shannon theorem), which was more than 3 times quicker than the ultradian fluctuations assessed in the literature.

To confirm that the differences remain negligible when interpreting raw or smoothed values, we tested, for 3 consecutive patients, the differences in the mean and SD for both types of values.

- For each electrode, we calculated the difference within the mean values and SD values for 5 EEG features (Determinism, Delta R, Theta R, Alpha R, Beta R).
- Then, we calculated the averaged differences for each feature (averaged differences of mean and averaged differences of SD).
- Finally, we proposed a more synthetic view of this comparison by providing a super-average across these differences per feature.

Eventually, we observed that the differences between the raw and smoothed data are negligible and comparable across the 3 tested patients (around 10^{-4} for means and 10^{-3} for SD).

	Super-average of differences between mean values	Super-average of differences between SD values
Test 1	-0,00031	0,00529
Test 2	-0,00021	0,00743
Test 3	-0,00037	0,00358

The results are started with a 2-page long succinct description of methods. This part is hard to follow as not enough detail is provided (e.g., line 212 mentions 4 groups of measures, 3 only are mentioned, in the figure only the 4th is mentioned). Furthermore, I wonder whether this is needed at all. Perhaps a VERY brief description of the data used (including important details as how favourable outcome is defined, and other information needed to place the results in the context) in the different analysis right before presenting the results is enough to convey the main findings convincingly.

This mistake was corrected, and the above-mentioned paragraph was reduced and now integrates a part of the Introduction section from the initial version of the manuscript which has also been simplified and reduced.

“In brief (see Fig. 1 and the Methods section for details), synchronised assessments of behavioural (continuous video of eye-opening/closing using a dual-camera recording), environmental (nursing periods assessed by continuous video, sound and light levels assessed by continuous sonometer and luxmeter recordings respectively), melatonin/cortisol urinary excretions (every 2 hours, to assess the endogenous drive by clock-controlled hormones), and

neurophysiological markers of brain function (13-channel EEG) were performed in 18 acute DOC patients.

We evaluated *long-term variables* (24-hour distribution, *long-term predictability*, and circadian/ultradian rhythmicity) of various short-term EEG features, the combination of which led to the analysis of 168 EEG parameters. The EEG features were selected through a hypothesis-free approach on the basis of their complementarity to describe EEG in its various dimensions and on their previous use in DOC patients. The EEG features retained were then grouped as power¹⁰, spectrum^{11, 12, 13, 14, 15}, complexity^{13, 14, 15, 16, 17, 18, 19, 20, 21}, and spatial variability²² (see Fig. 1 for description). Rhythmicity was also calculated for eye-opening periods (circadian and ultradian) and for hormonal results (circadian only due to a difference in the sampling rate with that of EEG and behaviour). Rhythmicity was then dichotomised as present or absent for each of these modalities.

First, a supervised data-driven analysis was used to evaluate the EEG fluctuations that were indicative of: i) the initial behavioural cortical function status (based on the CRS-R) and the final functional outcome (based on the GOS); ii) the circadian rhythmicities of behaviour and clock-controlled hormones²³. Second, circadian and ultradian rhythmicities regarding brain (from EEG), eye-opening moments (from video), and hormones (from urinary clock-controlled excretion) were also evaluated with the aim to classify patients according to their proximity with normal rhythmicity patterns. Third, the prognostic value of the simultaneous presence of these 3 circadian rhythms was compared to usual markers to predict a favourable outcome, defined as a full recovery of consciousness (i.e. patients reaching the Exit-MCS level, including some cases with severe disability).

The sample size is limited, which I can understand given the large number of different datasets used for every subject. That is a nice feat, yet I would therefore keep the dataset homogenous and not include chronic DoC patients, and neither the 2nd assessment of the two patients. Furthermore, I have serious concerns regarding the healthy volunteer “group” consisting of 3 subjects. I doubt that the conditions in which the HC’s were acquired was the

same as the patients (this is not explained clearly), meaning in a hospital room without getting out of bed. This is an important source of bias in the acquisition of the data. At the level of analysis, statistics comparing 3 subjects against 19 should be done very carefully and the interpretation even more so. I suggest to remove the HC group from the paper.

We agree that, due to logistical issues related to the lack of access to ICU rooms for healthy subjects, some biases could not be tackled appropriately. Therefore, the illustration based on the comparison of “healthy subjects vs DOC” has been displaced in the Supplementary Information (Supplementary Figure 7) and we now only present the patients’ results for this analysis in the main text.

However, all the results from the Figure 5 were built on the definition of normative data and thus based on this small sample of healthy volunteers, and can hence not be removed from our dataset. Importantly, previous studies on the circadian rhythms of healthy subjects for EEG have illustrated the robustness of EEG changes during experimentally modified nyctemeral periods. This consistency concerned for example 19 subjects for Aeschbach et al in 1999²⁴ and 7 subjects for Cajochen et al. in 2002²⁵. Both studies illustrate that a circadian modulation of the EEG spectrum can be observed even in a small group of subjects. The first study demonstrated a circadian modulation of the Theta and Alpha bands and a complementary Theta and Delta band modulation by the homeostatic process²⁴. This circadian control of Theta and Alpha bands, the homeostatic control of Delta and mixed control of Beta bands was confirmed in an independent study²⁵.

However, one issue that could be raised is that the EEG spectrum metrics were not the ones with the strongest circadian fit for healthy subjects, unlike the short-term complexity (Determinism, DFA) and Spatial Variability (Alpha, Beta) used in our study. As these metrics are either novel or rarely used in circadian literature, we cannot provide direct comparisons with our healthy group. The circadian rhythm observed in the literature for EEG spectrum appeared therefore as significant only at the population level and was not observed in our sample. On the contrary, these innovative EEG parameters appeared as having an even stronger circadian drive (see below) as they were constantly significant for each healthy subject, and could therefore be regarded as circadian markers at the individual level as well.

	DeltaR	ThetaR	AlphaR	BetaR	Determinism	DFA	AlphaR_Variability	BetaR_Variability
S01	0	0	0	0,156	0,2	0,118	0,387	0,399
S02	0,182	0,183	0,14	0	0,268	0,471	0,425	0,522
S03	0,413	0	0	0,14	0,474	0,204	0,449	0,636

Regarding the inclusion of a chronic DOC patient and 2 assessments for 2 patients, this only has a limited consequence on the overall results concerning the factors shared by all DOC patients (existence of behavioural or biology circadian rhythms, behavioural cortical function). Indeed, every patient in this cohort was behaviourally homogeneous as we did not include every patient (such as those with a late awakening) but only patients with a persistent DOC after 6 months. Excluding these “chronic” patients would reduce even more the

statistical power of the analysis for pathophysiological considerations which are shared by all DOC patients.

Importantly, these patients were not included in the most clinically relevant analysis “functional outcome”, which was the only one that was concerned by an inhomogeneity issue within this group (delay since coma), and which constitutes to the main message of the manuscript.

Was there any correction for multiple comparisons used? E.g., the EEG correlates were used to identify 3 different outcome measures (HC/DOC, that I suggest to remove; cortical function; functional outcome). Although the statistics used in either of these investigations is non-parametric, the tests are repeated 3x.

As stated in the section Methods/Statistics/Supervised data-driven analysis for comparing neurophysiological data with behavioural and hormonal assessments, a correction for multiple comparisons was used. Indeed, an even more severe correction for multiple comparisons than the one suggested was used (*5 and not *3) for these parametric tests because it was applied to every factor we tested (3+2 in two different paragraphs), and we chose to keep it that way despite the fact that we removed one of these factors from the main results.

“The statistical significance of between-group differences was tested by a Multivariate ANalysis Of VAriance [MANOVA] (RCore Team, 2015, Rcmdr package, *manova function*) with a corrected threshold for 5 factors using the Bonferroni method (corrected p-values < 0.01). In addition, the False Discovery Rate [FDR] and Likelihood ratio [LR]²⁶ were assessed *post hoc* to evaluate the statistical results at the model level using the False Positive Risk Web Calculator (version 1.7 Longstaff,C. and Colquhoun D, <http://fpr-calc.ucl.ac.uk/> Last accessed 2021-01-07, with a conservative setting : “p-equals case” method, *prior probability = 0.5, effect size = 1*)²⁶.

The CRS-R scores are used as proxy for cortical function. I find that a very unusual choice (that is not described very clearly from the beginning of the results) and suggest the authors to refer to that as behavioral function instead. As we know, many DoC patients are unable to show behavioral signs of consciousness despite being covertly conscious (e.g., see the work of Jan Claassen and Aurore Thibaut on CMD and MCS). If the authors would like to assess cortical function, they need to correlate the EEG findings to a direct neuroimaging measure of cortical function instead. Futhermore, if the CRS-R total scores are used, this is not informative for diagnosis/ behavioral “cortical” function, as total scores of reflexes only can overlap with “cortically mediated behavior”. A better choice would be to use the CRS-R*

index, a linear score considering both reflexes and non-reflexes with a cut-off
(<https://pubmed.ncbi.nlm.nih.gov/31319707/>).

The reviewer is right: we wanted to design an analysis based on the cortical function defined behaviourally from the “CRS-R * items” according to the interpretation from Lionel Naccache²⁷.

We agree that, in the absence of an active paradigm included in the protocol, the existence of a CMD for some patients could not be formally ruled out. As mentioned several times in the Introduction and Methods sections, we referred in these instances to the concept of CMS which is, by default, defined behaviourally based on the CRS-R. To avoid any further confusion, we now refer explicitly and not implicitly to the CMSb group (so neither the CMSa nor the MCS*)²⁷. In order to do so, the clinical factor “Cortical function” is now defined as “Behavioural cortical function” throughout the revised version of the manuscript.

Since the CRS-R total was never used in the behavioural classification process as it was considered irrelevant for our purposes, it was not mandatory to replace it by the CRS index.

The discussion regarding global and local states is interesting but needs refinement. In its current form it seems that the authors suggest that global state fluctuations could only occur at larger timescales. I am not sure this is supported by the literature (or cited work).

We agree that this concept was rarely used in the literature since its description. Therefore, it is not possible to mention many references to justify its use. The most recent description in a Nature Reviews Neuroscience paper (in press) by the author who first defined this concept seems to confirm its use for long-term modulation of arousal: “States of consciousness can be grouped into two classes: global states and local states. Global states concern an organism’s overall subjective profile and are associated with changes in arousal and behavioural responsiveness. Familiar global states include wakefulness, dreaming, sedation, the minimally conscious state, (perhaps) the psychedelic state, and so on”.

So, one could speculate that global states can fluctuate at larger time scales when prolonged recordings are used to assess arousal/wakefulness modulation but, as the brain function can change sharply in some physiological and pathological circumstances (see the flip-flop model and our proposal for a pathological acceleration of its changes in coma recovery), it is very well possible that the global state could change within a few minutes as well. The narcoleptic condition is a perfect model to address this issue, as patients can switch from wake to a paradoxical sleep at once with a huge change in their global state but a limited change in their awareness/conscious content.

Since we focused on long-term modulations in the present study, we indeed assumed to neglect possible changes occurring below the 20-min horizon allowed by the 10-min smoothing. Of note however, a complementary analysis was planned during the preprocessing step of the raw EEG analysis to look for very short-term global state changes, by selecting artefact-free moments that could allow to address this issue.

Table 1: the SD for the absolute alpha power are enormous. Is there something wrong with the data?

As the Absolute power are, by definition, not normalised, there was indeed a great variability between patients and within patients during the 24h. This variability reflects the difference in the EEG signal (Total power) as well as the difference in the spectral composition (Alpha power and others) and the differential changes of this Alpha power over time. These huge values for Absolute power were not specific to the Alpha band but were observed for the Total power and the other bands (with lower values for higher frequencies according to the power law relationship between EEG amplitude and frequency).

Total_Mean	DeltaA_Mea	ThetaA_Mea	AlphaA_Mea	BetaA_Mean	Total_Std	DeltaA_Std	ThetaA_Std	AlphaA_Std	BetaA_Std
23385690,8	19935506,2	2157292,28	158458,235	1685,89072	11810565,6	7608010,59	636304,466	30586,3224	391,289652
36901873,22	23853055	5513238,97	904706,035	5562,21348	35095049,5	10899927,7	2302329,49	305046,806	1296,04534
87954264,05	79997036,4	2402620,46	430004,976	5144,51138	32916829,8	25005377,6	571123,788	111279,631	932,829604
9480511,093	4421426,66	697367,63	54168,5674	1097,59198	28895901,9	1269180,98	122077,656	8238,1782	263,717351
53474133,19	40684922,4	6641605,76	425761,705	4728,34068	26645803,6	6160361,51	487746,66	28813,633	831,39832
187673520,6	38080584,4	5352648,37	424180,889	5100,46323	1168036779	25849292,3	2385415,84	127178,181	1289,96233
54067890,66	44107584,9	4808551,13	405797,342	4107,47408	40668059,5	26938396,5	2041474,33	98589,1167	1315,7643
11740601,92	7258552,08	2578741,17	151571,824	2026,75088	8392761,9	2921361,13	504961,581	30071,0052	468,204837
47245517,84	18330100,5	1292708,17	125233,112	1369,92558	315726808	9079490,74	591325,112	40760,7883	314,662528
6324147,743	3894913,05	522086,136	168540,152	1255,98097	8100320,93	4088691,22	296698,001	59213,4021	637,931326
14514467,93	10997541	2887587,63	183821,456	1011,86302	6578623,63	4787503	1086660,75	49118,0851	248,237763
23005878,21	15235912,4	4815739,15	526788,413	3645,5798	35415421,3	28604302,4	1001263,53	96578,0125	721,56661
10354967,23	4627569,1	1073979,54	375620,87	5568,94662	23308597,5	3666548,55	671163,231	145574,117	1192,76735
3581060,789	1465915,53	123371,366	14867,8975	223,084774	12612299	613052,707	62334,6671	5051,52389	41,7491823
108156319	80115601,2	13438127,8	2428873,9	25089,1499	54093994,4	28351288,9	3272583,53	527934,669	6312,48147
5140410,706	3238621,13	453589,796	72981,912	565,130651	15615024,4	655241,895	97784,4946	16606,1153	67,1967699
527309748,4	15937517,5	3465059,57	565985,342	3713,87354	2802047518	9405938,95	1303800,13	170252,886	1030,09434
62238601,03	56423975,5	4409594,03	455770,044	3231,91201	12959206,5	8861196,04	387114,75	47399,4741	402,257564
66621120,76	27493955,3	12620625,3	553161,523	5251,74392	122819964	15945500,3	6111853,46	100411,191	958,770393
47305985,12	38876238,6	6919128,34	696248,781	7382,97018	12824222,9	10073447,7	895391,528	135871,541	3443,30308
116092501,5	5709355,38	2277179,9	516872,267	4645,11823	633060642	3283515,44	528322,735	71320,7072	866,643554
34079988,85	24085645,2	3697736,71	734048,346	7075,88414	25017026,1	13714604,7	1716887,6	312652,826	2998,58759
13826861,92	10241233,8	2701173,6	441496,296	4764,76811	7731379,29	4939751,64	1082103,31	144569,689	1626,27688
31453063,89	19771709,1	3315353,31	769940,158	5954,33973	45920083,8	25198471,7	2355657,92	505281,792	2066,73959

Minor points:

Could the authors detail what “convenient thresholds” are in: Finally, the results were dichotomised as “presence” or “absence” of circadian/ultradian rhythms according to convenient thresholds for the correlation fits.

As stated in the Methods/Preprocessing/EEG analyses section, the threshold definition was built thanks to the normative values issued from the healthy subject analysis:

“Notably, the existence of this rhythmicity was descriptive and did not assume an unequivocal implication of the circadian pacemaker (SCN) in its genesis. The best ultradian fit was constrained around the Dominant Period provided by the spectrum function (+/- 5%), provided that the Dominant Period was not 24 hours (theoretical maximum = 12 hours; theoretical minimum = 20 minutes. If the Dominant Period was 24 hours, it was reported as such for this variable but without ultradian rhythmicity. Regression coefficients (R^2) were reported as continuous parameters if they reached a minimal threshold ($R^2 > 0.1$).”

Then:

“From the previous analyses, we built a new categorisation of patients according to the presence/absence of circadian (24-hour) rhythms and ultradian rhythms in EEG. In healthy participants, circadian rhythms were found for short-term complexity (Determinism, DFA) and the Alpha and Beta Spatial Variability (mean $R^2 > 0.2$, minimum $R^2 > 0.1$). In contrast, no normal ultradian-like EEG pattern was observed. The patients were categorised as presenting a “normal circadian EEG pattern” if more than one of these 4 EEG features had a high circadian fit ($R^2 > 0.2$). An “abnormal circadian EEG pattern” was defined if higher circadian fits were observed for any other EEG feature not found in healthy participants ($R^2 > 0.3$). An “abnormal ultradian EEG pattern” was retained if higher ultradian fits were observed for any EEG feature ($R^2 > 0.3$).”

However, as we agree that the use of the term convenient could be misleading, we propose to explicit this point by a paraphrase that is now introduced in the Results section of the revised version of the manuscript:

Rhythmicity variables were calculated for EEG and eye-opening periods (circadian and ultradian) and for hormonal results (circadian only because of the 2-hour sampling rate) and dichotomised as presence or absence of these rhythms using ad hoc threshold definitions for the coefficient of correlation of fits which were both sufficiently high and validated a visual fit.

The results sections should each mention the number of subjects in either group, as well as the mean and SD for the values of the metrics.

As suggested, the number of patients in each group has been added.

Regarding the presentation of the mean and SD values in the text, we would prefer avoiding this presentation that is classic but difficult to read in this instance (after many attempts to make it accessible, we finally chose to create Table 1 (A&B) to increase clarity and readability for the reader).

The authors describe in the discussion the importance of “up and down” states of wakefulness. I suspect they mean wake and sleep? I would consider rephrasing as up and down states usually refer to states modulated by slow waves/low delta.

First, it is important to note that the use of the “up-and-down” term is not used for the results but for the interpretation of the fluctuations as “an up-and-down shape” rather than any “up-and-down states”. This precision was added in the revised version of the manuscript when the concept is introduced.

“This result delineates a sharp up-and-down shape for the patients presenting the more physiological profile. This graphical description of EEG fluctuation shape should not be considered as up and down states since the states of wakefulness herein could not be labelled as an optimum regarded as wake and a nadir regarded as sleep. .”

Concerning slow waves: after having failed to build hypnograms from the present polysomnographic recordings, we voluntarily chose not to use the “wake and sleep” determinant for our description as we were not able to define visually which periods were attributable to what state for each moment and for each patient. Although the fluctuations of the slow waves could be legitimately considered as a robust way to define them, the first visual EEG analysis underlined that this could not be univocal: several patients presented a paradoxical pattern with slow waves and a concomitant presence of muscle activity, indicating that the patient was possibly awake. Then, this pattern should be distinguished from slow-wave sleep-like patterns on the same EEG and for other patients. These observations are in line with the recent proposal that the presence of slow waves may not systematically rule out the occurrence of consciousness¹⁷.

Here: an illustration of a wake-like increase in slow-wave level with muscle activity

Here: a sleep-like decrease in slow-wave level without muscle activity and with spindles.

If requested by the reviewers, an improved version of these examples could be added in the Figure 2 (“Illustration of exploratory analysis”) or in the Supplementary Information section.

How often was the CRS-R performed, in order to attenuate the effect of arousal fluctuations on clinical diagnosis.

The reviewer is right as this point should have been mentioned. This methodological issue was added in the Methods section/Population description at inclusion:

“The CRS-R was performed only once, at the end of the recording, to avoid modifying the spontaneous behaviour related to naturalistic ICU stimulation (and its neurophysiological correlate).”

In addition, our analysis started before the proposal by Wannez et al. in 2017 to assess CRS-R at least 5 times over a 10-day evaluation²⁸. Importantly this proposal concerned chronic DOC patients.

We therefore acknowledge that the only continuous behavioural assessment concerned the wakefulness dimension (eye-opening) and that the definition of the behavioural cortical function was based on a single-point evaluation. This limitation has been added in the limitations paragraph of the Supplementary Discussion section accordingly.

“However, it is worth noticing that we performed continuous behavioural assessment only for the wakefulness dimension (eye-opening) of consciousness. On the contrary, the definition of the behavioural cortical function was based on a single-point evaluation by a CRS-R at the end of each recording rather than a 5-point assessment over a 10-day moment, as proposed by Wannez et al. in 2017 for chronic DOC patients (after the beginning of the present study²⁸).”

How were discontinuities in the EEG signal treated?

As explained in the response concerning the smoothing, smoothing was used for less than 10-min missing data, then, as expressed in the Methods/Preprocessing/EEG analyses section:

“In case of missing values despite smoothing, a linear interpolation was performed before running the *spectrum function*.”

A description of measures specifically used in circadian rhythm research (e.g., acrophase) is needed.

The acrophase assessment was not the focus of the circadian measurements (the circadian fit was prominent) but it was used as an additional result to compare the metrics between each other. Acrophase assessments were described as such in the Methods/Statistics/Rhythm's comparison section:

“Among the patients presenting a hormonal circadian rhythmicity (standing for the closest indicator of the circadian system function), the phase relationship with behavioural and cerebral circadian (24-hour) rhythmicity was described qualitatively as the difference between the acrophases of each modality (i.e. the peak of the circadian fluctuation, according to the sinusoidal estimation). The EEG and behavioural cycles were considered “in phase” with the melatonin/cortisol peak (depending on which fit was significant) if at least one EEG acrophase (among the 4 features considered as normally circadian) or the eye-closing acrophase had a phase angle of less than 3 hours with the hormonal peak.”

Then, the results are described in the Supplementary section: Phase relationship between circadian rhythmicity

“In the group of patients with a “homogeneous presence of all circadian rhythms”, the phase angle between melatonin and cortisol peaks was 6h for Patient 1, 7h for Patient 2, 2h for Patient 3 and 4h for Patient 4. For EEG features, the phase relation was close to the melatonin peak for 3 patients (Patient 1 and 2 when considering DFA and Determinism; Patient 4 when considering Beta Spatial Variability) and with the cortisol peak for Patient 3 only (considering Alpha and Beta Spatial Variability). The eye-closing maximum was in a close phase relation with the melatonin peak for Patient 1-2 and with the cortisol nadir for Patient 3-4.”

Figures 3/4: The images in 3D space very strange. What do the small/big bright/dull spots represent. This should be explained in the caption.

The caption was corrected.

Figure 7: not all percentages add up

The figure was corrected to indicate the % of patients with another outcome than Exit-MCS and MCS.

Reviewer #2 (Remarks to the Author):

Gobert et al.: Recovered predictable fluctuations and 24-hour rhythmicities in coma are associated with a favourable outcome.

In this manuscript, Gobert et al. investigated markers of circadian/ultradian rhythms at several levels in a sample of patients suffering from disorders of consciousness following severe brain injury and coma. Importantly, on a descriptive level, they relate these markers to the patients' recovery and, on this level, establish markers of circadian rhythmicity as predictors for recovery.

Generally, the manuscript is nicely in line and expands previous research and adopts an interesting perspective of rhythms across the body including circadian rhythms in EEG signals. For now, the predictions and use of the additional measures are only at a descriptive level. Unfortunately, the manuscript in its current state was not easy to read and follow for me. This culminates in a very long discussion, where the main aspects unfortunately become lost. Likewise, the figures are extremely hard to understand, also because they are partly not readable/ visible. In addition to my detailed comments below, I think the manuscript will strongly benefit from a reduction to an essential storyline and a concise discussion.

Generally, I recommend the paper to be proof-read by a native speaker, who should also support the authors to adopt an easier writing style.

Abstract

- I am unsure whether "long-term modulations" in consciousness is really a good description of alterations between sleep and wakefulness, especially as it is mentioned in one breath with the contents of consciousness, i.e., awareness. I am thinking, perhaps "regular modulations in arousal" (a prerequisite for consciousness) might be a better alternative?

The abstract was modified accordingly.

"Compared with the amount of attention that has been devoted to assessing the contents of consciousness, the long-term fluctuations of wakefulness have been rarely studied in patients with a disorder of consciousness (DOC) after acute brain injuries."

- I would suggest to use "secretion" instead of "excretion". A hormone is secreted from a gland rather than excreted.

We agree that the hormonal function is based on a secretion by a gland and this point was corrected accordingly. However, we used and kept "excretion" on purpose in the manuscript when we report the renal excretion of the hormones secreted in blood.

“A 24-hour assessment of brain (EEG), behavioural (eye opening/closing from video), and biological (clock-controlled melatonin and cortisol secretion from urinary excretion)”

- *Please include a minimal degree of details about the patients (sample size, age range, time since injury, how many were in UWS/MCS).*

The abstract has been modified accordingly. However, it now exceeds 200 words and its length thus needs to be validated by the editors.

“18 acute brain-injured patients (mean age: 51 (\pm 16.5); time since injury: 25 (\pm 14) days; status: 3 Coma, 12 Unresponsive Wakefulness Syndrome [UWS], 3 Minimally Conscious State [MCS]).”

- *Language: “...has previously been related...” (please correct word order)*
- *Late or later “recovery of consciousness”?*

The abstract was modified accordingly.

- *Please mention in the abstract that this was an exploratory study and that the predictions were only on a descriptive level.*

The fact that this is an exploratory study is now mentioned in the abstract:

“While the recovery of consciousness has previously been related to a high and short-term complexity, this exploratory study shows the importance of the high predictability of the long-term generation of brain rhythms. It also highlights the importance of circadian body-brain rhythms in awakening.”

In addition, a more detailed paragraph regarding the interpretation of the results at a descriptive level has been added in the conclusion of the main manuscript.

“Although they were not cortically-mediated, these biomarkers were related to the later recovery of awareness in this exploratory study in which predictions were made on a descriptive level and should be confirmed in additional validation cohorts.”

Introduction

- I like the idea of starting the text with a question. However, to me, the first paragraph seems to take too many turns and it is not fully clear to me. Even stimulation is not “within the blink of an eye” and a sleeping person will eventually respond to stimulation, whereas a person in coma won’t. Indeed, using the Coma Recovery Scale-Revised (CRS-R) a patient will be diagnosed with “coma” at a given time if s/he remains completely unresponsive.

This paragraph was corrected to include these remarks.

“Although this usually becomes self-evident after a few stimulations, such an assessment remains trickier in comatose individuals suffering from severe brain damages that alter the spontaneous behavioural expressions of internal states, the responsiveness to environment including the communication of self-report, and wakefulness.”

- Line 113: doesn't wakefulness and (waking) awareness also disappear during REM?

The different position of each type of sleep have be distinguished in this theoretical perspective.

“The static description of wakefulness and awareness states is intrinsically limited but operational for healthy human participants: the nycthemeral routine (i.e. the transient disappearance of these states during sleep²⁹) is in phase with the environment and can be easily confirmed by behaviour, with co-variations in both dimensions (for non-Rapid Eye movement [REM] sleep at least, as their modulations during REM sleep is paradoxical by definition).”

- A bit of a provocative thought: isn't what the authors call “static” only really static if you only assess consciousness once? Why, from the authors' perspective isn't it enough to just assess consciousness repeatedly, say, every hour?

This comment raised an important question, and we thank the reviewer for it.

In the daily routine, the patients' state can be defined by several ways:

- He/She is in one state, but the assessment is not relevant enough to acknowledge its stability all day long.
- He/She presents a unique wake state, which allows to obtain the best assessment, thus defining the best position of the patient in the 2D over-simplistic representation of Wakefulness and Awareness. Then, fluctuations are attributable to sleep-like patterns (e.g. such as an MCS patient presenting a pathological sleep altering with a pathological MCS-like wake).

- However, a third circumstance could be more finely adapted to the situation of patients in transition within the coma recovery (in the ICU setting notably): they would not be appropriately defined by one single state but by a real rhythmic modulation of their state (e.g. such as an MCS patient truly shifting from a cortically-mediated function not expressed by behaviour toward a cortically-mediated function expressed by behaviour and then towards a non-responsive coma).

In order to answer this question, a complete behavioural assessment would be required, possibly even more often than every hour, if consciousness contents fluctuate with high frequency. Moreover, in order to not overlook cognitive motor dissociations, an additional non-behavioural assessment of the consciousness content would also be required.

However, this extremely rigorous protocol presents feasibility issues, in particular if the possibility to assess CMD shall be integrated in the debate.

- Lines 133 et seq.: according to the CRS-R minimal consciousness is also that someone follows an object with his/her eyes. What is the reason for introducing “situations of danger” in this context? Furthermore, I would appreciate a better introduction of the “embodiment of mind” hypothesis if it is really relevant. I do not think most people involved in disorders of consciousness research let alone circadian research are familiar with it. I think these are just two examples where the authors veer off the storyline.

To avoid losing the reader at this step, the issues raised by “responsiveness in ecological conditions” has been more clearly stated.

The concept of embodiment has been gathered in a dedicated paragraph in the Discussion section of the revised version of the manuscript, and this concept is now discussed into more detail to hopefully increase its readability and understanding.

“Interrogating further the harmony between EEG cyclicality, clock-controlled hormonal rhythms (directly driven by the hypothalamic circadian timing system), behaviour (such as the mesencephalic output to the oculomotor nucleus commanding eye-opening), and environment (to assess the reactivity to sensory inputs) may provide a comprehensive understanding of consciousness loss and reappearance, in accordance with a “world-body-brain” holistic perspective³⁰. F. Varela has introduced the “embodiment of mind” hypothesis³⁰ in cognitive neuroscience of consciousness. Instead of regarding the relationship between consciousness and its correlate as a “one-way causal explanation”, he proposed a “two-way account” in which consciousness “emerged from the organisation of complex systems” in interaction with the

environment. Due to this genealogy, the conscious process transcended the divisions between brain, body, and world as it was partly determined by each of them.

From a pragmatic point of view, this theory defines the “readiness for action” from the enaction principle. The neural conscious processes require a high responsiveness to the environment to remain viable over prolonged period. For example, in ecological conditions, integrating inputs from the outside is fundamental to face immediate situations of danger. Considering this integrated physiological issue, enaction explains how both alertness and cognitive ability could be immediately unified³¹.”

- Line 141: note that this is only true during sensitive phases, i.e. during the night (as the cited literature suggests)

This was specified accordingly.

“Third, wakefulness allows to react quickly to environmental constraints/stimuli, this ability being usually associated with the functionality of the ascending arousal network³². Complementary hypothalamic functions however, might be involved in wakefulness modulations: the SCN, which is sensitive to day/night-light/dark long-range fluctuations³³ can also present rapid modulations after short light exposure during sensitive phases in the night³⁴,^{35, 36, 37} but also over 24 hours³⁸.”

- Line 163 et seq.: I have difficulties understanding what the authors wish to say. Please consider revising.

A precision was provided to explicit the raised limitation of the cited work.

“iii) the inappropriate management of the circadian information issued from clock-controlled hormones which was performed at the population rather than individual scale³⁹.”

If yes, please add this in addition to “relatives”.

None of these patients had legal representatives as no patient was under legal protection and the inclusion criteria did not require to have a designated healthcare proxy (“*personne de confiance*” in France) for the patient to be included.

Methods

- *Patient characteristics: as they were probably not normally distributed, please also indicate median and 25%/ 75% quantiles. Average and SD is only informative in case of normally distributed values.*

These descriptive statistics were added in a dedicated table (Supplementary Table 1-B) to avoid overcrowding the main text.

- *Line 644: in addition to the CRS-R sum score at study inclusion, please also give the diagnosis. The sum score is not (necessarily) related to the diagnosis.*

The diagnoses (CRS-R based) are provided in the text (Methods section: mean at the population scale in Supplementary Table 1 and number of patients within every DOC group) and in the Supplementary Table 1 for individual diagnoses (Column “Classification” with all items of the CRS-R scales).

“The final diagnosis retained was Coma for 3 patients (absence of spontaneous eye-opening during the CRS-R assessment but possible eye-opening after stimulation), UWS/VS for 12 patients, and MCS/CMS for 3 patients (without response to simple command).”

- *Line 650: This refers to three additional patients if I am not mistaken? Please do not call it “three assessments” as one can easily think that these are assessments in the sense of a part of a study protocol.*

Indeed, this paragraph refers to 3 additional recordings including 1 new patient and 2 repeated assessments. It was modified to avoid any misinterpretation.

“Three recordings were added for patients at the chronic stage. It included: i) 2 longitudinal cases of acute patients previously recorded who were both alive and not recovered at 6 months: ii) one original patient classified as UWS/VS who was recorded 6.5 years after the injury.”

- *Line 664: after or during the 2-year follow-up? If after, how was this assessed systematically and when?*

We thank you for pointing out this mistake, the sentence was modified and the term “after” was removed (no follow-up was provided after 2 years).

“The primary endpoint was the best functional outcome that could be observed during a 2-year follow-up (recruitment period: January 2014 – October 2017, last follow-up in October 2019).”

- Line 705: how did you perform the sampling? Did you empty the catheter bag every 2 hours and sample from the urine collected during the past two hours? Please include additional details.

The Methods section was completed with more details concerning the sampling method.

“To that end, the urine bag was emptied before each 2-hour sampling, the total urinary output for this 2-hour period was recorded, and a fraction of this miction was collected, numbered, and kept at 4°C until the 12th sampling. Then, the 12 samples were sent at once to the hospital hormone laboratory.”

- Line 789: why was the “circadian” fit limited to one hour? Previous work has suggested that for DOC patients, the dominant rhythmicity can lie outside this range that has been found for healthy individuals. Also, free-running rhythms can sometimes be as long as 25.2 hours.

We limited the range to that used in healthy individuals because we see no convincing evidence in the literature that the endogenous circadian period (Tau) may be outside this range in DOC patients, especially looking at the best available studies in DOC patients:

1- the recent article by Blume C et al. estimated Tau on skin temperature recorded over 7 days, using the Lomb-Scargle approach, in 22 patients studied in entrained conditions (LD cycle)⁴⁰. Although skin temperature is not an unbiased marker of the phase and amplitude of the master circadian clock (modulated by exogenous and endogenous influences), despite the relatively short recording duration (period estimates change with recording duration and period is found stable after 2 weeks of recording [as shown by Kronauer, Klerman, Jewett and Czeisler’s work]), and although patients were in possibly entraining conditions (LD cycle), the circadian periods found are extremely valuable and may be close to what would be found using central/rectal temperature or melatonin profiling over 2 weeks in non-entrained conditions (as in Wright, Gronfier and other’s using forced-desynchrony). Looking at the data available in the first table of the article, we calculated that the 95% confidence interval (95%CI) of Tau on skin periods is 23.40-24.20 on the entire dataset (n=18), which is very close to the 23.50-24.50 range in healthy subjects. Excluding the shortest period (Tau of 19.86) which may be biased given that it is far outside Tau ranges in mammals (we are noting that the melatonin fit in that subject is not significant), we find on this dataset (n=17) that the

95%CI of Tau is 23.80-24.20, which is entirely comprised in the 23.50-24.50 range of healthy subjects.

2- Looking at the other article by Blume C et al., in which the same methods/markers are used, we reach the same conclusion⁴¹. The authors do not provide the values of the skin temp periods, but when we extract them from Figure 1c (using plot digitizer), we find that the 95%CI of Tau on skin periods is 23.90-24.60 on the entire dataset (n=18). Excluding the longest period which may be biased given that it is far outside Tau ranges in mammals (Tau of 26.29), we find that the 95%CI of Tau on skin periods is 23.80-24.40, which is again entirely comprised in the 23.50-24.50 range of healthy subjects. Altogether, we advocate that the 23.5-24.50 range used herein to constrain Tau in our analysis is appropriate.

Also, free-running rhythms can sometimes be as long as 25.2 hours.

We agree that free-running rhythms can sometimes be as long as 25.2 hours (they can be even longer according to Aschoff's and Weaver's work). But such long free-running rhythmicities have been found using "circadian paradigms" and circadian markers either 1- in free-running protocols in healthy participants (allowed to switch lights on/off when desired, known to affect and lengthen Tau - see Czeisler et al.⁴² and Duffy et al.⁴³ for a discussion on this), or 2- in patients with a circadian disorder (free-running/non-24 type, as in blind subjects or sighted free-runners). As discussed above, the range of periods found in DOC patients so far is not in favour of a high probability for long free-running periods in this medical condition.

Admittedly, DOC patients may rather undergo a "free running" protocol than a "constant routine"?

We are not sure what is meant by the reviewer here, but DOC patients placed either in free-running conditions in darkness or constant low light, or in forced-desynchrony conditions, with melatonin or temperature recordings for at least 2 weeks, would beautifully reveal the circadian drive and individual endogenous circadian periods. The classical constant routine condition is indeed not appropriate because it is too short to extract the period with sufficient precision (see response 2 paragraphs above).

- How did you select the EEG features you have computed for your analysis? Have they previously been implicated in the detection of consciousness? It sometimes sounds a bit as if they had been computed because the python package allowed to do so.

We agree with the reviewer that the data driven approach might sound a bit "hypothesis free" but at the start of our work we observed that a lot of metrics (or family of metrics) had been proposed as neural signatures of consciousness in general or of wakefulness changes in particular.

Therefore, a more thorough justification for this choice of metrics has been added in the first paragraph of the Results section:

“We evaluated long-term variables (24-hour distribution, long-term predictability, and circadian/ultradian rhythmicity) of various short-term EEG features, the combination of which led to the analysis of 168 EEG parameters. The EEG features were selected through a hypothesis-free approach on the basis of their complementarity to describe EEG in its various dimensions and on their previous use in DOC patients. The EEG features retained were then grouped as power¹⁰, spectrum^{11, 12, 13, 14, 15}, complexity^{13, 14, 15, 16, 17, 18, 19, 20, 21}, and spatial variability²² (see Fig. 1 for description).”

Due to this initial abundance of metrics, we tried to select non-redundant dimensions of the EEG signal based on a conjunct theoretical approach of time-series description. The spectrum (and power) features have a historical place as a challenged gold standard. The available complexity analyses were numerous in the literature (and even more in Python packages) so the selection presented herein was based on a preliminary analysis (data not shown) that enabled to select complementary measures susceptible to have a synergistic effect. The search for synergy was based on a complexity theoretical approach concerning the brain function, regarded as a non-linear dynamic system (see ⁴⁴ for an overview). It was designed to be complementary to a linear description of the single-channel EEG signal using power and spectrum (which were additionally synthesized by the AR4 model – that was not calculated using Python), and of a minimal integration of the spatial information by the spatial variability. These last metrics (spatial variability per band) were paradoxically the most innovative ones as they were built on an analogy with the Global Field Power metric⁴⁵, were easily computed compared to the more popular connectivity metrics, and appeared eventually as the most informative across all the results.

Throughout the analysis, we managed to keep features from multiple aspects of non-linear system description: i) based on information theory by using an entropy metric based on the temporal organisation of the EEG time-series (i.e. eventually assessed by SVD entropy); ii) based on fractal geometry by using a measure of the dependency of analysis to the window's size (i.e. eventually assessed by DFA); iii) based on chaos theory by using the temporal recurrence of the same event in EEG time-series (i.e. eventually assessed by Determinism). For example, we chose not to select laminarity as an alternative to Determinism, spectral flatness as a frequency-based alternative to entropy assessed in the temporal domain, permutation entropy as a redundant and less variable alternative to SVD-entropy for short-term analysis, means/skewness/kurtosis values of short-term EEG distributions as non-discriminant alternatives to EEG variance.

After this initial search for theoretical complementarity and technical elimination of duplicates within the groups of metrics, we chose to challenge them at once in a common data-driven approach that was not determined by the previous knowledge of EEG signature of

consciousness, assuming that most data in the literature were built on shorter EEG recordings and might therefore be irrelevant to determine the most appropriate metrics which could catch the long-term dynamic.

Concerning long-term variables, the choice of our metrics was also determined by an original search for *ad-hoc* tools describing mathematically the quality and features of the fluctuations we observed visually. That is what determined the use of long-term complexity analysis, which eventually appeared useful in our models. However, the use of distribution variables appeared as self-evident in this matter and has been previously described^{13, 18} and the fit variables (least square regression method) were of course abundantly used in the circadian literature⁴⁶, including applications on EEG metrics²⁴, and previously used by us in pathological circadian disruption by brain injury²³.

- *Did the nursing staff also provide informed consent for being filmed? Please mention.*

As only their hands could be seen on the camera, no informed consent was required. This has been added in the revised version of the manuscript.

“As the faces of the nursing staff were not filmed (the cameras focused on the patients’ faces and, incidentally, on the staff’s hands), informed consent from the staff was waived.”

- *How was an ultradian rhythm defined?*

As stated in the introduction, an ultradian rhythm was defined as an alternative cyclic pattern whose periods are shorter than 24 hours. Practically speaking, we search for a dominant frequency: if it exists (minimum = 20 min), we assessed the sinusoidal fit at the selected frequency, assuming that this dominant frequency could be circadian (peak of the spectrum at 24h) or ultradian (peak of the spectrum function < 24h).

- *Please give credit to the authors of the R packages used by citing the packages in addition to R.*

The citations were completed.

- *I would suggest the authors to visualise their statistical approach. It is currently very hard to understand. I am not sure, but if it makes sense, one could perhaps split Fig. 1 up into two figures.*

As suggested, the figure was divided into Part A for acquisition and Part B for pre-processing in order to increase clarity and readability.

- *If I am not mistaken, information about the luxmeter measurements are missing completely (i.e., device, sampling rate, direction of measurements, etc.)*

The luxmeter details are given in the Methods section and the direction of the device has been added:

“Environmental light and sound intensities were recorded using a dedicated luxmeter (CHY630, IDDM+[®]) and sonometer (SoundTest-Master, configured in dBA, FAST: 125ms, setup level between 30 and 130 dB, Laserliner[®]). Both devices were placed near the head to be as close as possible to the patients’ eyes (pointing to the same direction) and ears.”

- Do the authors know anything about the orientation of patient rooms and whether they had sunlight?

The position of each room in the unit and therefore their related orientation to the sun was not recorded specifically for each patient. The orientations of the rooms in the ICU were as follows: 2 to the east, 5 to the west, 7 to the south, and 6 to the north. Every room receives sunlight as their windows are open to the outdoors or to a large courtyard.

Results

-Line 218: why did the data have to be scored manually when there was a software?

The software was not dedicated to performing an automatic scoring but to write the “manual” scoring into a file using the same timeline.

-Why does 2-hourly sampling prevent the analysis of ultradian rhythms?

This is indeed not true and has been deleted following the suggestion of the first reviewer to simplify the summary of the methods in the Results section. The choice to not analyse ultradian rhythms in this case is now explained in the Methods section.

Methods – Long-term EEG description:

“Rhythmicity variables were calculated for EEG and eye-opening periods (circadian and ultradian) and for hormonal results (circadian only because of the 2-hour sampling rate preventing the assessment of ultradian rhythms for periods below 4 hours) and dichotomised as presence or absence of these rhythms using ad hoc threshold definitions for the coefficient of correlation of fits which were both sufficiently high and validated a visual fit.”

Methods – Urinary measures:

“The 2-hour sampling rate prevented us from assessing ultradian rhythms for periods below 4 hours (according to the Nyquist-Shannon theorem and more conservatively < 8h) which were of the highest importance for comparison with continuous (behavioural and neurophysiological) recordings.”

–Figure 1: Is there an x-axis as suggested by the arrows? If yes, what is the axis exactly? I find it hard to understand when the multiple assessments took place, i.e. the study protocol. I would also suggest to distinguish between raw data acquired and the variables computed.

The figure 1 has been changed to include these remarks (A for raw data and B for variables).

–Figure 2: Axis labels are so small they can only hardly be read or not at all. Even though it might be an issue with the resolution here, it is too small. Referring to the title, please note that the plural of “analysis” is “analyses”, which you may want to use here. Further, please avoid unnecessary abbreviations in the figure legend. Last, I would like to suggest using a colourblind-friendly palette such as the “viridis” palette in R.

The title and the axis labels were corrected according to the reviewer’s suggestions. The colours of the graphs on the left-side have been modified in Excel but as the right-side of the figure has a clear contrast and a limited complexity, it was not changed. For Figure 3, the viridis package was not compatible with Scatter 3D.

–Figure 3: is there an alternative to the black background? Unfortunately, the axes are barely visible and thus the figures are hard to read.

The 3D scatter plots were only used to illustrate the multidimensional clusterisation, for a readable axis, the reader can refer to the 2D box plots. A white front was tested for 3D scatter plots but the result was even less easy to read.

–Figure 5 is again a bit overwhelming I have to admit. Please try to visualise in an intuitive way. Figures should support the readers’ understanding and not a lot of extra effort to understand. What does “chronic” mean when it is about E-MCS yes/no?

A simpler version of this figure is now provided. The details concerning the abnormal presence of CR were transferred in the Supplementary Information section. The “chronic” statement implies that no outcome was available for this patient with an already fixed condition at the date of recording (P4 after 6 years) or was already indicated for the first recording (P12 and P15).

Figure 6: circadian/ultradian figures are impossible to read. Please adopt appropriate font sizes for axis labels. Also, what does the “bis” in the title mean.

The title was corrected and the “bis” (second recording for one patient) does no longer appeared in the manuscript.

The figure resolution was increased using JPG and we have uploaded each figure alone to avoid changes in resolution due to their inclusion in Word.

The size of the labels cannot be easily modified at this step but the figure legend and the figure format were corrected to state that the ordinate always corresponds to the “EEG features” indicated on the raw graphs and that abscises always correspond to the “time elapsed since recording”.

–Table 1: please harmonise font sizes. Also, please provide a table caption.

The format corrections were made accordingly.

–For all figures and tables, I encourage the authors to include them and their captions in the text no matter what the formatting guidelines say (personally, I always state in the cover letter that I will leave it like this for the reviewing process). You can still reformat later if at all requested (I have informed the editorial office that it is just super annoying like this). I admit that this may partly also have added to the difficulties I had to understand the figures – sorry!

We personally agree with this opinion, but decided to follow the editorial rules until they change.

Discussion

–10.5 pages of discussion (excluding the supplemental discussion) are too long. Importantly, this is not because I do not like to read so much, but because the main story line gets lost. Eventually, it will lead to the paper not being read/cited as much as it could, because readers do not understand its importance. I would therefore suggest to focus and address the 3-4 most important findings and discuss them in a concise manner.

As suggested, part of the discussion has been moved from the main document to the Supplementary Information section. The introduction and the discussion have been rewritten/reorganized to focus on the most relevant messages and hopefully improve the readability of the paper.

–Line 636: if the authors wish to announce with this paragraph what they will do next, I recommend omitting such “meta”-phrases and simply continue.

This paragraph was removed.

–Line 410: what is the group level? To which groups do you refer?

This occurrence of group-level analysis (e.g. in Fig 3 and 4) is just used by opposition to the individual-level analysis (e.g. in Fig. 5). It was not specific for any group of patients defined by behavioural features or outcome.

–Line 529: *I am sorry, I am a bit lost by now, but what EEG pattern do you refer to? Perhaps you can direct the reader to the relevant figure?*

This paragraph was corrected as follows to be more accurate:

“The shape of the EEG pattern expected from the integrated results (synthesised in Fig. 3 and 4 and illustrated in Fig. 6 and Supplementary Figures 1-2-3-4) might be reminiscent of the Flip-Flop balance model regulating sleep-wake cycles, as previously described²⁹.”

1. Bayne T, Hohwy J, Owen AM. Are There Levels of Consciousness? *Trends Cogn Sci* **20**, 405-413 (2016).
2. Laureys S. The neural correlate of (un)awareness: lessons from the vegetative state. *Trends in Cognitive Sciences* **9**, 556-559 (2005).
3. Klosch G, Zeitlhofer J, Ipsiroglu O. Revisiting the Concept of Vigilance. *Front Psychiatry* **13**, 874757 (2022).
4. van Schie MKM, Lammers GJ, Fronczek R, Middelkoop HAM, van Dijk JG. Vigilance: discussion of related concepts and proposal for a definition. *Sleep Med* **83**, 175-181 (2021).
5. Oken BS, Salinsky MC, Elsas SM. Vigilance, alertness, or sustained attention: physiological basis and measurement. *Clin Neurophysiol* **117**, 1885-1901 (2006).
6. PVS TM-STFo. Medical aspects of the persistent vegetative state (1). *The New England journal of medicine* **330**, 1499-1508 (1994).
7. Wielek T, *et al.* Sleep in patients with disorders of consciousness characterized by means of machine learning. *PLoS One* **13**, e0190458 (2018).
8. Piarulli A, Bergamasco M, Thibaut A, Cologan V, Gosseries O, Laureys S. EEG ultradian rhythmicity differences in disorders of consciousness during wakefulness. *J Neurol* **263**, 1746-1760 (2016).
9. Kleitman N. Basic rest-activity cycle--22 years later. *Sleep* **5**, 311-317 (1982).
10. Rossi Sebastiano D, *et al.* Significance of multiple neurophysiological measures in patients with chronic disorders of consciousness. *Clin Neurophysiol*, (2014).

11. Gosseries O, *et al.* Automated EEG entropy measurements in coma, vegetative state/unresponsive wakefulness syndrome and minimally conscious state. *Funct Neurol* **26**, 25-30 (2011).
12. Schiff ND, Nauvel T, Victor JD. Large-scale brain dynamics in disorders of consciousness. *Curr Opin Neurobiol* **25**, 7-14 (2014).
13. Sitt JD, *et al.* Large scale screening of neural signatures of consciousness in patients in a vegetative or minimally conscious state. *Brain* **137**, 2258-2270 (2014).
14. Stefan S, *et al.* Consciousness Indexing and Outcome Prediction with Resting-State EEG in Severe Disorders of Consciousness. *Brain Topogr* **31**, 848-862 (2018).
15. Ballanti S, *et al.* EEG-based methods for recovery prognosis of patients with disorders of consciousness: A systematic review. *Clin Neurophysiol* **144**, 98-114 (2022).
16. Valenza G, *et al.* EEG complexity drug-induced changes in disorders of consciousness: a preliminary report. *Conf Proc IEEE Eng Med Biol Soc* **2011**, 3724-3727 (2011).
17. Frohlich J, Toker D, Monti MM. Consciousness among delta waves: a paradox? *Brain* **144**, 2257-2277 (2021).
18. Mesin L, Costa P. Prognostic value of EEG indexes for the Glasgow outcome scale of comatose patients in the acute phase. *J Clin Monit Comput* **28**, 377-385 (2014).
19. Thul A, *et al.* EEG entropy measures indicate decrease of cortical information processing in Disorders of Consciousness. *Clinical Neurophysiology* **127**, 1419-1427 (2016).
20. Thiery T, *et al.* Long-range temporal correlations in the brain distinguish conscious wakefulness from induced unconsciousness. *Neuroimage* **179**, 30-39 (2018).
21. Mateos DM, Guevara Erra R, Wennberg R, Perez Velazquez JL. Measures of entropy and complexity in altered states of consciousness. *Cogn Neurodyn* **12**, 73-84 (2018).
22. Curley WH, *et al.* Electrophysiological correlates of thalamocortical function in acute severe traumatic brain injury. *Cortex* **152**, 136-152 (2022).
23. Gobert F, *et al.* Is circadian rhythmicity a prerequisite to coma recovery? Circadian recovery concomitant to cognitive improvement in two comatose patients. *J Pineal Res* **66**, e12555 (2019).
24. Aeschbach D, Matthews JR, Postolache TT, Jackson MA, Giesen HA, Wehr TA. Two circadian rhythms in the human electroencephalogram during wakefulness. *Am J Physiol* **277**, R1771-1779 (1999).
25. Cajochen C, Wyatt JK, Czeisler CA, Dijk DJ. Separation of circadian and wake duration-dependent modulation of EEG activation during wakefulness. *Neuroscience* **114**, 1047-1060 (2002).

26. Wasserstein RL, Schirm AL, Lazar NA. Moving to a World Beyond “ $p < 0.05$ ”. *The American Statistician* **73**, 1-19 (2019).
27. Naccache L. Minimally conscious state or cortically mediated state? *Brain*, (2017).
28. Wannez S, Heine L, Thonnard M, Gosseries O, Laureys S, Coma Science Group c. The repetition of behavioral assessments in diagnosis of disorders of consciousness. *Ann Neurol* **81**, 883-889 (2017).
29. Saper CB, Scammell TE, Lu J. Hypothalamic regulation of sleep and circadian rhythms. *Nature* **437**, 1257-1263 (2005).
30. Thompson E, Varela FJ. Radical embodiment: neural dynamics and consciousness. *Trends Cogn Sci* **5**, 418-425 (2001).
31. Ceruti M, Damiano L. Plural Embodiment(s) of Mind. Genealogy and Guidelines for a Radically Embodied Approach to Mind and Consciousness. *Front Psychol* **9**, (2018).
32. Edlow BL, Claassen J, Schiff ND, Greer DM. Recovery from disorders of consciousness: mechanisms, prognosis and emerging therapies. *Nat Rev Neurol*, (2020).
33. Saper CB, Lu J, Chou TC, Gooley J. The hypothalamic integrator for circadian rhythms. *Trends Neurosci* **28**, 152-157 (2005).
34. Perrin F, *et al.* Nonvisual responses to light exposure in the human brain during the circadian night. *Current biology : CB* **14**, 1842-1846 (2004).
35. Prayag AS, Munch M, Aeschbach D, Chellappa SL, Gronfier C. Light Modulation of Human Clocks, Wake, and Sleep. *Clocks Sleep* **1**, 193-208 (2019).
36. Rahman SA, Wright KP, Jr., Lockley SW, Czeisler CA, Gronfier C. Characterizing the temporal Dynamics of Melatonin and Cortisol Changes in Response to Nocturnal Light Exposure. *Scientific reports* **9**, 19720 (2019).
37. Prayag AS, Jost S, Avouac P, Dumortier D, Gronfier C. Dynamics of Non-visual Responses in Humans: As Fast as Lightning? *Front Neurosci* **13**, 126 (2019).
38. Khalsa SBS, Jewett ME, Cajochen C, Czeisler CA. A Phase Response Curve to Single Bright Light Pulses in Human Subjects. *The Journal of Physiology* **549**, 945-952 (2003).
39. Yang X-a, *et al.* Prognostic roles of sleep electroencephalography pattern and circadian rhythm biomarkers in the recovery of consciousness in patients with coma: a prospective cohort study. *Sleep Medicine*, (2020).
40. Blume C, *et al.* Healthier rhythm, healthier brain? Integrity of circadian melatonin and temperature rhythms relates to the clinical state of brain-injured patients. *Eur J Neurol* **26**, 1051-1059 (2019).

41. Blume C, *et al.* Significance of circadian rhythms in severely brain-injured patients: A clue to consciousness? *Neurology* **88**, 1933-1941 (2017).
42. Czeisler CA, *et al.* Stability, precision, and near-24-hour period of the human circadian pacemaker. *Science* **284**, 2177-2181 (1999).
43. Duffy JF, *et al.* Sex difference in the near-24-hour intrinsic period of the human circadian timing system. *Proc Natl Acad Sci U S A* **108 Suppl 3**, 15602-15608 (2011).
44. Stam CJ. Nonlinear dynamical analysis of EEG and MEG: review of an emerging field. *Clin Neurophysiol* **116**, 2266-2301 (2005).
45. Skrandies W. Global field power and topographic similarity. *Brain Topogr* **3**, 137-141 (1990).
46. Gronfier C, Wright KP, Jr., Kronauer RE, Czeisler CA. Entrainment of the human circadian pacemaker to longer-than-24-h days. *Proc Natl Acad Sci U S A* **104**, 9081-9086 (2007).

Reviewers' comments:

Reviewer #2 (Remarks to the Author):

Gobert et al.: Recovered predictable fluctuations and 24-hour rhythmicities in coma are associated with a favourable outcome

I thank the authors for their work on the manuscript, which has greatly improved. However, there are still some points that should to be addressed in my opinion. Overall, the manuscript is still rather challenging to read and follow.

Abstract

- I am still unsure whether "long-term modulations" in consciousness is the right term to use here. I have read the manuscript quite a while ago and have not remembered my comments in detail. I just caught myself stumbling across this term again. I am not convinced diurnal alterations between sleep and wakefulness are "long-term". For me, "long-term" is at least a scale of 2-3 weeks, but not 24 hours.

- Why not just say "from urine", in my opinion it does not need the word "excretion" at all?

Introduction

- Note that while circadian rhythms are genetically determined, the resulting phenotype always results from an interaction between biology and behaviour (e.g., light exposure). To me, it currently sounds almost a bit too hard-wired.

- I think I know what you mean by "The static description of wakefulness and awareness states...". However, I still find that part very complicated to read and hard to understand.

Figures

- I am afraid the font size is sometimes still extremely small. I would recommend to scale the picture on the screen to A4 and check if the font is readable.

Discussion

- "peculiar": do you maybe rather mean "particular"?

- You write "The present study demonstrates that multimodal 24-hour long assessments allowed a better prediction of long-term evolution" – what time scale are you referring to here when you write "long-term", probably not 24h? This is exactly what I mean: using the same term in different contexts in a manuscript will be misleading.

Reviewer #3 (Remarks to the Author):

I would like to thank the authors for carefully addressing the comments raised in the first round of revisions. Dr Florent Gobert and colleagues provide a very thorough presentation of their data and analysis, that is rather rare to come across. The introduction and discussion are easier to digest in their reshaped format. However, the results are still not easy to follow. The interpretation of the different measures, their results and relationship amongst each other requires a lot of work for the reader. It would really help if the authors would help the reader with more basic interpretation of the data instead of/alongside the high level (almost philosophical) interpretation that is provided currently.

1) While the paper has been professionally edited for English language, I do find that still the authors are frequently rather a-specific about their rationale and analysis. In order to get a full understanding of the extensive amount of work performed in this study, it requires a lot of back and forth between the different sections of the paper. It is a pity, because this will likely negatively influence the impact

of this work. Some examples (and more specific concerns are raised from point 4 onwards):

"The mean value of short-term complexity (with every features) was not discriminant but the circadian cyclicity of Determinism and Detrended Fluctuation Analysis was used as reference to define the reference circadian rhythms of brain function(EEG parameters selected from healthy subjects)." → After reading multiple times, it is still not clear what the authors did.

"Circadian rhythmicity for eye-opening/closing (Fig. 4-A and Table 1-B, with 8 occurrences of "behavioural circadian rhythmicity" vs 13 without), was associated with 3 EEG parameters (AIC = 8; p = 0.00432; FPR = 0.0411; LR =23.33)." → what are the 3 EEG parameters? In the beginning of the results section, 168 parameters are mentioned that are grouped in " power20, spectrum19, 21, 22, 23, 24, complexity19, 23, 24, 25, 26, 27, 28, 29, 30, and spatial variability31", so which 3 EEG parameters are used here?

In the images, the authors could directly provide information about the patient presented in the figures (or at least in the figure caption). In the current form the authors refer to a patient number, then the reader has to check supplementary table 1 to understand what (s)he is looking at.

2) Supplementary table 1 contains a mistake (and this might also influence the results regarding the predictability of outcome). Patient 5 has an outcome of 6 according to GOS while according to the CRS-R the patient died

3) The authors clarified that the values in Table 1 are absolute values, but nonetheless these very high values need a unit indication for clarification e.g., assuming it is in microvolts, 23 385 690,8 seems very high.

4) The authors have illustrated clearly why it has been difficult to build a standard hypnogram from the data, and we fully acknowledge that this is extremely challenging. Yet, what is missing in the following is the argument why, when the clearest circadian rhythm, the sleep-wake cycle, is impossible to establish, other rhythmic variables are more reliable. Perhaps the authors could clarify

5) Related to this, the relationship among the outcome measures is unclear. For example on line 239, the statistics for EEG correlates are of the final "functional outcome" are reported as (AIC = 8; p = 0.00422; FPR = 0.0451; LR = 21.18), while below the statistics for EEG correlates of a "behavioral circadian rhythmicity" are reported on line 254 as (AIC = 8; p = 0.00432; FPR = 0.0411; LR =23.33). Their similarity, given the same EEG predictors may suggest highly correlated outcome measures.

6) Point 5 and point 1 together raise new questions on the EEG variables used throughout the study. What specific predictors were significantly related to the outcome measure, how do they relate to each other and how do these in turn relate between different outcomes? As an example, take the statistics from point 5 and assume these outcome measures are highly correlated, how would you then explain finding different EEG correlates (e.g., if functional outcome and behavioral circadian rhythmicity are highly correlated, you may expect the same EEG correlates to predict them)? At the same time, what does it mean to find different EEG correlates within one outcome analysis, e.g., for "functional outcome", what does it mean that you have both a "higher magnitude of fluctuations" and a "higher long-term predictability" (line 240) as they could seem mutually exclusive.

7) The focus on "predictability" as a concept needs further refinement

a) Predictability seems in the end defined by entropy, with lower entropy meaning higher predictability. This inverse interpretation (the authors find lower entropy but write higher

predictability) is sometimes confusing.

b) With its extensive presence, it needs earlier explanation in the manuscript, e.g., predictability can also mean significant predictive statistical models used (for a lot of readers possibly the more straightforward interpretation).

c) The relationship between predictability and rhythmicity is unclear, e.g., a tight 30-minute ultradian rhythm can be as predictable as a circadian rhythm.

d) The predictability concept returns in the discussion on fascinating theoretical work by Varela. While in general, the manuscript's valuable empirical contributions first need further grounding/interpretability before transcending into more philosophical debates on its meaning, we fear the conclusions here are a bit too forward. The main concerns are that predictability, a main finding of the manuscript, in its maximized form also means a completely static system, in which no "readiness for action" can be achieved. In more general terms, it is not clear if predictability has anything to do with the theoretical concepts outlined here. As a side note, total "readiness for action" differences between ultradian and circadian oscillating systems might not be as straightforward as well.

8) One of the main results "enabled to identify a small group of patients who systematically regained consciousness" (line 340 in the discussion) needs additional context. This was concluded based on a 100% PPV, meaning that there are only True Positives and no False Positives. At the same time, sensitivity was low, likely the result of False Negatives. From an ethical, patient-care, perspective, this does not seem like a good result and should be interpreted with more care. The main consideration here is that a False Negative can lead to unwarranted conclusions, with potentially large implications when considering end-of-life decisions, while the implication of a False Positive is much less severe. Thus, ethical discussion on the usefulness and implication of this finding is in order.

Reviewers' comments:

Reviewer #2 (Remarks to the Author):

Gobert et al.: Recovered predictable fluctuations and 24-hour rhythmicities in coma are associated with a favourable outcome

I thank the authors for their work on the manuscript, which has greatly improved. However, there are still some points that should to be addressed in my opinion. Overall, the manuscript is still rather challenging to read and follow.

Abstract

- I am still unsure whether “long-term modulations” in consciousness is the right term to use here. I have read the manuscript quite a while ago and have not remembered my comments in detail. I just caught myself stumbling across this term again. I am not convinced diurnal alterations between sleep and wakefulness are “long-term”. For me, “long-term” is at least a scale of 2-3 weeks, but not 24 hours.

We agree that the word “long-term” can be used, according to the context, for different durations. Indeed, for neuronal firing and most brain processes, diurnal fluctuations are long-term changes, but they would be considered short-term fluctuations for infradian/seasonal rhythmicities. However, since the manuscript is now built around the dialectical opposition of short/long-term, we have chosen to retain the wording, to reinforce this definition in the text and to suppress the use of other occurrences of “long-term” that refer to a different context.

Consequently, this has been clarified in the Introduction section, as follows:

“long-term fluctuations (defined in this setting as being measurable at the scale of a 24-hour EEG).”

Additionally, each occurrence of “long-term” is now specified as “24h long-term” when referring to this specific context.

- Why not just say “from urine”, in my opinion it does not need the word “excretion” at all?

The abstract has been modified accordingly.

“biological (clock-controlled melatonin and cortisol secretion from urine)”

Introduction

- Note that while circadian rhythms are genetically determined, the resulting phenotype always results from an interaction between biology and behaviour (e.g., light exposure). To me, it currently sounds almost a bit too hard-wired.

We have added a sentence to this effect that reads at the end of the Introduction section:

“Overall, wakefulness and sleep result from the interplay between internal processes and external influences that lead to EEG fluctuations at different scales.”

- *I think I know what you mean by “The static description of wakefulness and awareness states...”. However, I still find that part very complicated to read and hard to understand.*

This paragraph of the Introduction section has been modified to hopefully increase readability.

It now reads: “This static description of consciousness may seem operational for healthy participants, but research in comatose and post-comatose patients shows that the assessment of consciousness modulations is dependent on the preservation of three complementary dynamic features: i) the access to complex cognitive functions¹, ii) the quality of arousal reactivity (defined as “an upward change in the level of alertness²”), and iii) the physiological rhythmicity of wakefulness during the sleep-wake cycle as our group has recently shown³. In physiology, the transient disappearance of consciousness during sleep⁴ is in phase with the environment and is confirmed by the eye-opening behaviour, assuming that both dimensions vary linearly (for non-Rapid Eye Movement [NREM] sleep to aware wakefulness, but not REM sleep during which, by definition, modulations differ paradoxically).”

Figures

- *I am afraid the font size is sometimes still extremely small. I would recommend to scale the picture on the screen to A4 and check if the font is readable.*

The font size has been corrected in every place of the main figures when it was possible and relevant for visualisation issues.

Discussion

- *“peculiar”*: do you maybe rather mean “particular”?

The use of the word peculiar was changed to particular in the context of the new proposed marker and it now reads:

“In DOC patients, the respective roles of ultradian and circadian rhythms remain unclear because, beyond the particular circumstance of a “homogeneous presence of all circadian rhythms” (considering Analysis N°2) among few acute patients, the unpredictable changes of wakefulness were not modelled by circadian rhythmicities.”

However, when referring to the “dimension of a basic brain functionality”, we chose the word peculiar to evoke that this dimension is a singular property of the brain.

- You write “The present study demonstrates that multimodal 24-hour long assessments allowed a better prediction of long-term evolution” – what time scale are you referring to here when you write “long-term”, probably not 24h? This is exactly what I mean: using the same term in different contexts in a manuscript will be misleading.

We agree, and as discussed in response to the first comment, the differences in “long-term” occurrences have been modified to avoid confusion by using “late evolution” in this specific case and by using “24h long-term” when referring to our fluctuation assessment.

It now reads:

“The present study demonstrates that multimodal 24-hour long assessments allowed a better prediction of late evolution.”

Reviewer #3 (Remarks to the Author):

I would like to thank the authors for carefully addressing the comments raised in the first round of revisions. Dr Florent Gobert and colleagues provide a very thorough presentation of their data and analysis, that is rather rare to come across. The introduction and discussion are easier to digest in their reshaped format. However, the results are still not easy to follow. The interpretation of the different measures, their results and relationship amongst each other requires a lot of work for the reader.

In order to increase readability, we have identified the 2 distinct analyses that we performed as “Analysis N°1” and “Analysis N°2” in the revised version of the manuscript (Abstract, methods summary at the end of the Introduction, Results, Discussion, and Complete methods + Fig 1). Analysis N°1 corresponds to the **Supervised data-driven**, as an interpretation of EEG alone. Analysis N°2 corresponds to the **Multimodal rhythmic analysis**, as an interpretation of EEG/Behaviour/Biology from a circadian point of view only.

It would really help if the authors would help the reader with more basic interpretation of the data instead of/alongside the high level (almost philosophical) interpretation that is provided currently.

The discussion has been restructured but its content has not been significantly modified as all secondary questions were already removed in the Supplementary discussion following the first round of reviewing. We consider every part of the remaining discussion as crucial for the appropriate understanding of the results.

However, to separate the rather descriptive part of the discussion from a more interpretative part, the “Theoretical perspectives” paragraph has been re-written to now include two sub-chapters explicitly related to the results from Analysis N°1 and Analysis N°2:

“i) Justification for assessing local and global states of consciousness considering the complementarity of short and long-term EEG metrics in Analysis N°1; and ii) Implications for the consciousness embodiment hypothesis considering Analysis N°2.”

1) While the paper has been professionally edited for English language, I do find that still the authors are frequently rather a-specific about their rationale and analysis. In order to get a full understanding of the extensive amount of work performed in this study, it requires a lot of back and forth between the different sections of the paper. It is a pity, because this will likely negatively influence the impact of this work. Some examples (and more specific concerns are raised from point 4 onwards):

We thank the reviewer for this constructive remark. We have modified the manuscript keeping in mind this comment and the others, to hopefully increase readability. Please find our specific answers below.

“The mean value of short-term complexity (with every features) was not discriminant but the circadian cyclicity of Determinism and Detrended Fluctuation Analysis was used as reference to define the reference circadian rhythms of brain function (EEG parameters selected from healthy subjects).” → After reading multiple times, it is still not clear what the authors did.

This paragraph has been modified to avoid confusion: the point of this sentence was to recall, in the Discussion section, the way in which Analysis N°2 was built, based on the assumption that “normative” values (from healthy subjects) were required to do so.

It now reads:

“Based on Analysis N°1, the most discriminant EEG features describing fluctuations at a short-term scale (accounting for the thalamo-cortical wakefulness effector⁵) were related to spectral analysis, spatial variability, and EEG complexity. In particular, Determinism is a feature assessing the predictability of EEG times series. It is, by definition, inversely related to complexity⁶. The mean value of short-term complexity (with every feature) was not discriminant. Conversely, the circadian cyclicity of two features assessing short-term complexity (Determinism and Detrended Fluctuation Analysis) was pertinent for Analysis N°2: the existence of circadian cyclicity for both was considered as a reference to define brain function circadian rhythmicity (as they were included in the 4 EEG parameters selected from healthy subjects).”

“Circadian rhythmicity for eye-opening/closing (Fig. 4-A and Table 1-B, with 8 occurrences of “behavioural circadian rhythmicity” vs 13 without), was associated with 3 EEG parameters (AIC = 8; p = 0.00432; FPR = 0.0411; LR =23.33).” → what are the 3 EEG parameters? In the beginning of the results section, 168 parameters are mentioned that are grouped in “ power 20, spectrum19, 21, 22, 23, 24, complexity19, 23, 24, 25, 26, 27, 28, 29, 30, and spatial variability31”, so which 3 EEG parameters are used here?

Since the EEG parameters were specific to each factor, to avoid confusion during the reading of the manuscript we had initially decided to describe all these parameters in a table (Table 1). These are also described to a lesser extent in Figures 3 and 4. Then to avoid redundancy, we had also decided to use a merely interpretative description of the parameter’s names in the text. In the revised version of the manuscript, we now refer to Table 1 each time we mention the parameters associated with each factor.

It now reads for this example:

“Circadian rhythmicity for eye-opening/closing (Fig. 4-A and Table 1-B, with 8 occurrences of “behavioural circadian rhythmicity” vs 13 without), was associated with 3 EEG parameters (AIC = 8; p = 0.00432; FPR = 0.0411; LR =23.33). These 3 parameters (detailed in

Table 1) indicated a lower long-term predictability (for spatial variability), a higher magnitude of fluctuation within the 24-hour distribution and a shorter period of rhythmicity (for spectrum; Table 1-C). Notably, the group of patients without eye-opening circadian rhythmicity seemed heterogeneous as their 3-dimensional cluster was wide because of a higher dispersion regarding the Dominant Period of Delta Relative Power.”

As described in Table 1, the 3 EEG parameters used here were (for this example only):

- Standard Deviation for AR4 Total
- Permutation Entropy for Theta Spatial Variability
- Dominant Period for Delta Relative Power

In the images, the authors could directly provide information about the patient presented in the figures (or at least in the figure caption). In the current form the authors refer to a patient number, then the reader has to check supplementary table 1 to understand what (s)he is looking at.

As suggested, Part A of the Supplementary Table 1 was inserted as Part B into a revised version of Figure 5 and the figure/supplementary table captions were modified accordingly. In addition, a short paragraph of clinical description has been added in the caption of illustrative figures (Figure 6 and Supplementary figures 2-4).

For example, it now reads:

“Figure 6: Illustration N°1 with Patients N°1 (A) and N°10 (B) from Figure 5

These patients presented a complete opposition of their rhythmic patterns with “homogeneous presence of all circadian rhythms” (A) and a “homogeneous absence of all circadian rhythms” (B)

Patient 1 was a 58 y-o male presenting a haemorrhagic stroke. GCS at admission was 6 without pupilar abnormalities. At the date of evaluation, the patient remained UWS/VS (with a CRS-R = 5 and GCS = 9). The neurophysiological battery was favourable, and the patient evolved towards a favourable awakening outcome despite a persisting severe disability.

Patient 10 was a 21 y-o male presenting a severe TBI. GCS at admission was 6 with a unilateral mydriasis. At the date of evaluation, the patient remained comatose (with a CRS-R = 3 and GCS = 5). The neurophysiological battery was not favourable excepted EEG reactivity but the patient evolved towards a favourable awakening outcome despite a persisting severe disability.

Raw data (continuous curves with uniformed scales across each illustration on the left) and circadian fits (plots with ad hoc scales on the right) are presented for behavioural data (eye-opening moments in black), environmental data (luxmeter in yellow), clock-controlled hormones (blue for melatonin and orange for cortisol with raw values for the continuous curves and log transformed values for the plots), and EEG with a common timescale in abscise (24h). The 4 EEG features (red: Determinism, blue: Detrended Fluctuation Analysis, violet: Alpha Spatial Variability, green: Beta Spatial Variability) are those presenting a significant circadian rhythm among healthy participants. EEG is defined as “circadian” when more than one feature presented a circadian fit. Plots with significant circadian fits are indicated by red frames.”

2) *Supplementary table 1 contains a mistake (and this might also influence the results regarding the predictability of outcome). Patient 5 has an outcome of 6 according to GOS while according to the CRS-R the patient died.*

We would like to thank the reviewer for pointing out this editing mistake which thankfully has no consequence on the above-mentioned results, which were based on the last column (Best GOS during 2 years) of the original supplementary Table 1 but which have now been moved to Figure 5; there was no mistake in that column. To be more precise, the GOS for that patient was 6 and the CRS-R was “na” not for death but due to a return to a foreign country (the “na: death” was accidentally entered by a spreadsheet autofill function).

3) *The authors clarified that the values in Table 1 are absolute values, but nonetheless these very high values need a unit indication for clarification e.g., assuming it is in microvolts, 23 385 690,8 seems very high.*

As a power value, the unit is in μV^2 , which can explain why the values appear so high. The table has been modified accordingly to indicate the unit, where applicable.

It now reads:

“The units are given in the table (hours, μV^2) when available (except for correlations, relative power ratio and derivatives, complexity measures).”

4) *The authors have illustrated clearly why it has been difficult to build a standard hypnogram from the data, and we fully acknowledge that this is extremely challenging. Yet, what is missing in the following is the argument why, when the clearest circadian rhythm, the sleep-wake cycle, is impossible to establish, other rhythmic variables are more reliable. Perhaps the authors could clarify*

Indeed, contrary to previous studies in the chronic⁷ setting, we failed to build such a classical and synthetic view in a way that remained consistent for all patients in this acute setting. Applying the general rules built on brain function from healthy, or at least conscious, individuals was not possible here due to the heterogeneity of their atypical slow background EEG rhythm. We found that adapting these rules very specifically to each patient would have introduced an overfitting bias and a lack of generalisability of our results, as it has been previously observed in the literature in the acute⁸ setting, where complete hypnograms could not be performed.

Therefore, the incipit of the Discussion section has been modified to allow a more general discussion of this issue. It now reads:

“Contrary to previous studies that have focused on sleep-wake classification to delineate the wakefulness changes during the nycthemeral period in chronic DOC⁷, we were unable to build such a classical and synthetic view in a way that remained consistent at the population level. In contrast, choosing a mathematical fluctuation description allowed us to explicit a general rule. In addition, using linear descriptions of EEG features in Analysis N°1 enabled to

avoid defining sharp thresholds of classification based on common⁹ or specific⁷ minimal requirements on 30-sec EEG epochs. In other words, we assumed a fine-grained mathematisation of brain function fluctuations – revealing its real shape and complexity – instead of conforming it to previous arbitrary ordinal categorisation without nuance.”

5) Related to this, the relationship among the outcome measures is unclear. For example on line 239, the statistics for EEG correlates are of the final “functional outcome” are reported as (AIC = 8; $p = 0.00422$; FPR = 0.0451; LR = 21.18), while below the statistics for EEG correlates of a “behavioral circadian rhythmicity” are reported on line 254 as (AIC = 8; $p = 0.00432$; FPR = 0.0411; LR = 23.33). Their similarity, given the same EEG predictors may suggest highly correlated outcome measures.

We agree that this would have been the case if the EEG parameters related to each factor were the same. However, as stated in our reply to Point 1), this is not the case as the EEG parameters differed for each factor.

6) Point 5 and point 1 together raise new questions on the EEG variables used throughout the study. What specific predictors were significantly related to the outcome measure, how do they relate to each other and how do these in turn relate between different outcomes? As an example, take the statistics from point 5 and assume these outcome measures are highly correlated, how would you then explain finding different EEG correlates (e.g., if functional outcome and behavioural circadian rhythmicity are highly correlated, you may expect the same EEG correlates to predict them)?

Since the EEG parameters differed according to each factor studied, evaluating whether these factors were correlated was not possible in this way. However, such correlations cannot be ruled out, but can indirectly be inferred based on Analysis N°2 (see Fig.5). For instance:

- For the factor “behavioural circadian rhythmicity”: no relation to outcome when present alone (orange line)
- For the factor “hormonal circadian rhythmicity”: no relation to outcome when present alone (yellow line)
- For the factor “behavioural cortical function”: the relationship with outcome is well described in the literature¹⁰ and was confirmed in our setting (Supplementary Table 3).

At the same time, what does it mean to find different EEG correlates within one outcome analysis, e.g., for “functional outcome”, what does it mean that you have both a “higher magnitude of fluctuations” and a “higher long-term predictability” (line 240) as they could seem mutually exclusive.

On the contrary: having several variables helps describe more precisely the shape of the fluctuations, that can have both a high magnitude AND a high predictability, as illustrated below.

7) The focus on “predictability” as a concept needs further refinement

a) Predictability seems in the end defined by entropy, with lower entropy meaning higher predictability. This inverse interpretation (the authors find lower entropy but write higher predictability) is sometimes confusing.

The variables used for 24h long-term EEG description included indeed, by definition, the term entropy (SVD and permutation entropy, which are based on Shannon’s theoretical proposal relating complexity to the quantity of information provided by any signal or time series¹¹). Entropy metrics are extensively used in neurophysiology to evaluate EEG changes after anaesthesia¹², DOC¹³, or neurological¹⁴/neurosensory¹⁵ diseases. Moreover, in the literature, the term predictability is used as an inverse interpretation of complexity, which is measured by entropy metrics (see the recent review by Lau et al.¹⁶).

In our manuscript, with the aim to increase clarity and readability, we chose to qualify fluctuations of high/low entropy by the inverse interpretation, i.e. low/high predictability. However, when dealing with SVD and permutation entropy variables, the canonical name entropy was kept.

b) With its extensive presence, it needs earlier explanation in the manuscript, e.g., predictability can also mean significant predictive statistical models used (for a lot of readers possibly the more straightforward interpretation).

Thank you for this suggestion. Accordingly, the relationship between predictability and entropy has been introduced earlier in the revised version of the manuscript:

- in the abstract:
“long-term predictability (related to a low entropy of fluctuations)”
- in the final paragraph of the Introduction section:
“increased predictability of brain activity (measured by low values of entropy variables,

evaluating fluctuations reproducibility across time)”

Concerning the statistical models, in order to avoid confusion, we had previously chosen the dedicated expression “predictive value” rather than “predictability”.

c) The relationship between predictability and rhythmicity is unclear, e.g., a tight 30-minute ultradian rhythm can be as predictable as a circadian rhythm.

We agree that the relationship between predictability and rhythmicity is not trivial. To avoid confusion, the conclusions about the role of predictability (referring to low entropy herein) and about the homogeneity in circadian rhythms are now described and discussed separately in Analysis N°1 and N°2, respectively. Then, to make their relationship clearer in the manuscript as well, a complementary paragraph has been added in the Supplementary Discussion section.

It now reads:

Theoretical relationship between predictability and rhythmicity (considering both Analysis N°1 and Analysis N°2)

In our work, the relationship between predictability and rhythmicity was found incidentally but was not hypothesised a priori. It is not trivial to explain: to have both a high ultradian fit and a high predictability, the *24h long-term fluctuation pattern* has to be regular and sinusoidal enough to correlate with the expected theoretical sinusoidal shape at the considered period, which is specific to each patient. For this reason, the circadian rhythm (which represents the basis of Analysis N°2 but for which a single variable is studied in Analysis N°1) cannot be qualified as predictable at the 24h scale since no recurrence is allowed at this measurement scale for this fluctuation period. To evaluate whether circadian rhythms are predictable, the measurement scale would need to be of several days or weeks.

More theoretically, predictability and rhythmicity can be regarded as related to the two complementary ways of describing EEG complexity proposed by Lau et al.: rhythmicity stands as a particular sinusoidal manner of assessing regularity (defined as “the general amount of repetitions of patterns in the system’s trajectory”) while predictability is defined as a non-linear “temporal evolution of the system states”¹⁶.

d) The predictability concept returns in the discussion on fascinating theoretical work by Varela. While in general, the manuscript's valuable empirical contributions first need further grounding/interpretability before transcending into more philosophical debates on its

meaning, we fear the conclusions here are a bit too forward. The main concerns are that predictability, a main finding of the manuscript, in its maximized form also means a completely static system, in which no “readiness for action” can be achieved. In more general terms, it is not clear if predictability has anything to do with the theoretical concepts outlined here. As a side note, total “readiness for action” differences between ultradian and circadian oscillating systems might not be as straightforward as well.

We agree that predictability is not related to the Varela hypothesis, and is therefore not mentioned in the paragraph about embodiment because no clear relationship could be theoretically drawn between them.

As now clearly stated in the revised version of the manuscript, the concept of embodiment is proposed as a further interpretation of Analysis N°2, which introduced the idea of a brain-body relationship by considering hormonal and behavioural rhythms together with brain function. Predictability is not mentioned in this paragraph, which now reads:

“Theoretical perspectives

Implications for the consciousness embodiment hypothesis considering Analysis N°2

Interrogating further the harmony between EEG cyclicity, clock-controlled hormonal rhythms (directly driven by the hypothalamic circadian timing system), behaviour (such as the mesencephalic output to the oculomotor nucleus commanding eye-opening), and environment (to assess the reactivity to sensory inputs) may provide a comprehensive understanding of consciousness loss and reappearance, in accordance with a “world-body-brain” holistic perspective¹⁷ (see Fig. 5). F. Varela has introduced the “embodiment of mind” hypothesis¹⁷ in cognitive neuroscience of consciousness.”

Altogether, we chose to assume this rather holistic approach, but we did not expect predictability nor circadian/ultradian rhythms of brain function *per se* to permit any insight on the Varela hypothesis. On the contrary, it is only because our results allowed a decentered view of neurophysiology (by considering the mutual relationship between behaviour and biology), that this philosophical perspective was opened.

8) *One of the main results “enabled to identify a small group of patients who systematically regained consciousness” (line 340 in the discussion) needs additional context. This was concluded based on a 100% PPV, meaning that there are only True Positives and no False Positives. At the same time, sensitivity was low, likely the result of False Negatives. From an ethical, patient-care, perspective, this does not seem like a good result and should be interpreted with more care. The main consideration here is that a False Negative can lead to unwarranted conclusions, with potentially large implications when considering end-of-life*

decisions, while the implication of a False Positive is much less severe. Thus, ethical discussion on the usefulness and implication of this finding is in order.

We agree that this could be an ethical concern, but it is the case for other classical markers used in neuroprognostication such as MMN, P3, and music P3, the sensitivity of which goes sometimes decrescendo while the PPV for awakening is high¹⁸.

We propose to illustrate this legitimate doubt by adding the following paragraph in the Supplementary discussion because the main text is limited in length. However, if the reviewers and the editor allow it, the entire paragraph could be moved as a part of the main manuscript as we agree that it is a very clinically relevant and sensitive topic:

As for previous biomarkers used for neuroprognostication purposes (e.g. novelty P3: Se = 0.71, PPV = 0.81 ; P3b: Se = 0.46, PPV = 0.92 for Fischer et al. in 2008¹⁸), the existence of such a predictive pattern (see Supplementary Table 3) with low sensitivity/NPV but high specificity/VPP for a favourable outcome implies a high risk of false negative results that should be interpreted carefully. The clinical management should not be modified if one fails to demonstrate a “homogeneous presence of all circadian rhythms”, as is it currently unclear whether some patients are not on the verge of further recovery. In the same vein, the historical interpretation of MMN absence was described as a possibly transient phenomenon. Therefore, only positive results should be interpreted as it they have been shown to precede the return of behavioural signs of consciousness within 48h¹⁹.

References:

1. Naccache L. Minimally conscious state or cortically mediated state? *Brain*, (2017).
2. van Schie MKM, Lammers GJ, Fronczek R, Middelkoop HAM, van Dijk JG. Vigilance: discussion of related concepts and proposal for a definition. *Sleep Med* **83**, 175-181 (2021).
3. Gobert F, *et al.* Is circadian rhythmicity a prerequisite to coma recovery? Circadian recovery concomitant to cognitive improvement in two comatose patients. *J Pineal Res* **66**, e12555 (2019).
4. Saper CB, Scammell TE, Lu J. Hypothalamic regulation of sleep and circadian rhythms. *Nature* **437**, 1257-1263 (2005).

5. Steriade M. Grouping of brain rhythms in corticothalamic systems. *Neuroscience* **137**, 1087-1106 (2006).
6. Ouyang G, Li X, Dang C, Richards DA. Using recurrence plot for determinism analysis of EEG recordings in genetic absence epilepsy rats. *Clin Neurophysiol* **119**, 1747-1755 (2008).
7. Rossi Sebastiano D, *et al.* Sleep patterns associated with the severity of impairment in a large cohort of patients with chronic disorders of consciousness. *Clin Neurophysiol* **129**, 687-693 (2018).
8. Valente M, *et al.* Sleep organization pattern as a prognostic marker at the subacute stage of post-traumatic coma. *Clinical neurophysiology : official journal of the International Federation of Clinical Neurophysiology* **113**, 1798-1805 (2002).
9. Moser D, *et al.* Sleep classification according to AASM and Rechtschaffen & Kales: effects on sleep scoring parameters. *Sleep* **32**, 139-149 (2009).
10. Faugeras F, *et al.* Survival and consciousness recovery are better in the minimally conscious state than in the vegetative state. *Brain Inj* **32**, 72-77 (2018).
11. Shannon CE. A Mathematical Theory of Communication. *The Bell System Technical Journal*, (1948).
12. Liang Z, *et al.* EEG entropy measures in anesthesia. *Frontiers in computational neuroscience* **9**, 16 (2015).
13. Thul A, *et al.* EEG entropy measures indicate decrease of cortical information processing in Disorders of Consciousness. *Clinical Neurophysiology* **127**, 1419-1427 (2016).
14. Song Y, Zhang J. Discriminating preictal and interictal brain states in intracranial EEG by sample entropy and extreme learning machine. *J Neurosci Methods* **257**, 45-54 (2016).
15. Saint-Amour D, Lacourse K, Simard M, Lipppe S. Entropy estimation of resting-state EEG variability in amblyopia. *Journal of vision* **15**, 654 (2015).
16. Lau ZJ, Pham T, Chen SHA, Makowski D. Brain entropy, fractal dimensions and predictability: A review of complexity measures for EEG in healthy and neuropsychiatric populations. *Eur J Neurosci* **56**, 5047-5069 (2022).
17. Thompson E, Varela FJ. Radical embodiment: neural dynamics and consciousness. *Trends Cogn Sci* **5**, 418-425 (2001).
18. Fischer C, Dailler F, Morlet D. Novelty P3 elicited by the subject's own name in comatose patients. *Clin Neurophysiol* **119**, 2224-2230 (2008).

19. Kane NM, Curry SH, Butler SR, Cummins BH. Electrophysiological indicator of awakening from coma. *Lancet* **341**, 688 (1993).

REVIEWERS' COMMENTS:

Reviewer #2 (Remarks to the Author):

The authors have addressed all my comments, thank you.

Reviewer #4 (Remarks to the Author):

I was asked by the editorial office to weigh in as an ad-hoc reviewer. I was asked to specifically look at the points 1, 3 as well as 5-7. However, I found the rebuttal letter as well as the manuscript rather cumbersome to read. Generally, my take on the manuscript is that the claims are overdrawn given the data. The authors write that "As the number of EEG parameters was high (168), a supervised data-driven 920 analysis was conducted to find the most relevant 24h long-term variables descriptive 921 of EEG features by using clinical, behavioural, and biological factors." Given the sample size of N=22, this is not a procedure that is advisable and I do not suggest acceptance of this work. I detail the other specific comments below.

Ad 1)

I think this is addressed adequately

Ad 3)

Yes, such values can occur when the power is presented. It is, however, then advisable to present log-transformed values.

Ad 5)

I think that the authors addressed the point. Such similarities can happen. However, the authors may want to present a correlation analysis just to show that the values are not identical (i.e. the correlation is not 1).

Ad 6)

This is indeed a point raised by the reviewer that is also not fully clear to me and the author's response is not very understandable. Of course, the EEG parameters may have differed, but inter-correlations could have been calculated to examine inter-correlations of predictors. This was not done, but should be done. This point has not been addressed adequately. A problem for this analysis, however, is the limited sample size (168 EEG parameters). Generally, I am skeptical about the reliability of the findings. Very many different variables have been tested in a rather limited sample, which leaves the strong impression of a "fishing expedition" with low likelihood of replicability. This is critical given the biomarker character of the study.

Ad 7a)

The authors have addressed this point.

Ad 7b)

The authors have addressed this point.

Ad 7c)

The authors have addressed this point.

Ad 7d)

The authors have addressed this point.

Reviewers' comments:

Reviewer #2 (Remarks to the Author):

The authors have addressed all my comments, thank you.

We thank the reviewer for her/his final acceptance.

Reviewer #4 (Remarks to the Author):

I was asked by the editorial office to weigh in as an ad-hoc reviewer. I was asked to specifically look at the points 1, 3 as well as 5-7. However, I found the rebuttal letter as well as the manuscript rather cumbersome to read. Generally, my take on the manuscript is that the claims are overdrawn given the data. The authors write that "As the number of EEG parameters was high (168), a supervised data-driven analysis was conducted to find the most relevant 24h long-term variables descriptive of EEG features by using clinical, behavioural, and biological factors." Given the sample size of N=22, this is not a procedure that is advisable and I do not suggest acceptance of this work. I detail the other specific comments below.

We want to thank the Reviewer for his complementary remarks after the questions raised by Reviewer N°3. The specific statistical issue raised by point N°6 was answered hereafter.

More generally, we agree that conclusions driven from a limited number of subjects should always been regarded with caution¹. However, we would like to emphasize that statistical power does not only rely on number of subjects, but also on number of points per subjects (on which analyses are computed and compared parameters extracted), inter-individual variability and overall variability of the results (dispersion around the means/medians), and Type-I and Type II error threshold chosen. In addition, ours results, interpretations and conclusions do not rely solely on the supervised data-driven analysis, but also on the independent methodology of Analysis N°2 and their comparison.

Ad 1)

I think this is addressed adequately.

Ad 3)

Yes, such values can occur when the power is presented. It is, however, then advisable to present log-transformed values.

Ad 5)

I think that the authors addressed the point. Such similarities can happen. However, the authors may want to present a correlation analysis just to show that the values are not identical (i.e. the correlation is not 1).

Ad 6)

This is indeed a point raised by the reviewer that is also not fully clear to me and the author's response is not very understandable. Of course, the EEG parameters may have differed, but

inter-correlations could have been calculated to examine inter-correlations of predictors. This was not done, but should be done. This point has not been addressed adequately. A problem for this analysis, however, is the limited sample size (168 EEG parameters).

We apologize for the lack of clarity. Indeed, this previous response did not satisfactorily answer the question raised by Reviewer 4 on the “inter-correlations of predictors” because this initial point 6 was about the correlations of outcome (i.e. in our definition, of factors): “assume these outcome measures are highly correlated, how would you then explain finding different EEG correlates (e.g., if functional outcome and behavioural circadian rhythmicity are highly correlated, you may expect the same EEG correlates to predict them)?” That is why we tried to find out some external arguments (based on the Analysis N°2) to demonstrate which factors of Analysis N°1 were directly related between them or not.

Concerning the inter-correlations between predictors, we performed a dedicated analysis in the initial steps of EEG processing/analysis, to address the question of the “originality” of each EEG features, according to the correlations between time series (two-by-two, Spearman correlations, adjusted P-values based on Holm’s method). The originality of one particular EEG features compared to the other was defined as the “absence of correlation (positive or negative) presenting an adjusted P-values higher than 0.05”.

The results were not shown as they did not appear to have a significant predictive value or to be clinically relevant, but we understand that they could be valuable in the intellectual process of validating a multiparameter data-driven study.

Therefore, this analysis is now included in the supplementary information file (with Supplementary Table 3 and Supplementary data 2), with correlations compared and classified across and within the 4 groups of EEG features (power, spectrum, short-term complexity and spatial variability).

The conclusion of this analysis re-inforces the legitimacy for our complementary selection of features performed statistically and without a priori by this logistic regression (supervised data-driven study corresponding to the Analysis N°1).

The supplementary text now reads (p22):

“Altogether, it appeared that the highest number of original data could be obtained by the comparison between spatial variability and the three other groups (spectrum, then short-term complexity, then power).

This result is in line with the Analysis N°1. It corroborates in particular the validity of the choice provided by the data-driven analysis in which the models for the 5 factors have selected EEG parameters related to:

- spatial variability and spectrum in 3 cases (“Behavioural circadian rhythmicity”, “Hormonal circadian rhythmicity” and “Functional outcome” factors).
- spatial variability and complexity in 1 case (“Disorders of consciousness” factor).

- spatial variability, spectrum and power in 1 case (“Behavioural cortical Function” factor)

They are also in accordance with the Analysis N°2 as the 4 EEG features whose circadian (24h) rhythmicity was regarded as normal (among healthy subjects: Alpha and Beta Spatial Variability, Determinism and DFA) were also selected among the most original comparison between EEG features:

- at the group-level (spatial variability and complexity, see above)
- at the feature-level (Spatial Variability of the Alpha band and Determinism, see above).”

Generally, I am skeptical about the reliability of the findings. Very many different variables have been tested in a rather limited sample, which leaves the strong impression of a “fishing expedition” with low likelihood of replicability. This is critical given the biomarker character of the study.

To answer this legitimate cause of scepticism concerning the risk of low replicability and more generally the risk of overfitting bias (related to the concept of “fishing expedition”), another paragraph has been added in the Discussion section (paragraph concerning the study limitations, which has been removed from the Supplementary Information document to the main manuscript).

The manuscript now reads (p21):

“However, this overfitting bias remained in absence of replication by a validation cohort.

Therefore, the interpretation of these results in the clinical context should be taken with caution, with a risk of overinterpreting non-replicated EEG data from Analysis N°1 as biomarkers with potentially harmful consequences for future patients. Therefore, instead, we used the results from the Analysis N°2 in the comparison to current neuro-prognostication tools. Nonetheless, despite the statistical weakness related to the limited sample size and the exploratory nature of the analysis, an indirect argument in favour of the consistency of our results consists in their redundancy through different approaches. Indeed, we emphasise the place of fluctuations: i) for EEG only in a hypothesis-free paradigm (Analysis N°1); ii) for

multimodal metrics in a hypothesis-driven approach focused on the circadian system
(Analysis N°2).”

The biomarker nature of this finding was also modulated in another part of the discussion.
The manuscript now reads (p24):

“Concerning the clinical interpretation, it is of note that the results from analysis N°1 were not proposed as a biomarker for prognosis in absence of a sufficient sample to allow a validation cohort.”

In addition, two paragraphs were added in the Methods section in relation with this remark.
The manuscript now reads (p42):

“Concerning reproducibility, due to the exploratory nature of the study and the difficulties to tackle synchronously each recording in an ICU environment for rare patients, no replicates were performed.”

(...)

“The choice of metrics was based on their mathematical complementary to describe several EEG dimensions of fluctuations in the short and long-term perspectives without a priori. It should be noted that this strategy was appropriate to the exploratory nature of this work as the aim was not to confirm the superiority of a candidate biomarker. It was rather to define which group of parameters – assessing either mean values of EEG feature or the description of these features’ fluctuations over 24h – contained more information associated with a close-to-physiology brain functioning after injury.”

Ad 7a)

The authors have addressed this point.

Ad 7b)

The authors have addressed this point.

Ad 7c)

The authors have addressed this point.

Ad 7d)

The authors have addressed this point.

- 1. Münch M, Raab C, Biehl M, Schleif F-M. Data-Driven Supervised Learning for Life Science Data. *Frontiers in Applied Mathematics and Statistics* **6**, (2020).
-